# On Exact Computation with an Infinitely Wide Neural Net[*]

Sanjeev Arora[†]        Simon S. Du[‡]        Wei Hu[§]        Zhiyuan Li[¶]

Ruslan Salakhutdinov[‖]                Ruosong Wang[**]

## Abstract

How well does a classic deep net architecture like AlexNet or VGG19 classify on a standard dataset such as CIFAR-10 when its "width"— namely, number of channels in convolutional layers, and number of nodes in fully-connected internal layers — is allowed to increase to infinity? Such questions have come to the forefront in the quest to theoretically understand deep learning and its mysteries about optimization and generalization. They also connect deep learning to notions such as *Gaussian processes* and *kernels*. A recent paper [Jacot et al., 2018] introduced the *Neural Tangent Kernel (NTK)* which captures the behavior of fully-connected deep nets in the infinite width limit trained by gradient descent; this object was implicit in some other recent papers. An attraction of such ideas is that a pure kernel-based method is used to capture the power of a fully-trained deep net of infinite width.

The current paper gives the first efficient exact algorithm for computing the extension of NTK to convolutional neural nets, which we call *Convolutional NTK (CNTK)*, as well as an efficient GPU implementation of this algorithm. This results in a significant new benchmark for performance of a pure kernel-based method on CIFAR-10, being 10% higher than the methods reported in [Novak et al., 2019], and only 6% lower than the performance of the corresponding finite deep net architecture (once batch normalization etc. are turned off). Theoretically, we also give the first *non-asymptotic* proof showing that a fully-trained sufficiently wide net is indeed equivalent to the kernel regression predictor using NTK.

## 1   Introduction

How well does a classic deep net architecture like AlexNet or VGG19 perform on a standard dataset such as CIFAR-10 when its "width"— namely, number of channels in convolutional layers, and number of nodes in fully-connected internal layers — is allowed to increase to infinity? Questions about these "infinite limits" of deep nets have naturally emerged in the ongoing effort to understand the power of deep learning. In mathematics it is often easier to study objects in the infinite limit. Furthermore, the infinite limit could conceivably make sense in deep learning, since *over-parametrization* seems to help optimization a lot and doesn't hurt generalization much [Zhang et al., 2017]: deep neural nets with millions of parameters work well even for datasets with 50k training examples. So why not imagine nets whose width goes to infinity?

---

[*]The latest full version of this paper can be found at https://arxiv.org/abs/1904.11955.

[†]Princeton University and Institute for Advanced Study. Email: arora@cs.princeton.edu

[‡]Institute for Advanced Study. Email: ssdu@ias.edu

[§]Princeton University. Email: huwei@cs.princeton.edu

[¶]Princeton University. Email: zhiyuanli@cs.princeton.edu

[‖]Carnegie Mellon University. Email:rsalakhu@cs.cmu.edu

[**]Carnegie Mellon University. Email: ruosongw@andrew.cmu.edu

Allowing width to go to infinity also connects deep learning in an interesting way with other areas of machine learning. A single hidden-layer neural network with i.i.d. random parameters, in the limit of infinite width, is a function drawn from a *Gaussian process (GP)* [Neal, 1996]. This model as well as analogous ones with multiple layers [Lee et al., 2018, Matthews et al., 2018], convolutional filters [Novak et al., 2019, Garriga-Alonso et al., 2019] and other architectures [Yang, 2019] make up the GP view of deep learning. These correspond to infinitely wide deep nets whose all parameters are chosen randomly (with careful scaling), and only the top (classification) layer is optimized.

From now on we will use *weakly-trained nets* to refer to nets whose layers receive random initialization and only the top layer is trained by gradient descent. We use *fully-trained* to refer to nets whose all parameters are trained by gradient descent. It has long been known that weakly-trained convolutional nets have reasonable performance on MNIST and CIFAR-10. Weakly-trained nets that are fully-connected instead of convolutional, can also be thought of as "multi-layer random kitchen sinks," which also have a long history.

Weakly-trained nets — whether of finite or infinite width — also define interesting kernels. Specifically, if $f(\boldsymbol{\theta}, \boldsymbol{x}) \in \mathbb{R}$ denotes the output of the network on input $\boldsymbol{x}$ where $\boldsymbol{\theta}$ denotes the parameters in the network, and $\mathcal{W}$ is an initialization distribution over $\boldsymbol{\theta}$ (usually Gaussian), then training just the top layer with an $\ell_2$ loss is equivalent to kernel regression for the following kernel:

$$\ker(\boldsymbol{x}, \boldsymbol{x}') = \mathop{\mathbb{E}}_{\boldsymbol{\theta} \sim \mathcal{W}}[f(\boldsymbol{\theta}, \boldsymbol{x}) \cdot f(\boldsymbol{\theta}, \boldsymbol{x}')], \tag{1}$$

where $\boldsymbol{x}, \boldsymbol{x}'$ are two inputs. This kernel method makes sense when the width goes to infinity.

The objects of interest in this paper are not weakly-trained nets, but fully-trained nets. In the finite case, analysis of optimization and generalization of fully-trained nets is of course an open problem. One may also ask:

*Can we understand the power of fully-trained nets whose width goes to infinity?*

*A priori* this question doesn't seem any easier than the finite case, and empirical evaluation seems computationally infeasible due to the infinite limit. They also do not correspond to a kernel method in any obvious way.

Recent papers suggest that neural nets whose width greatly exceeds the number of training data points can rapidly reduce training error to 0 via gradient descent, and under some conditions, the trained net also exhibits good generalization [Du et al., 2019, 2018b, Li and Liang, 2018, Allen-Zhu et al., 2018a,b, Zou et al., 2018, Arora et al., 2019, Cao and Gu, 2019]. Extra-wideness plays a crucial role in the proof: it is shown that as width increases, training causes increasingly smaller changes (in a proportionate sense) in the parameters. This raises the possibility that as one increases the width to infinity, a certain limiting behavior can emerge even in the fully-trained net. A recent paper by Jacot et al. [2018] isolated a notion implicit in the above papers, which they called the *Neural Tangent Kernel (NTK)*. They suggested — via a proof that is slightly heuristic — that this fixed kernel characterizes the behavior of fully-connected infinite width neural networks whose layers have been trained by gradient descent. The NTK is different from the Gaussian process kernels discussed earlier, and is defined using the *gradient* of the output of the randomly initialized net with respect to its parameters, i.e.,

$$\ker(\boldsymbol{x}, \boldsymbol{x}') = \mathop{\mathbb{E}}_{\boldsymbol{\theta} \sim \mathcal{W}} \left\langle \frac{\partial f(\boldsymbol{\theta}, \boldsymbol{x})}{\partial \boldsymbol{\theta}}, \frac{\partial f(\boldsymbol{\theta}, \boldsymbol{x}')}{\partial \boldsymbol{\theta}} \right\rangle. \tag{2}$$

Here, the gradient $\frac{\partial f(\boldsymbol{\theta}, \boldsymbol{x})}{\partial \boldsymbol{\theta}}$ appears from considering gradient descent, as will be explained in Section 3. One may also generalize the NTK to convolutional neural nets, and we call the corresponding kernel *Convolutional Neural Tangent Kernel (CNTK)*.

Though NTK and CNTK are defined by an infinite limit, a recent paper [Lee et al., 2019] attempted to understand their properties via a finite approximation of the infinite limit kernel by Monte Carlo methods. However, as will be shown in Section B, using random features generated from practically sized nets can degrade the performance a lot. It was still open what is the full power of *exact* CNTK on modern datasets. This is a challenging question especially for CNTK with pooling operations, since when convolution with pooling is involved, it was believed that exact computation of kernels (for either convolutional Gaussian process kernel or CNTK) is infeasible for large datasets like CIFAR-10 [Novak et al., 2019].

**Our contributions.** We give an exact and efficient dynamic programming algorithm to compute CNTKs for ReLU activation (namely, to compute $\ker(\boldsymbol{x}, \boldsymbol{x}')$ given $\boldsymbol{x}$ and $\boldsymbol{x}'$). Using this algorithm — as well as implementation tricks for GPUs — we can settle the question of the performance of fully-trained infinitely wide nets with a variety of architectures. For instance, we find that their performance on CIFAR-10 is within $5\%$ of the performance of the same architectures in the finite case (note that the proper comparison in the finite case involves turning off batch norm, data augmentation, etc., in the optimization). In particular, the CNTK corresponding to a 11-layer convolutional net with global average pooling achieves $77\%$ classification accuracy. This is $10\%$ higher than the best reported performance of a Gaussian process with fixed kernel on CIFAR-10 [Novak et al., 2019].[8]

Furthermore, we give a more rigorous, non-asymptotic proof that the NTK captures the behavior of a fully-trained wide neural net under weaker condition than previous proofs. We also experimentally show that the random feature methods for approximating CNTK in earlier work do not compute good approximations, which is clear from their much worse performance on CIFAR.

## 1.1 Notation

We use bold-faced letters for vectors, matrices and tensors. For a vector $\boldsymbol{a}$, let $[\boldsymbol{a}]_i$ be its $i$-th entry; for a matrix $\boldsymbol{A}$, let $[\boldsymbol{A}]_{i,j}$ be its $(i, j)$-th entry; for a 4th-order tensor $\boldsymbol{T}$, let $[\boldsymbol{A}]_{ij,i'j'}$ be its $(i, j, i', j')$-th entry. Let $\boldsymbol{I}$ be the identity matrix, and $[n] = \{1, 2, \ldots, n\}$. Let $\boldsymbol{e}_i$ be an indicator vector with $i$-th entry being $1$ and other entries being $0$, and let $\boldsymbol{1}$ denote the all-one vector. We use $\odot$ to denote the entry-wise product and $\otimes$ to denote the tensor product. We use $\langle \cdot, \cdot \rangle$ to denote the standard inner product. We use $\mathrm{diag}(\cdot)$ to transform a vector to a diagonal matrix. We use $\sigma(\cdot)$ to denote the activation function, such as the rectified linear unit (ReLU) function: $\sigma(z) = \max\{z, 0\}$, and $\dot{\sigma}(\cdot)$ to denote the derivative of $\sigma(\cdot)$. Denote by $\mathcal{N}(\boldsymbol{\mu}, \boldsymbol{\Sigma})$ the Gaussian distribution with mean $\boldsymbol{\mu}$ and covariance $\boldsymbol{\Sigma}$.

## 2 Related Work

From a Gaussian process (GP) viewpoint, the correspondence between infinite neural networks and kernel machines was first noted by Neal [1996]. Follow-up work extended this correspondence to more general shallow neural networks [Williams, 1997, Roux and Bengio, 2007, Hazan and Jaakkola, 2015]. More recently, this was extended to deep and convolutional neural networks [Lee et al., 2018, Matthews et al., 2018, Novak et al., 2019, Garriga-Alonso et al., 2019] and a variety of other architectures [Yang, 2019]. However, these kernels, as we discussed in Section 1, represent weakly-trained nets, instead of fully-trained nets.

Beyond GPs, the connection between neural networks and kernels is also studied in the compositional kernel literature. Cho and Saul [2009] derived a closed-form kernel formula for rectified polynomial activations, which include ReLU as a special case. Daniely et al. [2016] proposed a general framework to transform a neural network to a compositional kernel and later Daniely [2017] showed for sufficiently wide neural networks, stochastic gradient descent can learn functions that lie in the corresponding reproducing kernel Hilbert space. However, the kernels studied in these works still correspond to weakly-trained neural networks.

This paper is inspired by a line of recent work on over-parameterized neural networks [Du et al., 2019, 2018b, Du and Hu, 2019, Li and Liang, 2018, Allen-Zhu et al., 2018b,a, Zou et al., 2018, Cao and Gu, 2019]. These papers established that for (convolutional) neural networks with large but finite width, (stochastic) gradient descent can achieve zero training error. A key component in these papers is showing that the weight matrix at each layer is close to its initialization. This observation implies that the kernel defined in Equation (2) is still close to its initialization. Arora et al. [2019] explicitly used this observation to derive generalization bounds for two-layer over-parameterized neural networks. Chizat and Bach [2018] argued that these results in the kernel regime may be too simple to be able to explain the success of deep learning, while on the other hand, out results show that CNTK is at least able to perform well on tasks like CIFAR-10 classification. Also see the survey Fan et al. [2019] for recent advance in deep learning theory.

Jacot et al. [2018] derived the exact same kernel from kernel gradient descent. They showed that if the number of neurons per layer goes to infinity in a sequential order, then the kernel remains unchanged for a finite training time. They termed the derived kernel *Neural Tangent Kernel (NTK)*. We follow the same naming convention and name its convolutional extension *Convolutional Neural Tangent Kernel (CNTK)*. Later, Yang [2019] derived a formula of CNTK as well as a mechanistic way to derive NTK for different architectures. Comparing with [Yang, 2019], our CNTK formula has a more explicit convolutional structure and results in an efficient GPU-friendly computation method. Recently, Lee et al. [2019] tried to empirically verify the theory in [Jacot et al., 2018] by studying the linearization of neural nets. They observed that in the first few iterations, the linearization is close to the actual neural net. However, as will be shown in Section B, such linearization can decrease the classification accuracy by $5\%$ even on a "CIFAR-2" (airplane V.S. car) dataset. Therefore, exact kernel evaluation is important to study the power of NTK and CNTK.

# 3  Neural Tangent Kernel

In this section we describe fully-connected deep neural net architecture and its infinite width limit, and how training it with respect to the $\ell_2$ loss gives rise to a kernel regression problem involving the neural tangent kernel (NTK). We denote by $f(\boldsymbol{\theta}, \boldsymbol{x}) \in \mathbb{R}$ the output of a neural network where $\boldsymbol{\theta} \in \mathbb{R}^N$ is all the parameters in the network and $\boldsymbol{x} \in \mathbb{R}^d$ is the input.[9] Given a training dataset $\{(\boldsymbol{x}_i, y_i)\}_{i=1}^n \subset \mathbb{R}^d \times \mathbb{R}$, consider training the neural network by minimizing the squared loss over training data: $\ell(\boldsymbol{\theta}) = \frac{1}{2} \sum_{i=1}^n (f(\boldsymbol{\theta}, \boldsymbol{x}_i) - y_i)^2$. The proof of the following lemma uses simple differentiation and appears in Section C.

**Lemma 3.1.** *Consider minimizing the squared loss $\ell(\boldsymbol{\theta})$ by gradient descent with infinitesimally small learning rate: $\frac{\mathrm{d}\boldsymbol{\theta}(t)}{\mathrm{d}t} = -\nabla \ell(\boldsymbol{\theta}(t))$. Let $\boldsymbol{u}(t) = (f(\boldsymbol{\theta}(t), \boldsymbol{x}_i))_{i \in [n]} \in \mathbb{R}^n$ be the network outputs on all $\boldsymbol{x}_i$'s at time $t$, and $\boldsymbol{y} = (y_i)_{i \in [n]}$ be the desired outputs. Then $\boldsymbol{u}(t)$ follows the following evolution, where $\boldsymbol{H}(t)$ is an $n \times n$ positive semidefinite matrix whose $(i,j)$-th entry is $\left\langle \frac{\partial f(\boldsymbol{\theta}(t), \boldsymbol{x}_i)}{\partial \boldsymbol{\theta}}, \frac{\partial f(\boldsymbol{\theta}(t), \boldsymbol{x}_j)}{\partial \boldsymbol{\theta}} \right\rangle$:*

$$\frac{\mathrm{d}\boldsymbol{u}(t)}{\mathrm{d}t} = -\boldsymbol{H}(t) \cdot (\boldsymbol{u}(t) - \boldsymbol{y}). \tag{3}$$

The statement of Lemma 3.1 involves a matrix $\boldsymbol{H}(t)$. Below we define a deep net architecture whose width is allowed to go to infinity, while fixing the training data as above. In the limit, it can be shown that the matrix $\boldsymbol{H}(t)$ remains *constant* during training i.e., equal to $\boldsymbol{H}(0)$. Moreover, under a random initialization of parameters, the random matrix $\boldsymbol{H}(0)$ converges in probability to a certain deterministic kernel matrix $\boldsymbol{H}^*$ as the width goes to infinity, which is the *Neural Tangent Kernel* $\ker(\cdot, \cdot)$ (Equation (2)) evaluated on the training data. If $\boldsymbol{H}(t) = \boldsymbol{H}^*$ for all $t$, then Equation (3) becomes

$$\frac{\mathrm{d}\boldsymbol{u}(t)}{\mathrm{d}t} = -\boldsymbol{H}^* \cdot (\boldsymbol{u}(t) - \boldsymbol{y}). \tag{4}$$

Note that the above dynamics is identical to the dynamics of *kernel regression* under gradient flow, for which at time $t \to \infty$ the final prediction function is (assuming $\boldsymbol{u}(0) = \boldsymbol{0}$)

$$f^*(\boldsymbol{x}) = (\ker(\boldsymbol{x}, \boldsymbol{x}_1), \ldots, \ker(\boldsymbol{x}, \boldsymbol{x}_n)) \cdot (\boldsymbol{H}^*)^{-1} \boldsymbol{y}. \tag{5}$$

In Theorem 3.2, we rigorously prove that a fully-trained sufficiently wide ReLU neural network is equivalent to the kernel regression predictor (5) on any given data point.

**Fully-connected deep neural net and its infinite width limit.**    Now we define a fully-connected neural net formally. Let $\boldsymbol{x} \in \mathbb{R}^d$ be the input, and denote $\boldsymbol{g}^{(0)}(\boldsymbol{x}) = \boldsymbol{x}$ and $d_0 = d$ for notational convenience. We define an $L$-hidden-layer fully-connected neural network recursively:

$$\boldsymbol{f}^{(h)}(\boldsymbol{x}) = \boldsymbol{W}^{(h)} \boldsymbol{g}^{(h-1)}(\boldsymbol{x}) \in \mathbb{R}^{d_h}, \quad \boldsymbol{g}^{(h)}(\boldsymbol{x}) = \sqrt{\frac{c_\sigma}{d_h}} \sigma \left( \boldsymbol{f}^{(h)}(\boldsymbol{x}) \right) \in \mathbb{R}^{d_h}, \qquad h = 1, 2, \ldots, L,$$
$$\tag{6}$$

where $\boldsymbol{W}^{(h)} \in \mathbb{R}^{d_h \times d_{h-1}}$ is the weight matrix in the $h$-th layer ($h \in [L]$), $\sigma : \mathbb{R} \to \mathbb{R}$ is a coordinate-wise activation function, and $c_\sigma = \left( \mathbb{E}_{z \sim \mathcal{N}(0,1)} \left[ \sigma\left(z\right)^2 \right] \right)^{-1}$. The last layer of the neural network is

$$
\begin{aligned}
f(\boldsymbol{\theta}, \boldsymbol{x}) = f^{(L+1)}(\boldsymbol{x}) &= \boldsymbol{W}^{(L+1)} \cdot \boldsymbol{g}^{(L)}(\boldsymbol{x}) \\
&= \boldsymbol{W}^{(L+1)} \cdot \sqrt{\frac{c_\sigma}{d_L}} \sigma \left( \boldsymbol{W}^{(L)} \cdot \sqrt{\frac{c_\sigma}{d_{L-1}}} \sigma \left( \boldsymbol{W}^{(L-1)} \cdots \sqrt{\frac{c_\sigma}{d_1}} \sigma \left( \boldsymbol{W}^{(1)} \boldsymbol{x} \right) \right) \right),
\end{aligned}
$$

where $\boldsymbol{W}^{(L+1)} \in \mathbb{R}^{1 \times d_L}$ is the weights in the final layer, and $\boldsymbol{\theta} = \left( \boldsymbol{W}^{(1)}, \dots, \boldsymbol{W}^{(L+1)} \right)$ represents all the parameters in the network.

We initialize all the weights to be i.i.d. $\mathcal{N}(0,1)$ random variables, and consider the limit of large hidden widths: $d_1, d_2, \dots, d_L \to \infty$. The scaling factor $\sqrt{c_\sigma / d_h}$ in Equation (6) ensures that the norm of $\boldsymbol{g}^{(h)}(\boldsymbol{x})$ for each $h \in [L]$ is approximately preserved at initialization (see [Du et al., 2018b]). In particular, for ReLU activation, we have $\mathbb{E} \left[ \left\| \boldsymbol{g}^{(h)}(\boldsymbol{x}) \right\|^2 \right] = \|\boldsymbol{x}\|^2$ ($\forall h \in [L]$).

Recall from [Lee et al., 2018] that in the infinite width limit, the pre-activations $\boldsymbol{f}^{(h)}(\boldsymbol{x})$ at every hidden layer $h \in [L]$ has all its coordinates tending to i.i.d. centered Gaussian processes of covariance $\Sigma^{(h-1)} : \mathbb{R}^d \times \mathbb{R}^d \to \mathbb{R}$ defined recursively as: for $h \in [L]$,

$$
\begin{aligned}
\Sigma^{(0)}(\boldsymbol{x}, \boldsymbol{x}') &= \boldsymbol{x}^\top \boldsymbol{x}', \\
\boldsymbol{\Lambda}^{(h)}(\boldsymbol{x}, \boldsymbol{x}') &= \begin{pmatrix} \Sigma^{(h-1)}(\boldsymbol{x}, \boldsymbol{x}) & \Sigma^{(h-1)}(\boldsymbol{x}, \boldsymbol{x}') \\ \Sigma^{(h-1)}(\boldsymbol{x}', \boldsymbol{x}) & \Sigma^{(h-1)}(\boldsymbol{x}', \boldsymbol{x}') \end{pmatrix} \in \mathbb{R}^{2 \times 2}, \\
\Sigma^{(h)}(\boldsymbol{x}, \boldsymbol{x}') &= c_\sigma \mathop{\mathbb{E}}_{(u,v) \sim \mathcal{N}\left(\boldsymbol{0}, \boldsymbol{\Lambda}^{(h)}\right)} \left[ \sigma\left(u\right) \sigma\left(v\right) \right].
\end{aligned} \tag{7}
$$

To give the formula of NTK, we also need to define a derivative covariance:

$$
\dot{\Sigma}^{(h)}(\boldsymbol{x}, \boldsymbol{x}') = c_\sigma \mathop{\mathbb{E}}_{(u,v) \sim \mathcal{N}\left(\boldsymbol{0}, \boldsymbol{\Lambda}^{(h)}\right)} \left[ \dot{\sigma}(u) \dot{\sigma}(v) \right]. \tag{8}
$$

The final NTK expression for the fully-connected neural network is

$$
\Theta^{(L)}(\boldsymbol{x}, \boldsymbol{x}') = \sum_{h=1}^{L+1} \left( \Sigma^{(h-1)}(\boldsymbol{x}, \boldsymbol{x}') \cdot \prod_{h'=h}^{L+1} \dot{\Sigma}^{(h')}(\boldsymbol{x}, \boldsymbol{x}') \right), \tag{9}
$$

where we let $\dot{\Sigma}^{(L+1)}(\boldsymbol{x}, \boldsymbol{x}') = 1$ for convenience. We refer readers to Section D for the derivation of this formula. Rigorously, for ReLU activation, we have the following theorem that gives a concrete bound on the hidden widths that is sufficient for convergence to the NTK at initialization:

**Theorem 3.1** (Convergence to the NTK at initializatoin). *Fix $\epsilon > 0$ and $\delta \in (0, 1)$. Suppose $\sigma\left(z\right) = \max(0, z)$ and $\min_{h \in [L]} d_h \geq \Omega(\frac{L^{14}}{\epsilon^4} \log(L/\delta))$. Then for any inputs $\boldsymbol{x}, \boldsymbol{x}' \in \mathbb{R}^{d_0}$ such that $\|\boldsymbol{x}\| \leq 1, \|\boldsymbol{x}'\| \leq 1$, with probability at least $1 - \delta$ we have:*

$$
\left| \left\langle \frac{\partial f(\boldsymbol{\theta}, \boldsymbol{x})}{\partial \boldsymbol{\theta}}, \frac{\partial f(\boldsymbol{\theta}, \boldsymbol{x}')}{\partial \boldsymbol{\theta}} \right\rangle - \Theta^{(L)}(\boldsymbol{x}, \boldsymbol{x}') \right| \leq \epsilon.
$$

The proof of Theorem 3.1 is given in Section E. Theorem 3.1 improves upon previous results [Jacot et al., 2018, Yang, 2019] that also established similar convergence in the following sense:

1. Previous results are asymptotic, i.e., they require the widths to go to infinity, while Theorem 3.1 gives a non-asymptotic bound on the required layer widths.
2. Jacot et al. [2018] required sequential limit, i.e., $d_1, \dots, d_L$ go to infinity one by one, and Yang [2019] let $d_1, \dots, d_L$ go to infinity at the same rate. On the other hand, Theorem 3.1 only requires $\min_{h \in [L]} d_h$ to be sufficiently large, which is the weakest notion of limit.

**Equivalence between wide neural net and kernel regression with NTK.** Built on Theorem 3.1, we can further incorporate the training process and show the equivalence between a fully-trained sufficiently wide neural net and the kernel regression solution using the NTK, as described in Lemma 3.1 and the discussion after it.

Recall that the training data are $\{(\boldsymbol{x}_i, y_i)\}_{i=1}^n \subset \mathbb{R}^d \times \mathbb{R}$, and $\boldsymbol{H}^* \in \mathbb{R}^{n \times n}$ is the NTK evaluated on these training data, i.e., $[\boldsymbol{H}^*]_{i,j} = \Theta^{(L)}(\boldsymbol{x}_i, \boldsymbol{x}_j)$. Denote $\lambda_0 = \lambda_{\min}(\boldsymbol{H}^*)$. For a testing point $\boldsymbol{x}_{te} \in \mathbb{R}^d$, we let $\ker_{ntk}(\boldsymbol{x}_{te}, \boldsymbol{X}) \in \mathbb{R}^n$ be the kernel evaluated between the testing point and $n$ training points, i.e., $[\ker_{ntk}(\boldsymbol{x}_{te}, \boldsymbol{X})]_i = \Theta^{(L)}(\boldsymbol{x}_{te}, \boldsymbol{x}_i)$. The prediction of kernel regression using NTK on this testing point is $f_{ntk}(\boldsymbol{x}_{te}) = (\ker_{ntk}(\boldsymbol{x}_{te}, \boldsymbol{X}))^\top (\boldsymbol{H}^*)^{-1} \boldsymbol{y}$.

Since the above solution corresponds to the linear dynamics in Equation (4) with zero initialization, in order to establish equivalence between neural network and kernel regression, we would like the initial output of the neural network to be small. Therefore, we apply a small multiplier $\kappa > 0$, and let the final output of the neural network be $f_{nn}(\boldsymbol{\theta}, \boldsymbol{x}) = \kappa f(\boldsymbol{\theta}, \boldsymbol{x})$. We let $f_{nn}(\boldsymbol{x}_{te}) = \lim_{t \to \infty} f_{nn}(\boldsymbol{\theta}(t), \boldsymbol{x}_{te})$ be the prediction of the neural network at the end of training.

The following theorem establishes the equivalence between the fully-trained wide neural network $f_{nn}$ and the kernel regression predictor $f_{ntk}$ using the NTK.

**Theorem 3.2** (Equivalence between trained net and kernel regression). *Suppose* $\sigma(z) = \max(0, z)$, $1/\kappa = \text{poly}(1/\epsilon, \log(n/\delta))$ *and* $d_1 = d_2 = \cdots = d_L = m$ *with* $m \geq \text{poly}(1/\kappa, L, 1/\lambda_0, n, \log(1/\delta))$. *Then for any* $\boldsymbol{x}_{te} \in \mathbb{R}^d$ *with* $\|\boldsymbol{x}_{te}\| = 1$, *with probability at least* $1 - \delta$ *over the random initialization, we have*

$$|f_{nn}(\boldsymbol{x}_{te}) - f_{ntk}(\boldsymbol{x}_{te})| \leq \epsilon.$$

The proof of Theorem 3.2 is given in Section F. We remark that one can generalize our proof to more advanced architectures, such as convolutinal neural network, ResNet, etc.

Theorem 3.2 is, to our knowledge, the first result that rigorously shows the equivalence between a fully-trained neural net and a deterministic kernel predictor. Compared with similar results by [Jacot et al., 2018, Lee et al., 2019], our bound is non-asymptotic whereas theirs are asymptotic. Compared with [Arora et al., 2019, Allen-Zhu et al., 2018b,a, Du et al., 2019, 2018b, Li and Liang, 2018, Zou et al., 2018], our theorem is a more precise characterization of the learned neural network. That is, the prediction is essentially a kernel predictor. Therefore, to study the properties of over-parameterized nets, such as their generalization power, it is sufficient to study the corresponding NTK.

While this theorem only gives guarantee for a single point, using a union bound, we can show that this guarantee holds for (exponentially many) finite testing points. Combing this with the standard analysis of hold-out validation set, we can conclude that a fully-trained wide neural net enjoys the same generalization ability as its corresponding NTK.

# 4 Convolutional Neural Tangent Kernel

In this section we study convolutional neural nets (CNNs) and their corresponding CNTKs. We study two architectures, vanilla CNN and CNN with global average pooling (GAP). In this section we define vanilla CNN and present its corresponding CNTK formula. The derivation of this formula is deferred to Section G. We present the definition of CNN with GAP and its CNTK in Section H.

To formally define CNNs, we first introduce some notation. We let $P$ be the width and $Q$ be the height of the image. We use $q \in \mathbb{Z}_+$ to denote the filter size. In practice, $q = 1, 3$, or $5$. We use standard zero padding and set stride size to be $1$ to make sure the input of each layer has the same size. For a convolutional filter $\boldsymbol{w} \in \mathbb{R}^{q \times q}$ and an image $\boldsymbol{x} \in \mathbb{R}^{P \times Q}$, the convolution operator is defined as

$$[\boldsymbol{w} * \boldsymbol{x}]_{ij} = \sum_{a=-\frac{q-1}{2}}^{\frac{q-1}{2}} \sum_{b=-\frac{q-1}{2}}^{\frac{q-1}{2}} [\boldsymbol{w}]_{a+\frac{q+1}{2}, b+\frac{q+1}{2}} [\boldsymbol{x}]_{a+i, b+j} \text{ for } i \in [P], j \in [Q]. \quad (10)$$

Equation (10) shows that patch $[\boldsymbol{w} * \boldsymbol{x}]_{ij}$ depends on $[\boldsymbol{x}]_{i-\frac{q-1}{2}:i+\frac{q-1}{2}, j-\frac{q-1}{2}:j+\frac{q-1}{2}}$. Our CNTK formula also relies on this dependency. For $(i, j, i', j') \in [P] \times [Q] \times [P] \times [Q]$, define

$$\mathcal{D}_{ij,i'j'}$$
$$= \{(i+a, j+b, i'+a', j'+b') \in [P] \times [Q] \times [P] \times [Q] \mid -(q-1)/2 \leq a, b, a', b' \leq (q-1)/2\}.$$

Lastly, for a tensor $\boldsymbol{T} \in \mathbb{R}^{P \times Q \times P \times Q}$, we denote by $[\boldsymbol{T}]_{\mathcal{D}_{ij,i'j'}} \in \mathbb{R}^{q \times q \times q \times q}$ a sub-tensor and we let $\text{tr}(\boldsymbol{T}) = \sum_{i,j} \boldsymbol{T}_{ij,ij}$.

A vanilla CNN consisting of $L$ convolution layers and one fully-connected layer is formally defined as follows:

- Let $\boldsymbol{x}^{(0)} = \boldsymbol{x} \in \mathbb{R}^{P \times Q \times C^{(0)}}$ be the input image where $C^{(0)}$ is the number of channels.
- For $h = 1, \ldots, L, \beta = 1, \ldots, C^{(h)}$, the intermediate outputs are defined as

$$
\tilde{\boldsymbol{x}}^{(h)}_{(\beta)} = \sum_{\alpha=1}^{C^{(h-1)}} \boldsymbol{W}^{(h)}_{(\alpha),(\beta)} * \boldsymbol{x}^{(h-1)}_{(\alpha)}, \quad \boldsymbol{x}^{(h)}_{(\beta)} = \sqrt{\frac{c_\sigma}{C^{(h)} \times q \times q}} \sigma\left(\tilde{\boldsymbol{x}}^{(h)}_{(\beta)}\right),
$$

where each $\boldsymbol{W}^{(h)}_{(\alpha),(\beta)} \in \mathbb{R}^{q \times q}$ is a filter with standard Gaussian initialization.

- The final output is defined as $f(\boldsymbol{\theta}, \boldsymbol{x}) = \sum_{\alpha=1}^{C^{(L)}} \left\langle \boldsymbol{W}^{(L+1)}_{(\alpha)}, \boldsymbol{x}^{(L)}_{(\alpha)} \right\rangle$ where $\boldsymbol{W}^{(L+1)}_{(\alpha)} \in \mathbb{R}^{P \times Q}$ is a weight matrix with standard Gaussian initialization.

For this architecture, using the same reasoning as in Section D, we obtain the following convolutional neural tangent kernel formula. The details are provided in Section G.

**CNTK formula.** We let $\boldsymbol{x}, \boldsymbol{x}'$ be two input images.

- For $\alpha = 1, \ldots, C^{(0)}, (i, j, i', j') \in [P] \times [Q] \times [P] \times [Q]$, define

$$
\boldsymbol{K}^{(0)}_{(\alpha)}(\boldsymbol{x}, \boldsymbol{x}') = \boldsymbol{x}_{(\alpha)} \otimes \boldsymbol{x}'_{(\alpha)} \text{ and } \left[\boldsymbol{\Sigma}^{(0)}(\boldsymbol{x}, \boldsymbol{x}')\right]_{ij,i'j'} = \sum_{\alpha=1}^{C^{(0)}} \text{tr}\left(\left[\boldsymbol{K}^{(0)}_{(\alpha)}(\boldsymbol{x}, \boldsymbol{x}')\right]_{\mathcal{D}_{ij,i'j'}}\right).
$$

- For $h \in [L]$,
  - For $(i, j, i', j') \in [P] \times [Q] \times [P] \times [Q]$, define

$$
\boldsymbol{\Lambda}^{(h)}_{ij,i'j'}(\boldsymbol{x}, \boldsymbol{x}') = \begin{pmatrix} \left[\boldsymbol{\Sigma}^{(h-1)}(\boldsymbol{x}, \boldsymbol{x})\right]_{ij,ij} & \left[\boldsymbol{\Sigma}^{(h-1)}(\boldsymbol{x}, \boldsymbol{x}')\right]_{ij,i'j'} \\ \left[\boldsymbol{\Sigma}^{(h-1)}(\boldsymbol{x}', \boldsymbol{x})\right]_{i'j',ij} & \left[\boldsymbol{\Sigma}^{(h-1)}(\boldsymbol{x}', \boldsymbol{x}')\right]_{i'j',i'j'} \end{pmatrix} \in \mathbb{R}^{2 \times 2}.
$$

  - Define $\boldsymbol{K}^{(h)}(\boldsymbol{x}, \boldsymbol{x}'), \dot{\boldsymbol{K}}^{(h)}(\boldsymbol{x}, \boldsymbol{x}') \in \mathbb{R}^{P \times Q \times P \times Q}$: for $(i, j, i', j') \in [P] \times [Q] \times [P] \times [Q]$,

$$
\left[\boldsymbol{K}^{(h)}(\boldsymbol{x}, \boldsymbol{x}')\right]_{ij,i'j'} = \frac{c_\sigma}{q^2} \cdot \mathop{\mathbb{E}}_{(u,v) \sim \mathcal{N}\left(\mathbf{0}, \boldsymbol{\Lambda}^{(h)}_{ij,i'j'}(\boldsymbol{x}, \boldsymbol{x}')\right)} [\sigma(u) \sigma(v)], \tag{11}
$$

$$
\left[\dot{\boldsymbol{K}}^{(h)}(\boldsymbol{x}, \boldsymbol{x}')\right]_{ij,i'j'} = \frac{c_\sigma}{q^2} \cdot \mathop{\mathbb{E}}_{(u,v) \sim \mathcal{N}\left(\mathbf{0}, \boldsymbol{\Lambda}^{(h)}_{ij,i'j'}(\boldsymbol{x}, \boldsymbol{x}')\right)} [\dot{\sigma}(u) \dot{\sigma}(v)]. \tag{12}
$$

  - Define $\boldsymbol{\Sigma}^{(h)}(\boldsymbol{x}, \boldsymbol{x}') \in \mathbb{R}^{P \times Q \times P \times Q}$: for $(i, j, i', j') \in [P] \times [Q] \times [P] \times [Q]$,

$$
\left[\boldsymbol{\Sigma}^{(h)}(\boldsymbol{x}, \boldsymbol{x}')\right]_{ij,i'j'} = \text{tr}\left(\left[\boldsymbol{K}^{(h)}(\boldsymbol{x}, \boldsymbol{x}')\right]_{D_{ij,i'j'}}\right).
$$

Note that $\boldsymbol{\Sigma}(\boldsymbol{x}, \boldsymbol{x}')$ and $\dot{\boldsymbol{\Sigma}}(\boldsymbol{x}, \boldsymbol{x}')$ share similar structures as their NTK counterparts in Equations (7) and (8). The only difference is that we have one more step, taking the trace over patches. This step represents the convolution operation in the corresponding CNN. Next, we can use a recursion to compute the CNTK:

1. First, we define $\boldsymbol{\Theta}^{(0)}(\boldsymbol{x}, \boldsymbol{x}') = \boldsymbol{\Sigma}^{(0)}(\boldsymbol{x}, \boldsymbol{x}')$.
2. For $h = 1, \ldots, L-1$ and $(i, j, i', j') \in [P] \times [Q] \times [P] \times [Q]$, we define

$$
\left[\boldsymbol{\Theta}^{(h)}(\boldsymbol{x}, \boldsymbol{x}')\right]_{ij,i'j'} = \text{tr}\left(\left[\dot{\boldsymbol{K}}^{(h)}(\boldsymbol{x}, \boldsymbol{x}') \odot \boldsymbol{\Theta}^{(h-1)}(\boldsymbol{x}, \boldsymbol{x}') + \boldsymbol{K}^{(h)}(\boldsymbol{x}, \boldsymbol{x}')\right]_{D_{ij,i'j'}}\right).
$$

3. For $h = L$, we define $\boldsymbol{\Theta}^{(L)}(\boldsymbol{x}, \boldsymbol{x}') = \dot{\boldsymbol{K}}^{(L)}(\boldsymbol{x}, \boldsymbol{x}') \odot \boldsymbol{\Theta}^{(L-1)}(\boldsymbol{x}, \boldsymbol{x}') + \boldsymbol{K}^{(L)}(\boldsymbol{x}, \boldsymbol{x}')$.
4. The final CNTK value is defined as $\text{tr}\left(\boldsymbol{\Theta}^{(L)}(\boldsymbol{x}, \boldsymbol{x}')\right)$.

In Section H we give the CNTK formula for CNNs with GAP, which is similar to vanilla CNNs. To compute the CNTK matrix corresponding to a CNN with GAP that has $L$ convolution layers and one fully-connected layer on $n$ samples, the time complexity is $O(n^2 P^2 Q^2 L)$. Previous work assumed that directly computing convolutional kernel (with pooling) exactly is computationally infeasible, and thus resorted to approximations like Monte Carlo sampling [Novak et al., 2019]. We are able to scale the exact CNTK computation to the full CIFAR-10 dataset and 20-layer CNN with GAP. We present our efficient computation approach in Section I.

| Depth | CNN-V | CNTK-V | CNTK-V-2K | CNN-GAP | CNTK-GAP | CNTK-GAP-2K |
|---|---|---|---|---|---|---|
| 3 | 59.97% | 64.47% | 40.94% | 63.81% | 70.47% | 49.71% |
| 4 | 60.20% | 65.52% | 42.54% | 80.93% | 75.93% | 51.06% |
| 6 | 64.11% | 66.03% | 43.43% | 83.75% | 76.73% | 51.73% |
| 11 | 69.48% | 65.90% | 43.42% | 82.92% | **77.43%** | 51.92% |
| 21 | 75.57% | 64.09% | 42.53% | 83.30% | 77.08% | 52.22% |

Table 1: Classification accuracies of CNNs and CNTKs on the CIFAR-10 dataset. CNN-V represents vanilla CNN and CNTK-V represents the kernel corresponding to CNN-V. CNN-GAP represents CNN with GAP and CNTK-GAP represents the kernel correspondong to CNN-GAP. CNTK-V-2K and CNTK-GAP-2K represent training CNTKs with only 2,000 training data.

## 5    Experiments

We evaluate the performances of CNNs and their corresponding CNTKs on the CIFAR-10 dataset. The implementation details are in Section A. We also compare the performances between CNTKs and their corresponding random features. Due to space limit, we defer these results on random features to Section B.

**Results.**    We test two types of architectures, vanilla CNN and CNN with global average pooling (GAP), as described in Sections 4 and H. We also test CNTKs with only 2,000 training data to see whether their performances are consistent with CNTKs and CNNs using the full training set. The results are summarized in Table 1. Notice that in Table 1, depth is the total number of layers (including both convolution layers and fully-connected layers).

Several comments are in sequel. First, CNTKs are very powerful kernels. The best kernel, 11-layer CNTK with GAP, achieves 77.43% classification accuracy on CIFAR-10. This results in a significant new benchmark for performance of a pure kernel-based method on CIFAR-10, being $10\%$ higher than methods reported in [Novak et al., 2019].

Second, we find that for both CNN and CNTK, depth can affect the classification accuracy. This observation demonstrates that depth not only matters in deep neural networks but can also affect the performance of CNTKs.

Third, the global average pooling operation can significantly increase the classification accuracy by 8% - 10% for both CNN and CNTK. Based on this finding, we expect that many techniques that improve the performance of neural networks are in some sense universal, i.e., these techniques can also benefit kernel methods.

Fourth, we find that there is still a 5% - 6% performance gap between CNTKs and CNNs. Since CNTKs exactly correspond to infinitely wide CNNs, this performance gap implies that finite width has its benefits. Therefore, it is likely that recent theoretical work on over-parameterization that operates in the NTK regime cannot fully explain the success of neural networks yet, and we believe it is an interesting open problem to characterize this gap.

**Potential application in neural architecture search.**    Finally, we find that performances of CNTK-V-2Ks and CNTK-GAP-2Ks are highly correlated to their CNN-V, CNTK-V, CNN-GAP and CNTK-GAP counterparts. Again we see CNTK-GAP-2Ks outperform CNTK-V-2Ks by a large margin (about $8\%$ - $9\%$). One potential application of this observation is to guide neural architecture search. We can compute the kernel on a small training data, test it on a validation set, and choose neural network architectures based on the performance of this small kernel on the validation set. We leave large scale experiments of this idea for future work.

## 6    Conclusion

By giving the first practical algorithm for computing CNTKs exactly, this paper allows investigation of the behavior of infinitely wide (hence infinitely over-parametrized) deep nets, which turns out to

not be much worse than that of their finite counterparts. We also give a fully rigorous proof that a sufficiently wide net is approximately equivalent to the kernel regression predictor, thus yielding a powerful new off-the-shelf kernel. We leave it as an open problem to understand the behavior of infinitely wide nets with features such as Batch Normalization or Residual Layers. Of course, one can also hope that the analysis of infinite nets provides rigorous insight into finite ones.

## Acknowledgments

We thank Jason D. Lee, Haochuan Li and Xiyu Zhai for useful discussions. S. Arora, W. Hu and Z. Li are supported by NSF, ONR, Simons Foundation, Schmidt Foundation, Mozilla Research, Amazon Research, DARPA and SRC. R. Salakhutdinov and R. Wang are supported in part by NSF IIS-1763562, Office of Naval Research grant N000141812861, and Nvidia NVAIL award. We thank Amazon Web Services for providing compute time for the experiments in this paper.

## Footnotes

[8]We only consider fixed kernels defined without using the training data. We do not compare to methods that tune the kernels using training data [Van der Wilk et al., 2017] or use a neural network to extract features and then applying a kernel method on top of them [Mairal et al., 2014].

[9]For simplicity, we only consider a single output here. The generalization to multiple outputs is straightforward.

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
