[Supplementary Material]

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

# A    Experiment Details

**Setup.**    Due to efficiency considerations, for all experiments, we use no data augmentation. Tricks like batch normalization, dropout, weight decay, etc. are not used for proper comparison. We fix the filter $q$ to be 3 and stride to be 1. We use zero padding to make sure the number of patches keeps unchanged after each convolutional layer. We set the number of convolution layers to be 2, 3, 5, 10, or 20. For both CNNs and CNTKs, we use the quadratic loss as the objective function.

Following Novak et al. [2019], for a label $c \in \{1, \ldots, 10\}$, we use $-0.1 \cdot \mathbf{1} + \boldsymbol{e}_c$ as its encoding. For example, if the class label is 3, we use $(-0.1, -0.1, 0.9, -0.1, \ldots, -0.1)$ as its encoding. During training time, we calculate $(\boldsymbol{H}^*)^{-1} \boldsymbol{Y}$, where $\boldsymbol{H}^*$ is the CNTK matrix on inputs, and the $i$-th row of $\boldsymbol{Y} \in \mathbb{R}^{n \times 10}$ is the encoding of the label of the $i$-th data. During testing time, for a test data point $\boldsymbol{x}_{te}$, we calculate

$$\boldsymbol{f}^*(\boldsymbol{x}_{te}) = (\ker(\boldsymbol{x}_{te}, \boldsymbol{x}_1), \ldots, \ker(\boldsymbol{x}_{te}, \boldsymbol{x}_n)) \cdot (\boldsymbol{H}^*)^{-1} \boldsymbol{Y}$$

and choose the class with largest value as the prediction.

The architecture of CNNs is as described in Section 4 and Section H. We set the number of the channels of the network as 1024 and $\kappa$ as 0.05. To train CNNs, we use stochastic gradient descent (SGD) with fixed learning rate. We report the best average performance over 3 trials among the different learning rate chosen from $\{0.1, 1, 10\}$. The test accuracy is measured by taking average of the 10 epochs after reaching full training accuracy except the depth-3 vanilla CNN, which couldn't attain full training accuracy within 3000 epochs for all learning rates

Our neural networks are trained using the PyTorch package, using (possibly multiple) NVIDIA Tesla V100 GPUs. We calculate the kernel values using the CuPy[10] package. For time-consuming operations, we write native CUDA codes to speed up the calculation. All experiments are performed on Amazon Web Services (AWS).

# B    Additional Experiments on Random Features

We verify the importance of using the exact kernels instead of the approximated ones from random features (as done in [Lee et al., 2019]). The random features are generated by taking the gradient of the randomly initialized CNNs with respect to the weight matrices. For all CNNs we set the number of channels to be 128. We compare the performances of the exact kernels and the random kernels on a CIFAR-2 dataset, i.e., the first two class in CIFAR-10. For each kernel generated by random features, we test 10 times and report the median. The results are summarized in Table 2.

| Depth | RF from Vanilla CNTK | Vanilla CNTK | RF for CNTK-GAP | CNTK-GAP |
|-------|----------------------|--------------|------------------|----------|
| 3     | 87.25%               | 92.15%       | 51.10%           | 71.05%   |
| 4     | 87.78%               | 92.80%       | 52.85%           | 94.50%   |
| 6     | 88.73%               | 93.10%       | 53.98%           | 95.25%   |
| 11    | 87.80%               | 93.05%       | 56.55%           | 95.40%   |
| 21    | 85.35%               | 91.95%       | 90.65%           | 95.70%   |

Table 2: Classification accuracies of random kernels generated from random features and exact CNTKs on CIFAR-2.

Note that even on the simple CIFAR-2 dataset, random features have much worse accuracies than exact kernels by a large margin. This experiment demonstrates the importance of using the exact kernels instead of approximated ones.

# C  Proof of Lemma 3.1

*Proof of Lemma 3.1.* The parameters $\boldsymbol{\theta}$ evolve according to the differential equation

$$\frac{\mathrm{d}\boldsymbol{\theta}(t)}{\mathrm{d}t} = -\nabla\ell(\boldsymbol{\theta}(t)) = -\sum_{i=1}^{n}\left(f(\boldsymbol{\theta}(t), \boldsymbol{x}_i) - y_i\right)\frac{\partial f(\boldsymbol{\theta}(t), \boldsymbol{x}_i)}{\partial\boldsymbol{\theta}}, \tag{13}$$

where $t \geq 0$ is a continuous time index. Under Equation (13), the evolution of the network output $f(\boldsymbol{\theta}(t), \boldsymbol{x}_i)$ can be written as

$$\frac{\mathrm{d}f(\boldsymbol{\theta}(t), \boldsymbol{x}_i)}{\mathrm{d}t} = -\sum_{j=1}^{n}(f(\boldsymbol{\theta}(t), \boldsymbol{x}_j) - y_j)\left\langle\frac{\partial f(\boldsymbol{\theta}(t), \boldsymbol{x}_i)}{\partial\boldsymbol{\theta}}, \frac{\partial f(\boldsymbol{\theta}(t), \boldsymbol{x}_j)}{\partial\boldsymbol{\theta}}\right\rangle, \qquad \forall i \in [n]. \tag{14}$$

Since $\boldsymbol{u}(t) = (f(\boldsymbol{\theta}(t), \boldsymbol{x}_i))_{i\in[n]} \in \mathbb{R}^n$ is the network outputs on all $\boldsymbol{x}_i$'s at time $t$, and $\boldsymbol{y} = (y_i)_{i\in[n]}$ is the desired outputs, Equation (14) can be written more compactly as

$$\frac{\mathrm{d}\boldsymbol{u}(t)}{\mathrm{d}t} = -\boldsymbol{H}(t)\cdot(\boldsymbol{u}(t) - \boldsymbol{y}), \tag{15}$$

where $\boldsymbol{H}(t) \in \mathbb{R}^{n\times n}$ is a kernel matrix defined as $[\boldsymbol{H}(t)]_{i,j} = \left\langle\frac{\partial f(\boldsymbol{\theta}(t), \boldsymbol{x}_i)}{\partial\boldsymbol{\theta}}, \frac{\partial f(\boldsymbol{\theta}(t), \boldsymbol{x}_j)}{\partial\boldsymbol{\theta}}\right\rangle (\forall i, j \in [n])$. $\square$

# D  NTK Derivation

In this section we derive NTK for the fully-connected neural net defined in Section 3.

First we explain how the Gaussian process covariance in Equation (7) is obtained. The intuition is that $\left[\boldsymbol{f}^{(h+1)}(\boldsymbol{x})\right]_i = \sum_{j=1}^{d_h}\left[\boldsymbol{W}^{(h+1)}\right]_{i,j}\left[\boldsymbol{g}^{(h)}(\boldsymbol{x})\right]_j$ is a centered Gaussian process conditioned on $\boldsymbol{f}^{(h)}$ ($\forall i \in [d_{h+1}]$), with covariance

$$\begin{aligned}
\mathbb{E}\left[\left[\boldsymbol{f}^{(h+1)}(\boldsymbol{x})\right]_i \cdot \left[\boldsymbol{f}^{(h+1)}(\boldsymbol{x}')\right]_i \middle| \boldsymbol{f}^{(h)}\right] &= \left\langle\boldsymbol{g}^{(h)}(\boldsymbol{x}), \boldsymbol{g}^{(h)}(\boldsymbol{x}')\right\rangle \\
&= \frac{c_\sigma}{d_h}\sum_{j=1}^{d_h}\sigma\left(\left[\boldsymbol{f}^{(h)}(\boldsymbol{x})\right]_j\right)\sigma\left(\left[\boldsymbol{f}^{(h)}(\boldsymbol{x}')\right]_j\right),
\end{aligned} \tag{16}$$

which converges to $\Sigma^{(h)}(\boldsymbol{x}, \boldsymbol{x}')$ as $d_h \to \infty$ given that each $\left[\boldsymbol{f}^{(h)}\right]_j$ is a centered Gaussian process with covariance $\Sigma^{(h-1)}$. This yields the inductive definition in Equation (7).

Recall that we need to compute the value that $\left\langle\frac{\partial f(\boldsymbol{\theta}, \boldsymbol{x})}{\partial\boldsymbol{\theta}}, \frac{\partial f(\boldsymbol{\theta}, \boldsymbol{x}')}{\partial\boldsymbol{\theta}}\right\rangle$ converges to at random initialization in the infinite width limit. We can write the partial derivative with respect to a particular weight matrix $\boldsymbol{W}^{(h)}$ in a compact form:

$$\frac{\partial f(\boldsymbol{\theta}, \boldsymbol{x})}{\partial\boldsymbol{W}^{(h)}} = \mathbf{b}^{(h)}(\boldsymbol{x})\cdot\left(\boldsymbol{g}^{(h-1)}(\boldsymbol{x})\right)^\top, \qquad h = 1, 2, \ldots, L+1,$$

where

$$\mathbf{b}^{(h)}(\boldsymbol{x}) = \begin{cases} 1 \in \mathbb{R}, & h = L+1, \\ \sqrt{\frac{c_\sigma}{d_h}}\boldsymbol{D}^{(h)}(\boldsymbol{x})\left(\boldsymbol{W}^{(h+1)}\right)^\top\mathbf{b}^{(h+1)}(\boldsymbol{x}) \in \mathbb{R}^{d_h}, & h = 1, \ldots, L, \end{cases} \tag{17}$$

$$\boldsymbol{D}^{(h)}(\boldsymbol{x}) = \operatorname{diag}\left(\dot{\sigma}\left(\boldsymbol{f}^{(h)}(\boldsymbol{x})\right)\right) \in \mathbb{R}^{d_h\times d_h}, \qquad h = 1, \ldots, L. \tag{18}$$

Then, for any $h \in [L+1]$, we can compute

$$\begin{aligned}
\left\langle\frac{\partial f(\boldsymbol{\theta}, \boldsymbol{x})}{\partial\boldsymbol{W}^{(h)}}, \frac{\partial f(\boldsymbol{\theta}, \boldsymbol{x}')}{\partial\boldsymbol{W}^{(h)}}\right\rangle &= \left\langle\mathbf{b}^{(h)}(\boldsymbol{x})\cdot\left(\boldsymbol{g}^{(h-1)}(\boldsymbol{x})\right)^\top, \mathbf{b}^{(h)}(\boldsymbol{x}')\cdot\left(\boldsymbol{g}^{(h-1)}(\boldsymbol{x}')\right)^\top\right\rangle \\
&= \left\langle\boldsymbol{g}^{(h-1)}(\boldsymbol{x}), \boldsymbol{g}^{(h-1)}(\boldsymbol{x}')\right\rangle\cdot\left\langle\mathbf{b}^{(h)}(\boldsymbol{x}), \mathbf{b}^{(h)}(\boldsymbol{x}')\right\rangle.
\end{aligned}$$

Note that we have established in Equation (16) that

$$\left\langle \boldsymbol{g}^{(h-1)}(\boldsymbol{x}), \boldsymbol{g}^{(h-1)}(\boldsymbol{x}') \right\rangle \to \Sigma^{(h-1)}\left(\boldsymbol{x}, \boldsymbol{x}'\right).$$

For the other factor $\left\langle \mathbf{b}^{(h)}(\boldsymbol{x}), \mathbf{b}^{(h)}(\boldsymbol{x}') \right\rangle$, from Equation (17) we get

$$
\begin{aligned}
&\left\langle \mathbf{b}^{(h)}(\boldsymbol{x}), \mathbf{b}^{(h)}(\boldsymbol{x}') \right\rangle \\
&= \left\langle \sqrt{\frac{c_\sigma}{d_h}} \boldsymbol{D}^{(h)}(\boldsymbol{x}) \left(\boldsymbol{W}^{(h+1)}\right)^\top \mathbf{b}^{(h+1)}(\boldsymbol{x}), \sqrt{\frac{c_\sigma}{d_h}} \boldsymbol{D}^{(h)}(\boldsymbol{x}') \left(\boldsymbol{W}^{(h+1)}\right)^\top \mathbf{b}^{(h+1)}(\boldsymbol{x}') \right\rangle.
\end{aligned}
\tag{19}
$$

Although $\boldsymbol{W}^{(h+1)}$ and $\mathbf{b}_{h+1}(\boldsymbol{x})$ are dependent, the Gaussian initialization of $\boldsymbol{W}^{(h+1)}$ allows us to replace $\boldsymbol{W}^{(h+1)}$ with a fresh new sample $\widetilde{\boldsymbol{W}}^{(h+1)}$ without changing its limit: (This is made rigorous for ReLU activation in Theorem 3.1.)

$$
\begin{aligned}
&\left\langle \sqrt{\frac{c_\sigma}{d_h}} \boldsymbol{D}^{(h)}(\boldsymbol{x}) \left(\boldsymbol{W}^{(h+1)}\right)^\top \mathbf{b}^{(h+1)}(\boldsymbol{x}), \sqrt{\frac{c_\sigma}{d_h}} \boldsymbol{D}^{(h)}(\boldsymbol{x}') \left(\boldsymbol{W}^{(h+1)}\right)^\top \mathbf{b}^{(h+1)}(\boldsymbol{x}') \right\rangle \\
&\approx \left\langle \sqrt{\frac{c_\sigma}{d_h}} \boldsymbol{D}^{(h)}(\boldsymbol{x}) \left(\widetilde{\boldsymbol{W}}^{(h+1)}\right)^\top \mathbf{b}^{(h+1)}(\boldsymbol{x}), \sqrt{\frac{c_\sigma}{d_h}} \boldsymbol{D}^{(h)}(\boldsymbol{x}') \left(\widetilde{\boldsymbol{W}}^{(h+1)}\right)^\top \mathbf{b}^{(h+1)}(\boldsymbol{x}') \right\rangle \\
&\to \frac{c_\sigma}{d_h} \mathrm{tr}\left( \boldsymbol{D}^{(h)}(\boldsymbol{x}) \boldsymbol{D}^{(h)}(\boldsymbol{x}') \right) \left\langle \mathbf{b}^{(h+1)}(\boldsymbol{x}), \mathbf{b}^{(h+1)}(\boldsymbol{x}') \right\rangle \\
&\to \dot{\Sigma}^{(h)}\left(\boldsymbol{x}, \boldsymbol{x}'\right) \left\langle \mathbf{b}^{(h+1)}(\boldsymbol{x}), \mathbf{b}^{(h+1)}(\boldsymbol{x}') \right\rangle.
\end{aligned}
$$

Applying this approximation inductively in Equation (19), we get

$$\left\langle \mathbf{b}^{(h)}(\boldsymbol{x}), \mathbf{b}^{(h)}(\boldsymbol{x}') \right\rangle \to \prod_{h'=h}^{L} \dot{\Sigma}^{(h')}(\boldsymbol{x}, \boldsymbol{x}').$$

Finally, since $\left\langle \frac{\partial f(\boldsymbol{\theta}, \boldsymbol{x})}{\partial \boldsymbol{\theta}}, \frac{\partial f(\boldsymbol{\theta}, \boldsymbol{x}')}{\partial \boldsymbol{\theta}} \right\rangle = \sum_{h=1}^{L+1} \left\langle \frac{\partial f(\boldsymbol{\theta}, \boldsymbol{x})}{\partial \boldsymbol{W}^{(h)}}, \frac{\partial f(\boldsymbol{\theta}, \boldsymbol{x}')}{\partial \boldsymbol{W}^{(h)}} \right\rangle$, we obtain the final NTK expression for the fully-connected neural network:

$$\Theta^{(L)}(\boldsymbol{x}, \boldsymbol{x}') = \sum_{h=1}^{L+1} \left( \Sigma^{(h-1)}(\boldsymbol{x}, \boldsymbol{x}') \cdot \prod_{h'=h}^{L+1} \dot{\Sigma}^{(h')}(\boldsymbol{x}, \boldsymbol{x}') \right).$$

## E   Proof of Theorem 3.1

### E.1   Notation and Some Properties of ReLU

**Definition E.1** ($k$-homogeneous function). *A function $f : \mathbb{R} \to \mathbb{R}$ is said to be $k$-homogeneous, if $f(\lambda x) = \lambda^k f(x)$ for all $x \in \mathbb{R}, \lambda > 0$.*

**Definition E.2.** *Let $\mathcal{S}^+$ be the set of* positive semi-definite kernels *over $\mathbb{R}^d$, that is*

$$\mathcal{S}^+ = \left\{ K : \mathbb{R}^d \times \mathbb{R}^d \to \mathbb{R} \middle| \forall N \in \mathbb{N}, \boldsymbol{x}_1, \ldots \boldsymbol{x}_N \in \mathbb{R}^d, c_1, \ldots, c_N \in \mathbb{R}, \sum_{i=1}^{N} \sum_{j=1}^{N} c_i c_j K(x_i, x_j) \geq 0. \right\}$$

Let $\sigma : \mathbb{R} \to \mathbb{R}$ be the activation function, and $\mathcal{T}_\sigma : \mathcal{S}^+ \to \mathcal{S}^+$ be the operator induced by $\sigma$,

$$\forall \boldsymbol{x}, \boldsymbol{x}' \in \mathbb{R}^d, \quad \mathcal{T}_\sigma(K)(\boldsymbol{x}, \boldsymbol{x}') = c_\sigma \mathop{\mathbb{E}}_{(u,v)\sim\mathcal{N}\left(\boldsymbol{0}, \boldsymbol{K}|_{\boldsymbol{x},\boldsymbol{x}'}\right)} \left[ \sigma(u) \sigma(v) \right],$$

where $\boldsymbol{K}|_{\boldsymbol{x},\boldsymbol{x}'} \in \mathbb{R}^{2 \times 2}$, $\boldsymbol{K}|_{\boldsymbol{x},\boldsymbol{x}'} = \begin{bmatrix} K(\boldsymbol{x}, \boldsymbol{x}) & K(\boldsymbol{x}, \boldsymbol{x}') \\ K(\boldsymbol{x}', \boldsymbol{x}) & K(\boldsymbol{x}', \boldsymbol{x}') \end{bmatrix}$.

For convenience, we use $t_\sigma(\mathbf{\Sigma})$ to denote $c_\sigma \mathbb{E}_{(u,v)\sim\mathcal{N}(\mathbf{0},\mathbf{\Sigma})}\left[\sigma(u)\sigma(v)\right]$, and define $\hat{t}_\sigma(\rho)$ as

$$\hat{t}_\sigma(\rho) = c_\sigma \mathop{\mathbb{E}}_{(u,v)\sim\mathbf{\Sigma}'}\left[\sigma(u)\sigma(v)\right], \text{ with } \mathbf{\Sigma}' = \begin{bmatrix} 1 & \rho \\ \rho & 1 \end{bmatrix}$$

When $\sigma$ is k-homogeneous function, we have

$$t_\sigma(\mathbf{\Sigma}) = c_\sigma \left(\Sigma_{11}\Sigma_{22}\right)^{\frac{k}{2}} \mathop{\mathbb{E}}_{(u,v)\sim\mathcal{N}(\mathbf{0},\mathbf{\Sigma}')}\left[\sigma(u)\sigma(v)\right] \quad \text{with} \quad \mathbf{\Sigma}' = \begin{bmatrix} 1 & \frac{\Sigma_{12}}{\sqrt{\Sigma_{11}\Sigma_{22}}} \\ \frac{\Sigma_{12}}{\sqrt{\Sigma_{11}\Sigma_{22}}} & 1 \end{bmatrix}.$$

Thus $t_\sigma(\mathbf{\Sigma})$ can be written as $c_\sigma \left(\Sigma_{11}\Sigma_{22}\right)^{\frac{k}{2}} \hat{t}(\frac{\Sigma_{12}}{\sqrt{\Sigma_{11}\Sigma_{22}}})$,

**Fact E.1** (Some facts about $\sigma(z) = \max(0, z)$ and $\mathcal{T}_\sigma$).

1. *For all activation function $\sigma$, $t_\sigma\left(\begin{bmatrix} 1 & 1 \\ 1 & 1 \end{bmatrix}\right) = 1$.*

2. *For all 1-homogeneous activation $\sigma$, $\hat{t}_\sigma(1) = 1$ and $t_\sigma\left(\begin{bmatrix} a & a \\ a & a \end{bmatrix}\right) = a^k$.*

3. *For $\sigma(z) = \max(0, z)$, $\hat{t}_\sigma(\rho) = \frac{\sqrt{1-\rho^2}+\rho\arcsin\rho}{\pi} + \frac{x}{2}$, $\hat{t}_{\dot\sigma}(\rho) = \frac{1}{2} + \frac{\arcsin\rho}{\pi}$ and $c_\sigma = c_{\dot\sigma} = 2$.*

**Lemma E.1** (Uniform Continuity of $\arcsin z$).

1. *For any $-\frac{\pi}{2} \le y' \le y \le \frac{\pi}{2}$, $\sin y - \sin y' \ge 2\sin^2\frac{y-y'}{2}$.*

2. $\sin y \ge \frac{2y}{\pi}$, $\forall y \in [0, \frac{\pi}{2}]$.

3. $\arcsin$ *is uniform continuous: for every $\epsilon \in \mathbb{R}^+$, $|z-z'| < \frac{2\epsilon^2}{\pi^2} \Rightarrow |\arcsin z - \arcsin z'| < \epsilon$.*

4. *For $\sigma(z) = \max(0, z)$, $\hat{t}_{\dot\sigma}$ is uniform continuous: for every $\epsilon \in \mathbb{R}^+$, $|z - z'| < 2\epsilon^2 \Rightarrow |\hat{t}_{\dot\sigma}(z) - \hat{t}_{\dot\sigma}(z')| < \epsilon$.*

*Proof of Lemma E.1.* (1). From $-\frac{\pi}{2} \le y' \le y \le \frac{\pi}{2}$ we know $\frac{-\pi}{2} + \frac{y-y'}{2} \le \frac{y+y'}{2} \le \frac{\pi}{2} - \frac{y-y'}{2}$, which implies that $\cos(\frac{y+y'}{2}) \ge \sin(\frac{y-y'}{2})$. Thus,

$$\sin y \sin y' = 2\cos\frac{y+y'}{2}\sin\frac{y-y'}{2} \ge 2\sin^2\frac{y-y'}{2}.$$

(2). Note that $\left(\frac{\sin y}{y}\right)' = \frac{y\cos y - \sin y}{y^2} = \frac{\cos y}{y^2}(y - \tan y) < 0$, $\frac{\sin y}{y}$ is decreasing on $[0, \frac{\pi}{2}]$. Thus $\frac{\sin y}{y} \ge \frac{1}{\frac{\pi}{2}} = \frac{2}{\pi}$, $\forall y \in [0, \frac{\pi}{2}]$.

(3). Let $y, y' \in [-\frac{\pi}{2}, \frac{\pi}{2}]$, such that $\sin y = z, \sin y' = z'$. W.l.o.g., we assume $y' < y, z' < z$. Combing (1) and (2), we have $z - z' = \sin y - \sin y' \ge 2\sin^2\frac{y-y'}{2} \ge \frac{2(y-y')^2}{\pi^2}$. Thus $z - z' \le \frac{2\epsilon^2}{\pi^2} \implies \arcsin z - \arcsin z' = y - y' \le \epsilon$. $\qquad\square$

Recall the definition in Equation (7) and (8), we have

$$\Sigma^{(0)}(\boldsymbol{x}, \boldsymbol{x}') = \boldsymbol{x}^\top \boldsymbol{x}',$$

$$\boldsymbol{\Lambda}^{(h)}(\boldsymbol{x}, \boldsymbol{x}') = \mathbf{\Sigma}^{(h-1)}\Big|_{\boldsymbol{x},\boldsymbol{x}'} = \begin{pmatrix} \Sigma^{(h-1)}(\boldsymbol{x}, \boldsymbol{x}) & \Sigma^{(h-1)}(\boldsymbol{x}, \boldsymbol{x}') \\ \Sigma^{(h-1)}(\boldsymbol{x}', \boldsymbol{x}) & \Sigma^{(h-1)}(\boldsymbol{x}', \boldsymbol{x}') \end{pmatrix} \in \mathbb{R}^{2\times 2},$$

$$\Sigma^{(h)}(\boldsymbol{x}, \boldsymbol{x}') = c_\sigma \mathop{\mathbb{E}}_{(u,v)\sim\mathcal{N}(\mathbf{0},\boldsymbol{\Lambda}^{(h)})}\left[\sigma(u)\sigma(v)\right],$$

$$\dot\Sigma^{(h)}(\boldsymbol{x}, \boldsymbol{x}') = c_\sigma \mathop{\mathbb{E}}_{(u,v)\sim\mathcal{N}(\mathbf{0},\boldsymbol{\Lambda}^{(h)})}\left[\dot\sigma(u)\dot\sigma(v)\right]$$

for $h = 1, \ldots, L$.

For $\sigma(z) = \max(z, 0)$, we have

$$\Sigma^{(h)}(\boldsymbol{x}, \boldsymbol{x}) = \|\boldsymbol{x}\|^2, \quad \forall 0 \le h \le L.$$

Let $\boldsymbol{D} = \boldsymbol{D}(\boldsymbol{x}, \boldsymbol{x}') = \boldsymbol{D}^{(h)}(\boldsymbol{x})\boldsymbol{D}^{(h)}(\boldsymbol{x}')$ is a 0-1 diagonal matrix. We define the following events:

- $\mathcal{A}^h(\boldsymbol{x}, \boldsymbol{x}', \epsilon_1) := \left\{\left|\boldsymbol{g}^{(h)}(x^{(0)})^\top \boldsymbol{g}^{(h)}(\boldsymbol{x}) - \boldsymbol{\Sigma}^{(h)}(\boldsymbol{x}^{(0)}, \boldsymbol{x})\right| \le \epsilon_1\right\}, \forall 0 \le h \le L$
- $\overline{\mathcal{A}}^h(\boldsymbol{x}, \boldsymbol{x}', \epsilon_1) = \mathcal{A}^h(\boldsymbol{x}, \boldsymbol{x}, \epsilon_1) \cap \mathcal{A}^h(\boldsymbol{x}, \boldsymbol{x}', \epsilon_1) \cap \mathcal{A}^h(\boldsymbol{x}', \boldsymbol{x}', \epsilon_1)$;
- $\overline{\mathcal{A}}(\boldsymbol{x}, \boldsymbol{x}', \epsilon_1) = \bigcup_{h=0}^{L} \overline{\mathcal{A}}^h(\epsilon_1)$.
- $\mathcal{B}^h(\boldsymbol{x}, \boldsymbol{x}', \epsilon_2) = \left\{\left|\langle \mathbf{b}^{(h)}(\boldsymbol{x}), \mathbf{b}^{(h)}(\boldsymbol{x}')\rangle - \prod_{h=h}^{L} \dot{\Sigma}^{(h)}(\boldsymbol{x}, \boldsymbol{x}')\right| < \epsilon_2\right\}$;
- $\overline{\mathcal{B}}^h(\boldsymbol{x}, \boldsymbol{x}', \epsilon_2) = \mathcal{B}^h(\boldsymbol{x}, \boldsymbol{x}, \epsilon_2) \cap \mathcal{B}^h(\boldsymbol{x}, \boldsymbol{x}', \epsilon_2) \cap \mathcal{B}^h(\boldsymbol{x}', \boldsymbol{x}', \epsilon_2)$;
- $\overline{\mathcal{B}}(\boldsymbol{x}, \boldsymbol{x}', \epsilon_2) = \bigcup_{h=1}^{L+1} \overline{\mathcal{B}}^h(\epsilon_2)$;
- $\overline{\mathcal{C}}(\boldsymbol{x}, \boldsymbol{x}', \epsilon_3) = \{|f(\boldsymbol{\theta}, \boldsymbol{x})| \le \epsilon_3, |f(\boldsymbol{\theta}, \boldsymbol{x}')| \le \epsilon_3\}$;
- $\mathcal{D}^h(\boldsymbol{x}, \boldsymbol{x}', \epsilon_4) = \left\{\left|2\frac{\mathrm{tr}(\boldsymbol{D}(\boldsymbol{x}, \boldsymbol{x}'))}{d_h} - \dot{\Sigma}^{(h)}(\boldsymbol{x}, \boldsymbol{x}')\right| < \epsilon_4\right\}$;
- $\overline{\mathcal{D}}^h(\boldsymbol{x}, \boldsymbol{x}', \epsilon_4) = \mathcal{D}^h(\boldsymbol{x}, \boldsymbol{x}, \epsilon_4) \cap \mathcal{D}^h(\boldsymbol{x}, \boldsymbol{x}', \epsilon_4) \cap \mathcal{D}^h(\boldsymbol{x}', \boldsymbol{x}', \epsilon_1)$;
- $\overline{\mathcal{D}}(\boldsymbol{x}, \boldsymbol{x}', \epsilon_4) = \bigcup_{h=1}^{L+1} \overline{\mathcal{D}}^h(\epsilon_4)$.

For simplicity, we will omit $\boldsymbol{x}, \boldsymbol{x}'$ when there's no ambiguity. For events $\mathcal{A}, \mathcal{B}$, we define the event $\mathcal{A} \Rightarrow \mathcal{B}$ as $\neg \mathcal{A} \wedge \mathcal{B}$.

**Lemma E.2.** $\mathbb{P}[\mathcal{A} \Rightarrow \mathcal{B}] \ge \mathbb{P}[\mathcal{B} \mid \mathcal{A}]$.

*Proof.* $\mathbb{P}[\mathcal{A} \Rightarrow \mathcal{B}] = \mathbb{P}[\neg \mathcal{A} \wedge \mathcal{B}] = 1 - \mathbb{P}[\mathcal{A} \vee \neg \mathcal{B}] = 1 - \mathbb{P}[\neg \mathcal{B} \mid \mathcal{A}]\mathbb{P}[\mathcal{A}] \ge 1 - \mathbb{P}[\neg \mathcal{B} \mid \mathcal{A}] = \mathbb{P}[\mathcal{B} \mid \mathcal{A}]$. $\square$

For matrix $\boldsymbol{A}$, define the projection matrix for the column space of $\boldsymbol{A}$, $\boldsymbol{\Pi}_{\boldsymbol{A}} := \boldsymbol{A}\boldsymbol{A}^\dagger$ and the orthogonal projection matrix $\boldsymbol{\Pi}_{\boldsymbol{A}}^\perp = I - \boldsymbol{A}\boldsymbol{A}^\dagger$. For two random variables $X$ and $Y$, $X \stackrel{\mathrm{d}}{=}_{\mathcal{A}} Y$ means $X$ is equal to $Y$ in distribution conditioned on the $\sigma$-algebra generated by $\mathcal{A}$.

**Lemma E.3.** *Let $\boldsymbol{w} \sim \mathcal{N}(0, \boldsymbol{I}_d)$, $\boldsymbol{G} \in \mathbb{R}^{d \times k}$ be some fixed matrix, and random vector $\boldsymbol{F} = \boldsymbol{w}^\top \boldsymbol{G}$, then conditioned on the value of $\boldsymbol{F}$, $\boldsymbol{w}$ remains gaussian in the null space of the row space of $\boldsymbol{G}$. Mathematically, it means*

$$\boldsymbol{\Pi}_{\boldsymbol{G}}^\perp \boldsymbol{w} \stackrel{d}{=}_{\boldsymbol{F} = \boldsymbol{w}^\top \boldsymbol{G}} \boldsymbol{\Pi}_{\boldsymbol{G}}^\perp \widetilde{\boldsymbol{w}},$$

*where $\widetilde{\boldsymbol{w}} \sim \mathcal{N}(0, \boldsymbol{I}_d)$ is a fresh i.i.d. copy of $\boldsymbol{w}$.*

*Proof.* This lemma is straightforward when $\boldsymbol{\Pi}_{\boldsymbol{G}}^\perp$ is a diagonal matrix.

In general, let $\boldsymbol{G} = \boldsymbol{U}\boldsymbol{G}'$, where $\boldsymbol{U} \in \mathbb{R}^{d \times d}$ is orthogonal and $\boldsymbol{\Pi}_{\boldsymbol{G}'}^\perp$ is diagonal. Now we have

$$\boldsymbol{\Pi}_{\boldsymbol{G}}^\perp \boldsymbol{w} = \boldsymbol{U}\boldsymbol{\Pi}_{\boldsymbol{G}'}^\perp \boldsymbol{U}^\top \boldsymbol{w} \stackrel{\mathrm{d}}{=}_{\boldsymbol{F} = (\boldsymbol{U}^\top \boldsymbol{w})^\top \boldsymbol{G}'} \boldsymbol{U}\boldsymbol{\Pi}_{\boldsymbol{G}'}^\perp \boldsymbol{U}^\top \widetilde{\boldsymbol{w}}, = \boldsymbol{\Pi}_{\boldsymbol{G}}^\perp \widetilde{\boldsymbol{w}}$$

where we used the fact that if $\boldsymbol{w} \sim \mathcal{N}(0, \boldsymbol{I}_d)$, then for any orthogonal $\boldsymbol{U}$, $\boldsymbol{U}\boldsymbol{w} \sim \mathcal{N}(0, \boldsymbol{I}_d)$ twice. $\square$

## E.2 Proof Sketch

**Theorem 3.1** (Convergence to the NTK at initializatoin)**.** *Fix $\epsilon > 0$ and $\delta \in (0, 1)$. Suppose $\sigma(z) = \max(0, z)$ and $\min_{h \in [L]} d_h \ge \Omega(\frac{L^{14}}{\epsilon^4} \log(L/\delta))$. Then for any inputs $\boldsymbol{x}, \boldsymbol{x}' \in \mathbb{R}^{d_0}$ such that $\|\boldsymbol{x}\| \le 1, \|\boldsymbol{x}'\| \le 1$, with probability at least $1 - \delta$ we have:*

$$\left|\left\langle \frac{\partial f(\boldsymbol{\theta}, \boldsymbol{x})}{\partial \boldsymbol{\theta}}, \frac{\partial f(\boldsymbol{\theta}, \boldsymbol{x}')}{\partial \boldsymbol{\theta}}\right\rangle - \Theta^{(L)}(\boldsymbol{x}, \boldsymbol{x}')\right| \le \epsilon.$$

*Proof.* Recall that $\Theta^{(L)}(\boldsymbol{x}, \boldsymbol{x}') = \sum_{h=1}^{L+1} \left( \Sigma^{(h-1)}(\boldsymbol{x}, \boldsymbol{x}') \cdot \prod_{h'=h}^{L+1} \dot{\Sigma}^{(h')}(\boldsymbol{x}, \boldsymbol{x}') \right)$, thus it suffices to show that w.p. $1 - \delta$, for every $0 \leq h \leq L$, it holds that

$$\left| \left\langle \frac{\partial f(\boldsymbol{\theta}, \boldsymbol{x})}{\partial \boldsymbol{W}^{(h)}}, \frac{\partial f(\boldsymbol{\theta}, \boldsymbol{x}')}{\partial \boldsymbol{W}^{(h)}} \right\rangle - \Sigma^{(h-1)}(\boldsymbol{x}, \boldsymbol{x}') \cdot \prod_{h'=h}^{L+1} \dot{\Sigma}^{(h')}(\boldsymbol{x}, \boldsymbol{x}') \right| \leq \frac{\epsilon}{L+1},$$

which is a direct consequence of Theorem E.2 $\qquad \square$

**Theorem E.1** (Corollary 16 in [Daniely et al., 2016]). *Let* $\sigma(z) = \max(0, z), z \in \mathbb{R}$ *and* $[\boldsymbol{W}^{(h)}]_{ij} \overset{i.i.d.}{\sim} \mathcal{N}(0, 1), \forall h \in [L], i \in [d^{h+1}], j \in [d^h]$, *there exist constants* $c_1, c_2$, *such that if* $c_1 \frac{L^2 \log\left(\frac{8L}{\delta}\right)}{\epsilon^2} \leq \min_{1 \leq h \leq L} d_h$ *and* $\epsilon \leq \min(c_2, \frac{1}{L})$, *then for any fixed* $\boldsymbol{x}, \boldsymbol{x}' \in \mathbb{R}^{d_0}, \|\boldsymbol{x}\|, \|\boldsymbol{x}'\| \leq 1$, *we have w.p.* $\geq 1 - \delta$, $\forall 0 \leq h \leq L, \forall (\boldsymbol{x}^{(1)}, \boldsymbol{x}^{(2)}) \in \{(\boldsymbol{x}, \boldsymbol{x}), (\boldsymbol{x}, \boldsymbol{x}'), (\boldsymbol{x}', \boldsymbol{x}')\}$,

$$\left| \boldsymbol{g}^{(h)}(\boldsymbol{x}^{(2)})^\top \boldsymbol{g}^{(h)}(\boldsymbol{x}^{(1)}) - \boldsymbol{\Sigma}^{(h)}(\boldsymbol{x}^{(2)}, \boldsymbol{x}^{(1)}) \right| \leq \epsilon.$$

*In other words, if* $\min_{h \in [L]} d_h \geq c_1 \frac{L^2 \log(\frac{L}{\delta_1})}{\epsilon_1^2}$, $\epsilon_1 \leq \min(c_2, \frac{1}{L})$, *then for fixed* $\boldsymbol{x}, \boldsymbol{x}'$,

$$\mathbb{P}\left[ \overline{\mathcal{A}}(\epsilon_1) \right] \geq 1 - \delta_1.$$

**Theorem E.2.** *Let* $\sigma(z) = \max(0, z), z \in \mathbb{R}$, *if* $[\boldsymbol{W}^{(h)}]_{ij} \overset{i.i.d.}{\sim} N(0, 1), \forall h \in [L+1], i \in [d^{h+1}], j \in [d^h]$, *there exist constants* $c_1, c_2$, *such that if* $\min_{h \in [L]} d_h \geq c_1 \frac{L^2 \log(\frac{L}{\delta})}{\epsilon^4}, \epsilon \leq \frac{c_2}{L}$, *then for any fixed* $\boldsymbol{x}, \boldsymbol{x}' \in \mathbb{R}^{d_0}, \|\boldsymbol{x}\|, \|\boldsymbol{x}'\| \leq 1$, *we have w.p.* $1 - \delta, \forall 0 \leq h \leq L, \forall (\boldsymbol{x}^{(1)}, \boldsymbol{x}^{(2)}) \in \{(\boldsymbol{x}, \boldsymbol{x}), (\boldsymbol{x}, \boldsymbol{x}'), (\boldsymbol{x}', \boldsymbol{x}')\}$,

$$\left| \boldsymbol{g}^{(h)}(\boldsymbol{x}^{(2)})^\top \boldsymbol{g}^{(h)}(\boldsymbol{x}^{(1)}) - \boldsymbol{\Sigma}^{(h)}(\boldsymbol{x}^{(2)}, \boldsymbol{x}^{(1)}) \right| \leq \frac{\epsilon^2}{2},$$

*and*

$$\left| \left\langle \mathbf{b}^{(h)}(\boldsymbol{x}^{(1)}), \mathbf{b}^{(h)}(\boldsymbol{x}^{(2)}) \right\rangle - \prod_{h'=h}^{L} \dot{\Sigma}^{(h')}(\boldsymbol{x}^{(1)}, \boldsymbol{x}^{(2)}) \right| < 3L\epsilon.$$

*In other words, if* $\min_{h \in [L]} d_h \geq c_1 \frac{L^2 \log(\frac{L}{\delta_1})}{\epsilon_1^4}$, $\epsilon_1 \leq \frac{c_2}{L}$, *then for fixed* $\boldsymbol{x}, \boldsymbol{x}'$,

$$\mathbb{P}\left[ \overline{\mathcal{A}}\left(\frac{\epsilon_1^2}{8}\right) \bigwedge \overline{\mathcal{B}}(3L\epsilon_1) \right] \geq 1 - \delta$$

Note that for $c_\sigma = 2$ for $\sigma(z) = \max(0, z)$, by definition of $\mathbf{b}^{(h)}$, we have

$$\left\langle \mathbf{b}^{(h)}(\boldsymbol{x}), \mathbf{b}^{(h)}(\boldsymbol{x}') \right\rangle = \frac{2}{d_h} \mathbf{b}^{(h+1)}(\boldsymbol{x})^\top \boldsymbol{W}^{(h+1)} \boldsymbol{D}^{(h)}(\boldsymbol{x}) \boldsymbol{D}^{(h)}(\boldsymbol{x}') \left( \boldsymbol{W}^{(h+1)} \right)^\top \mathbf{b}^{(h+1)}(\boldsymbol{x}').$$

Intuitively, when $d_h$ is large, we can replace $\boldsymbol{W}^{(h+1)}$ by a fresh i.i.d copy $\widetilde{\boldsymbol{W}}$ with a small difference by $\tilde{O}(\frac{1}{\sqrt{d_h}})$ as below. Similar techniques are used in [Yang, 2019].

$$
\begin{aligned}
\left\langle \mathbf{b}^{(h)}(\boldsymbol{x}), \mathbf{b}^{(h)}(\boldsymbol{x}') \right\rangle &= \frac{2}{d_h} \mathbf{b}^{(h+1)}(\boldsymbol{x})^\top \boldsymbol{W}^{(h+1)} \boldsymbol{D}^{(h)}(\boldsymbol{x}) \boldsymbol{D}^{(h)}(\boldsymbol{x}') \left( \boldsymbol{W}^{(h+1)} \right)^\top \mathbf{b}^{(h+1)}(\boldsymbol{x}') \\
&\approx \frac{2}{d_h} \mathbf{b}^{(h+1)}(\boldsymbol{x})^\top \widetilde{\boldsymbol{W}} \boldsymbol{D}^{(h)}(\boldsymbol{x}) \boldsymbol{D}^{(h)}(\boldsymbol{x}') \widetilde{\boldsymbol{W}}^\top \mathbf{b}^{(h+1)}(\boldsymbol{x}') \\
&\approx \mathrm{tr}\left( \frac{2}{d_h} \boldsymbol{D}^{(h)}(\boldsymbol{x}) \boldsymbol{D}^{(h)}(\boldsymbol{x}') \right) \mathbf{b}^{(h+1)}(\boldsymbol{x})^\top \mathbf{b}^{(h+1)}(\boldsymbol{x}') \\
&\approx \dot{\Sigma}^{(h)}(\boldsymbol{x}^{(1)}, \boldsymbol{x}^{(2)}) \prod_{h'=h+1}^{L} \dot{\Sigma}^{(h')}(\boldsymbol{x}^{(1)}, \boldsymbol{x}^{(2)})
\end{aligned}
\tag{20}
$$

The proof is based on a careful control of the following events.

**Lemma E.4.**
$$\mathbb{P}\left[\overline{\mathcal{A}}^L\left(\epsilon_1^2/2\right)\Longrightarrow \overline{\mathcal{C}}\left(2\sqrt{\log\frac{4}{\delta_3}}\right)\right]\geq 1-\delta_3, \quad \forall \epsilon_1 \in [0,1], \delta_3 \in (0,1).$$

**Lemma E.5.**
$$\mathbb{P}\left[\overline{\mathcal{A}}^{h+1}\left(\epsilon_1^2/2\right)\Longrightarrow \overline{\mathcal{D}}^h\left(\epsilon_1 + \sqrt{\frac{2\log\frac{6}{\delta_4}}{d_h}}\right)\right]\geq 1-\delta_4, \quad \forall \epsilon_1 \in [0,1], \delta_4 \in (0,1).$$

**Lemma E.6.**
$$\mathbb{P}\left[\overline{\mathcal{A}}\left(\epsilon_1^2/2\right)\Longrightarrow \overline{\mathcal{D}}\left(\epsilon_1 + \sqrt{\frac{2\log\frac{6L}{\delta_4}}{\min_h d_h}}\right)\right]\geq 1-\delta_4, \quad \forall \epsilon_1 \in [0,1], \delta_4 \in (0,1).$$

*Proof.* Apply union bound on Lemma E.5. □

**Lemma E.7.** *There exists constant $C, C' \in \mathbb{R}$, for any $\epsilon_2, \epsilon_3, \epsilon_4 \in [0,1]$, we have*

$$\mathbb{P}\left[\overline{\mathcal{A}}^L\left(\epsilon_1^2/2\right)\bigwedge \overline{\mathcal{B}}^{h+1}\left(\epsilon_2\right)\bigwedge \overline{\mathcal{C}}\left(\epsilon_3\right)\bigwedge \overline{\mathcal{D}}^h\left(\epsilon_4\right)\Longrightarrow \overline{\mathcal{B}}^h\left(\epsilon_2 + \frac{C'\epsilon_3}{\sqrt{d_h}} + 2\epsilon_4 + C\sqrt{\frac{\log\frac{1}{\delta_2}}{d_h}}\right)\right]\geq 1-\delta_2$$

*Proof of Theorem E.2.* We will use induction on Lemma E.7 to prove Theorem E.2. In the statement of Theorem E.1, we set $\delta_1 = \frac{\delta}{4}, \epsilon_1 = \frac{\epsilon^2}{8}$, for some $c_1, c_2$, we have

$$\mathbb{P}\left[\overline{\mathcal{A}}^L\left(\epsilon^2/8\right)\right]\geq 1-\delta/4 \tag{21}$$

In the statement of Lemma E.6, we set $\delta_4 = \frac{\delta_2}{4}$, and $\epsilon_1 = \frac{\epsilon}{2}$. Note that for $c_1$ large enough $\sqrt{\frac{2\log\frac{24L}{\delta}}{\min_h d_h}} \leq \frac{\epsilon}{2}$ and thus we have

$$\mathbb{P}\left[\overline{\mathcal{A}}\left(\epsilon^2/8\right)\Rightarrow \overline{\mathcal{D}}\left(\epsilon\right)\right]\geq \mathbb{P}\left[\overline{\mathcal{A}}\left(\epsilon^2/8\right)\Rightarrow \overline{\mathcal{D}}\left(\epsilon/2 + \sqrt{\frac{2\log\frac{24L}{\delta}}{\min_h d_h}}\right)\right]\geq 1-\delta/4 \tag{22}$$

In the statement of Lemma E.4, we set $\delta_3 = \frac{\delta}{4}$, and $\epsilon_1 = \frac{\epsilon^2}{8}$, we have

$$\mathbb{P}\left[\overline{\mathcal{A}}^L\left(\epsilon^2/8\right)\Rightarrow \overline{\mathcal{C}}\left(2\sqrt{\log\frac{16}{\delta}}\right)\right]\geq 1-\delta/4 \tag{23}$$

Using union bound on Equation (21),(22),(23), we have

$$\mathbb{P}\left[\overline{\mathcal{A}}^L\left(\epsilon^2/8\right)\bigwedge \overline{\mathcal{C}}\left(2\sqrt{\log\frac{16}{\delta}}\right)\bigwedge \overline{\mathcal{D}}\left(\epsilon\right)\right]\geq 1-\frac{3\delta}{4} \tag{24}$$

Now we will begin the induction argument. First of all, note that $\mathbb{P}\left[\overline{\mathcal{B}}^{L+1}\left(0\right)\right]=1$ by definition.

For $1 \leq h \leq L$ in the statement of Lemma E.7, we set $\epsilon_2 = 3(L+1-h)\epsilon$, $\epsilon_3 = 3\sqrt{\log\frac{16}{\delta}}$, $\epsilon_2 = \epsilon$, $\delta_4 = \frac{\delta}{4L}$. Note that for $c_1$ large enough, $C\sqrt{\frac{\log\frac{1}{\delta_2}}{d_h}} + C'\sqrt{\frac{\log\frac{L}{\delta_2}}{d_h}} < \epsilon$. Thus we have

$$\mathbb{P}\left[\overline{\mathcal{B}}^{h+1}\left((3L-3h)\epsilon\right)\bigwedge \overline{\mathcal{C}}\left(3\sqrt{\log\frac{16}{\delta_2}}\right)\bigwedge \overline{\mathcal{D}}^h\left(\epsilon\right)\Rightarrow \overline{\mathcal{B}}^h\left((3L+2-3h)\epsilon + C\sqrt{\frac{\log\frac{1}{\delta}}{d_h}} + 3C'\sqrt{\frac{\log\frac{16}{\delta}}{d_h}}\right)\right]$$

$$\geq \mathbb{P}\left[\overline{\mathcal{B}}^{h+1}\left((3L-3h)\epsilon\right)\bigwedge \overline{\mathcal{C}}\left(3\sqrt{\log\frac{16}{\delta_2}}\right)\bigwedge \overline{\mathcal{D}}^h\left(\epsilon\right)\Rightarrow \overline{\mathcal{B}}^h\left((3L+3-3h)\epsilon\right)\right]$$

$$\geq 1-\frac{\delta}{4L} \tag{25}$$

Using union bound again on Equation (24) and Equation (25) for every $h$ in $\{1, 2, \ldots, L\}$, we have

$$
\begin{aligned}
&\mathbb{P}\left[\overline{\mathcal{A}}^L\left(\epsilon^2/8\right) \bigwedge \overline{\mathcal{C}}\left(\epsilon\right) \bigwedge \overline{\mathcal{D}}\left(\epsilon\right) \bigwedge \overline{\mathcal{B}}\left(3L\epsilon\right)\right] \\
\geq &\mathbb{P}\left[\overline{\mathcal{A}}^L\left(\epsilon^2/8\right) \bigwedge \overline{\mathcal{C}}\left(\epsilon\right) \bigwedge \overline{\mathcal{D}}\left(\epsilon\right) \bigwedge_{h=1}^{L} \overline{\mathcal{B}}^h\left(3(L+1-h)\epsilon\right)\right] \\
\geq &1 - \left(1 - \mathbb{P}\left[\overline{\mathcal{A}}^L\left(\epsilon^2/8\right) \bigwedge \overline{\mathcal{C}}\left(\epsilon\right) \bigwedge \overline{\mathcal{D}}\left(\epsilon\right)\right]\right) \\
&- \sum_{h=1}^{L}\left(1 - \mathbb{P}\left[\overline{\mathcal{B}}^{h+1}\left((3L-3h)\epsilon\right) \bigwedge \overline{\mathcal{C}}\left(\epsilon\right) \bigwedge \overline{\mathcal{D}}^h\left(\epsilon\right) \Rightarrow \overline{\mathcal{B}}^h\left((3L+3-3h)\epsilon\right)\right]\right) \\
\geq &1 - \delta
\end{aligned}
\tag{26}
$$

$\square$

## E.3 Proof of Lemma E.4

**Lemma E.4.**

$$
\mathbb{P}\left[\overline{\mathcal{A}}^L\left(\epsilon_1^2/2\right) \Longrightarrow \overline{\mathcal{C}}\left(2\sqrt{\log \frac{4}{\delta_3}}\right)\right] \geq 1 - \delta_3, \quad \forall \epsilon_1 \in [0,1], \delta_3 \in (0,1).
$$

*Proof.* For fixed $g^{(L)}(x)$, $f(\theta, x) = W^{(L+1)}g^{(L)}(x) \stackrel{\mathrm{d}}{=} N(0, \|g^{(L)}(x)\|^2)$. Thus by subgaussian concentration[cite], we know w.p. $\geq 1 - \delta$ over the randomness of $W^{(L+1)}$, $|f(\theta, x)| \leq \sqrt{2\log\frac{2}{\delta}}\|g^{(L)}(x)\|$.

For $\epsilon_1 \leq 1$, we have $\epsilon_1^2/2 < 1$, which implies $\|g^{(L)}(x)\|^2 \leq 1 + \frac{\epsilon_1^2}{2} \leq 2$, and thus taking union bound over $x, x'$, we have w.p. $\geq 1 - \delta$, $|f(\theta, x)| \leq 2\sqrt{\log\frac{2}{\delta}}, |f(\theta, x')| \leq 2\sqrt{\log\frac{2}{\delta}}$.

$$
\mathbb{P}\left[\overline{\mathcal{A}}^L\left(\epsilon_1^2/2\right) \Rightarrow \overline{\mathcal{C}}\left(2\sqrt{\log\frac{4}{\delta_3}}\right)\right] \geq \mathbb{P}\left[\overline{\mathcal{C}}\left(2\sqrt{\log\frac{4}{\delta_3}}\right) \mid \overline{\mathcal{A}}^L\left(\epsilon_1^2/2\right)\right] \geq 1 - \delta
$$

$\square$

## E.4 Proof of Lemma E.5

**Lemma E.5.**

$$
\mathbb{P}\left[\overline{\mathcal{A}}^{h+1}\left(\epsilon_1^2/2\right) \Longrightarrow \overline{\mathcal{D}}^h\left(\epsilon_1 + \sqrt{\frac{2\log\frac{6}{\delta_4}}{d_h}}\right)\right] \geq 1 - \delta_4, \quad \forall \epsilon_1 \in [0,1], \delta_4 \in (0,1).
$$

**Lemma E.8.** *Define* $G^{(h)}(x, x') = \begin{bmatrix} g^{(h)}(x)^\top g^{(h)}(x) & g^{(h)}(x)^\top g^{(h)}(x') \\ g^{(h)}(x')^\top g^{(h)}(x) & g^{(h)}(x')^\top g^{(h)}(x') \end{bmatrix}$, *we have for every* $1 \leq h \leq L$,

$$
\left\|G^{(h)}(x, x') - \Lambda^{(h)}(x, x')\right\|_\infty \leq \frac{\epsilon^2}{2} \Rightarrow \left|t_{\dot\sigma}\left(G^{(h)}(x, x')\right) - t_{\dot\sigma}\left(\Lambda^{(h)}(x, x')\right)\right| \leq \epsilon, \forall 0 \leq \epsilon \leq 1.
$$

*Proof.* For simplicity, we denote $G^{(h)}(x, x')$, $\Lambda^{(h)}(x, x')$ by $G, \Lambda$ respectively.

Since $\dot\sigma(z) = \mathbf{1}[z \geq 0]$ is 0-homogeneous, we have

$$
\begin{aligned}
t_{\dot\sigma}(G) =& \hat{t}_{\dot\sigma}\left(\frac{G_{12}}{\sqrt{G_{11}G_{22}}}\right) = \frac{1}{2} + \arcsin\frac{G_{12}}{\sqrt{G_{11}G_{22}}} \\
t_{\dot\sigma}(\Lambda) =& \hat{t}_{\dot\sigma}\left(\frac{\Lambda_{12}}{\sqrt{\Lambda_{11}\Lambda_{22}}}\right) = \frac{1}{2} + \arcsin\frac{\Lambda_{12}}{\sqrt{\Lambda_{11}\Lambda_{22}}} = \frac{1}{2} + \arcsin\Lambda_{12}
\end{aligned}
$$

It is easy to verify that $|\sqrt{G_{11}G_{22}} - 1| \leq \epsilon^2/2$, and thus

$$\left| \frac{G_{12}}{\sqrt{G_{11}G_{22}}} - \Lambda_{12} \right| \leq \left| \frac{G_{12}}{\sqrt{G_{11}G_{22}}} - \frac{\Lambda_{12}}{\sqrt{G_{11}G_{22}}} \right| + |\Lambda_{12}| \left| 1 - \frac{1}{\sqrt{G_{11}G_{22}}} \right| \leq \frac{\epsilon^2/2}{1 - \epsilon^2/2} + \frac{\epsilon^2/2}{1 - \epsilon^2/2} \leq 2\epsilon^2.$$

Thus, by Lemma E.1

$$|t_{\dot{\sigma}}(\boldsymbol{G}) - t_{\dot{\sigma}}(\boldsymbol{\Lambda})| \leq \left| \frac{1}{2} + \arcsin \frac{G_{12}}{\sqrt{G_{11}G_{22}}} - \frac{1}{2} + \arcsin \Lambda_{12} \right| \leq \epsilon.$$

$\square$

**Lemma E.9.** *For any $0 \leq h \leq L - 1$, any fixed $\{\boldsymbol{W}^{(i)}\}_{i=1}^h$, w.p. $1 - \delta$ over the randomness of $\boldsymbol{W}^{(h+1)} \in \mathbb{R}^{d^{h+1} \times d^h}$, we have*

$$\left| 2 \frac{\mathrm{tr}(\boldsymbol{D})}{d_h} - \hat{t}_{\dot{\sigma}}\left(\boldsymbol{G}^{(h)}(\boldsymbol{x}, \boldsymbol{x}')\right) \right| < \sqrt{\frac{2 \log \frac{2}{\delta}}{d_h}}.$$

*Proof.* Notice that $\mathbb{E}\left[2\frac{\mathrm{tr}(\boldsymbol{D})}{d_h}\right] = \hat{t}_{\dot{\sigma}}\left(\boldsymbol{G}^{(h)}(\boldsymbol{x}, \boldsymbol{x}')\right)$, the proof is completed by Chernoff Bound. $\square$

*Proof of Lemma E.5.* Note that $\dot{\Sigma}^{(h)}(\boldsymbol{x}, \boldsymbol{x}') = t_{\sigma'}\left(\boldsymbol{\Sigma}^{(h)}|_{\boldsymbol{x},\boldsymbol{x}'}\right) = \hat{t}_{\sigma'}\left(\boldsymbol{\Lambda}^{(h)}(\boldsymbol{x}, \boldsymbol{x}')\right)$.

Combining Lemma E.8 and Lemma E.9, we have for any $(\boldsymbol{x}, \boldsymbol{x}')$,

$$\mathbb{P}\left[\mathcal{D}^h\left(\boldsymbol{x}, \boldsymbol{x}', \epsilon_1 + \sqrt{\frac{2 \log \frac{6}{\delta}}{d_h}}\right) \mid \mathcal{A}^{h+1}\left(\boldsymbol{x}, \boldsymbol{x}', \epsilon_1^2/2\right)\right] \geq 1 - \frac{\delta}{3}.$$

Taking union bound over $(\boldsymbol{x}, \boldsymbol{x}), (\boldsymbol{x}, \boldsymbol{x}'), (\boldsymbol{x}', \boldsymbol{x}')$ for the choice of $(\boldsymbol{x}, \boldsymbol{x}')$, we have

$$\mathbb{P}\left[\overline{\mathcal{A}}^{h+1}\left(\epsilon_1^2/2\right) \Rightarrow \overline{\mathcal{D}}^h\left(\epsilon_1 + \sqrt{\frac{2 \log \frac{6}{\delta}}{d_h}}\right)\right] \geq \mathbb{P}\left[\overline{\mathcal{D}}^h\left(\epsilon_1 + \sqrt{\frac{2 \log \frac{6}{\delta}}{d_h}}\right) \mid \overline{\mathcal{A}}^{h+1}\left(\epsilon_1^2/2\right)\right] \geq 1 - \delta$$

$\square$

## E.5    Proof of Lemma E.7

**Lemma E.7.** *There exists constant $C, C' \in \mathbb{R}$, for any $\epsilon_2, \epsilon_3, \epsilon_4 \in [0, 1]$, we have*

$$\mathbb{P}\left[\overline{\mathcal{A}}^L\left(\epsilon_1^2/2\right) \bigwedge \overline{\mathcal{B}}^{h+1}\left(\epsilon_2\right) \bigwedge \overline{\mathcal{C}}\left(\epsilon_3\right) \bigwedge \overline{\mathcal{D}}^h\left(\epsilon_4\right) \Longrightarrow \overline{\mathcal{B}}^h\left(\epsilon_2 + \frac{C'\epsilon_3}{\sqrt{d_h}} + 2\epsilon_4 + C\sqrt{\frac{\log \frac{1}{\delta_2}}{d_h}}\right)\right] \geq 1 - \delta_2$$

The proof of Lemma E.7 is based on the following 3 claims, Claim E.1, E.2 and E.3.

**Claim E.1.** *If $\overline{\mathcal{A}}^L\left(\epsilon_1^2/2\right) \bigwedge \overline{\mathcal{B}}^{h+1}\left(\epsilon_2\right) \bigwedge \overline{\mathcal{C}}\left(\epsilon_3\right) \bigwedge \overline{\mathcal{D}}^h\left(\epsilon_4\right)$, then we have*

$$\left| \frac{2\mathrm{tr}(\boldsymbol{D})}{d_h} \left\langle \mathbf{b}^{(h)}(\boldsymbol{x}^{(2)}), \mathbf{b}^{(h)}(\boldsymbol{x}^{(1)}) \right\rangle - \prod_{h'=h}^L \dot{\Sigma}^{(h')}(\boldsymbol{x}^{(1)}, \boldsymbol{x}^{(2)}) \right| \leq \epsilon_2 + 2\epsilon_4.$$

*Proof.*

$$
\left| \frac{2\mathrm{tr}\,(\boldsymbol{D})}{d_h} \left\langle \mathbf{b}^{(h)}(\boldsymbol{x}^{(2)}), \mathbf{b}^{(h)}(\boldsymbol{x}^{(1)}) \right\rangle - \prod_{h'=h}^{L} \dot{\Sigma}^{(h')}(\boldsymbol{x}^{(1)}, \boldsymbol{x}^{(2)}) \right|
$$

$$
\leq \left| \frac{2\mathrm{tr}\,(\boldsymbol{D})}{d_h} - \dot{\Sigma}^{(h)}(\boldsymbol{x}^{(1)}, \boldsymbol{x}^{(2)}) \right| \cdot \left| \left\langle \mathbf{b}^{(h)}(\boldsymbol{x}^{(2)}), \mathbf{b}^{(h)}(\boldsymbol{x}^{(1)}) \right\rangle \right|
$$

$$
+ \left| \dot{\Sigma}^{(h)}(\boldsymbol{x}^{(1)}, \boldsymbol{x}^{(2)}) \right| \cdot \left| \left\langle \mathbf{b}^{(h)}(\boldsymbol{x}^{(2)}), \mathbf{b}^{(h)}(\boldsymbol{x}^{(1)}) \right\rangle - \prod_{h'=h+1}^{L} \dot{\Sigma}^{(h')}(\boldsymbol{x}^{(1)}, \boldsymbol{x}^{(2)}) \right|
$$

$$
\leq 2\epsilon_4 + \epsilon_2
$$

$\square$

For any fixed $h$, let $\boldsymbol{G} = [\boldsymbol{g}^{(h)}(\boldsymbol{x})\ \boldsymbol{g}^{(h)}(\boldsymbol{x}')]$,

**Claim E.2.** *w.p.* $\geq 1 - \frac{\delta_2}{2}$*, if* $\overline{\mathcal{A}}^L\left(\epsilon_1^2/2\right) \bigwedge \overline{\mathcal{B}}^{h+1}\left(\epsilon_2\right) \bigwedge \overline{\mathcal{C}}\left(\epsilon_3\right) \bigwedge \overline{\mathcal{D}}^h\left(\epsilon_4\right)$*, then we have for any* $(\boldsymbol{x}^{(1)}, \boldsymbol{x}^{(2)}) \in \{(\boldsymbol{x}, \boldsymbol{x}), (\boldsymbol{x}, \boldsymbol{x}'), (\boldsymbol{x}', \boldsymbol{x}')\}$,

$$
\left| \frac{2}{d_h} \mathbf{b}^{(h+1)}(\boldsymbol{x}^{(1)})^\top \boldsymbol{W}^{(h+1)} \Pi_{\boldsymbol{G}}^\perp \boldsymbol{D} \Pi_{\boldsymbol{G}}^\perp \left(\boldsymbol{W}^{(h+1)}\right)^\top \mathbf{b}^{(h+1)}(\boldsymbol{x}^{(2)}) - \frac{2\mathrm{tr}\,(\boldsymbol{D})}{d_h} \left\langle \mathbf{b}^{(h)}(\boldsymbol{x}^{(2)}), \mathbf{b}^{(h)}(\boldsymbol{x}^{(1)}) \right\rangle \right|
$$

$$
\leq 16\sqrt{\frac{\log \frac{6}{\delta_2}}{d_h}}.
$$

*As a by-product, for any* $\boldsymbol{x}^{(1)} \in \{\boldsymbol{x}, \boldsymbol{x}'\}$*, we have*

$$
\sqrt{\frac{2}{d_h}} \left\| \mathbf{b}^{(h+1)}(\boldsymbol{x}^{(1)})^\top \boldsymbol{W}^{(h+1)} \Pi_{\boldsymbol{G}}^\perp \boldsymbol{D} \right\| \leq 4\sqrt{\frac{\log \frac{6}{\delta_2}}{d_h}}.
$$

**Lemma E.10** (Gaussian chaos of order 2 [Boucheron et al., 2013])**.** *Let* $\boldsymbol{\xi} \sim N(0, \boldsymbol{I}_n)$ *be an n-dimensional unit gaussian random vector,* $\boldsymbol{A} \in \mathbb{R}^{n \times n}$ *be a symmetric matrix, then for any* $t > 0$*,*

$$
\mathbb{P}\left[ \left| \boldsymbol{\xi}^\top \boldsymbol{A} \boldsymbol{\xi} - \mathbb{E}\left[\boldsymbol{\xi}^\top \boldsymbol{A} \boldsymbol{\xi}\right] \right| > 2\|\boldsymbol{A}\|_F \sqrt{t} + 2\|\boldsymbol{A}\|_2 t \right] \leq 2\exp(-t).
$$

*Or,*

$$
\mathbb{P}\left[ \left| \boldsymbol{\xi}^\top \boldsymbol{A} \boldsymbol{\xi} - \mathbb{E}\left[\boldsymbol{\xi}^\top \boldsymbol{A} \boldsymbol{\xi}\right] \right| > t \right] \leq 2\exp\left(-\frac{t^2}{4(\|\boldsymbol{A}\|_F^2) + \|\boldsymbol{A}\|_2 t}\right).
$$

*Proof of E.2.* It suffices to prove this claim conditioned on every possible realization of

$$
\{\mathbf{b}^{(h+1)}(\boldsymbol{x}^{(1)}), \mathbf{b}^{(h+1)}(\boldsymbol{x}^{(2)}), \boldsymbol{f}^{(h)}(\boldsymbol{x}^{(1)}), \boldsymbol{f}^{(h)}(\boldsymbol{x}^{(2)})\}.
$$

Recall that $\boldsymbol{G} = \left[\boldsymbol{g}^{(h)}(\boldsymbol{x}^{(1)})\ \boldsymbol{g}^{(h)}(\boldsymbol{x}^{(2)})\right]$, we further define $\boldsymbol{F} = \left[\boldsymbol{f}^{(h)}(\boldsymbol{x}^{(1)})\ \boldsymbol{f}^{(h)}(\boldsymbol{x}^{(2)})\right]$. Applying Lemma E.3 on each row of $\boldsymbol{W}^{h+1}$, we have

$$
\boldsymbol{W}^{(h+1)} \Pi_{\boldsymbol{G}}^\perp \overset{\mathrm{d}}{=}_{\boldsymbol{F}=\boldsymbol{W}^{(h+1)}\boldsymbol{G}} \widetilde{\boldsymbol{W}} \Pi_{\boldsymbol{G}}^\perp, \tag{27}
$$

where $\widetilde{\boldsymbol{W}}$ is an iid copy of $\boldsymbol{W}^{(h+1)}$.

Note that $[\mathbf{b}^{(h+1)}(\boldsymbol{x})^\top \widetilde{\boldsymbol{W}}\quad \mathbf{b}^{(h+1)}(\boldsymbol{x}^{(1)})^\top \widetilde{\boldsymbol{W}}]^\top \in \mathbb{R}^{2d_h}$ follows a joint zero-mean gaussian distribution with covariance matrix $\boldsymbol{\Sigma} = \begin{bmatrix} b_{11}\boldsymbol{I}_n & b_{12}\boldsymbol{I}_n \\ b_{21}\boldsymbol{I}_n & b_{22}\boldsymbol{I}_n \end{bmatrix}$, where $b_{ij} = \mathbf{b}^{(h)}(\boldsymbol{x}^{(i)})^\top \mathbf{b}^{(h)}(\boldsymbol{x}^{(j)})$, for $i, j = 1, 2$. In other words, there exists $\boldsymbol{M} \in \mathbb{R}^{2d_h \times 2d_h}$, s.t. $\boldsymbol{M}\boldsymbol{M}^\top = \boldsymbol{\Sigma}$, and

$$
[\mathbf{b}^{(h+1)}(\boldsymbol{x})^\top \widetilde{\boldsymbol{W}}\quad \mathbf{b}^{(h+1)}(\boldsymbol{x}^{(1)})^\top \widetilde{\boldsymbol{W}}]^\top \overset{\mathrm{d}}{=} \boldsymbol{M}\boldsymbol{\xi},
$$

where $\boldsymbol{\xi} \sim N(\boldsymbol{0}, \boldsymbol{I}_{2d_h})$.

Thus conditioned on $\{\mathbf{b}^{(h+1)}(\boldsymbol{x}^{(1)}), \mathbf{b}^{(h+1)}(\boldsymbol{x}^{(2)}), \boldsymbol{g}^{(h)}(\boldsymbol{x}^{(1)}), \boldsymbol{g}^{(h)}(\boldsymbol{x}^{(2)})\}$, we have

$$
\begin{aligned}
&\mathbf{b}^{(h+1)}(\boldsymbol{x}^{(1)})^\top \boldsymbol{W}^{(h+1)} \Pi_{\boldsymbol{G}}^\perp \boldsymbol{D} \Pi_{\boldsymbol{G}}^\perp \left(\boldsymbol{W}^{(h+1)}\right)^\top \mathbf{b}^{(h+1)}(\boldsymbol{x}^{(2)}) \\
&\overset{\mathrm{d}}{=} \mathbf{b}^{(h+1)}(\boldsymbol{x}^{(1)})^\top \widetilde{\boldsymbol{W}} \Pi_{\boldsymbol{G}}^\perp \boldsymbol{D} \Pi_{\boldsymbol{G}}^\perp \left(\widetilde{\boldsymbol{W}}\right)^\top \mathbf{b}^{(h+1)}(\boldsymbol{x}^{(2)}) \\
&\overset{\mathrm{d}}{=} \left([\boldsymbol{I}_{d_h} \quad \mathbf{0}] \boldsymbol{M} \boldsymbol{\xi}\right)^\top \Pi_{\boldsymbol{G}}^\perp \boldsymbol{D} \Pi_{\boldsymbol{G}}^\perp \left([\mathbf{0} \quad \boldsymbol{I}_{d_h}] \boldsymbol{M} \boldsymbol{\xi}\right) \\
&\overset{\mathrm{d}}{=} \frac{1}{2} \boldsymbol{\xi}^\top \boldsymbol{M}^\top \begin{bmatrix} \mathbf{0} & \Pi_{\boldsymbol{G}}^\perp \boldsymbol{D} \Pi_{\boldsymbol{G}}^\perp \\ \Pi_{\boldsymbol{G}}^\perp \boldsymbol{D} \Pi_{\boldsymbol{G}}^\perp & \mathbf{0} \end{bmatrix} \boldsymbol{M} \boldsymbol{\xi}.
\end{aligned}
$$

Now we are ready to prove Claim E.1 by applying Lemma E.10. Let $\boldsymbol{A} = \frac{1}{2} \boldsymbol{M}^\top \begin{bmatrix} \mathbf{0} & \Pi_{\boldsymbol{G}}^\perp \boldsymbol{D} \Pi_{\boldsymbol{G}}^\perp \\ \Pi_{\boldsymbol{G}}^\perp \boldsymbol{D} \Pi_{\boldsymbol{G}}^\perp & \mathbf{0} \end{bmatrix} \boldsymbol{M}$, we have

$$
\mathbb{E}\left[\boldsymbol{\xi}^\top \boldsymbol{A} \boldsymbol{\xi}\right] = \mathrm{tr}\left(\boldsymbol{A}\right) = \frac{1}{2}\mathrm{tr}\left(\begin{bmatrix} \mathbf{0} & \Pi_{\boldsymbol{G}}^\perp \boldsymbol{D} \Pi_{\boldsymbol{G}}^\perp \\ \Pi_{\boldsymbol{G}}^\perp \boldsymbol{D} \Pi_{\boldsymbol{G}}^\perp & \mathbf{0} \end{bmatrix} \boldsymbol{\Sigma}\right) = b_{12}\mathrm{tr}\left(\Pi_{\boldsymbol{G}}^\perp \boldsymbol{D} \Pi_{\boldsymbol{G}}^\perp \boldsymbol{I}_n\right) = b_{12}\mathrm{tr}\left(\boldsymbol{D}\Pi_{\boldsymbol{G}}^\perp\right).
$$

Note that by definition $\Pi_{\boldsymbol{G}}^\perp = \boldsymbol{I}_{d_h} - \Pi_{\boldsymbol{G}}$, and $\mathrm{rank}(\Pi_{\boldsymbol{G}}) \leq 2$, we have

$$
\mathrm{tr}\left(\boldsymbol{D}\Pi_{\boldsymbol{G}}^\perp\right) = \mathrm{tr}\left(\boldsymbol{D}(\boldsymbol{I} - \Pi_{\boldsymbol{G}})\right) = \mathrm{tr}\left(D\right) - \mathrm{tr}\left(\boldsymbol{D}\Pi_{\boldsymbol{G}}\right) = \mathrm{tr}\left(D\right) - \mathrm{tr}\left(\Pi_{\boldsymbol{G}}\boldsymbol{D}\Pi_{\boldsymbol{G}}\right).
$$

Since $\mathbf{0} \preceq \boldsymbol{D} \preceq \boldsymbol{I}_{d_h}$, we have $0 \leq \mathrm{tr}\left(\Pi_{\boldsymbol{G}}\boldsymbol{D}\Pi_{\boldsymbol{G}}\right) \leq 2$, and thus $b_{12}(\mathrm{tr}\left(\boldsymbol{D}\right) - 2) \leq \mathbb{E}\left[\boldsymbol{\xi}^\top \boldsymbol{A} \boldsymbol{\xi}\right] \leq b_{12}\mathrm{tr}\left(\boldsymbol{D}\right)$. For the upper bound of spectrum, note that $\|\boldsymbol{M}\|_2^2 = \|\boldsymbol{\Sigma}\|_2 = \left\|\begin{bmatrix} b_{11} & b_{12} \\ b_{21} & b_{22} \end{bmatrix}\right\|_2 \leq b_{11} + b_{12}$, and $\mathbf{0} \preceq \Pi_{\boldsymbol{G}}^\perp, \boldsymbol{D} \preceq \boldsymbol{I}_{d_h}$, we have

$$
\|\boldsymbol{A}\|_2 \leq \frac{1}{2}\|\boldsymbol{M}\|_2^2 \left\|\Pi_{\boldsymbol{G}}^\perp \boldsymbol{D} \Pi_{\boldsymbol{G}}^\perp\right\|_2 \leq \frac{1}{2}\|\boldsymbol{M}\|_2^2 \left\|\Pi_{\boldsymbol{G}}^\perp\right\|_2 \|\boldsymbol{D}\|_2 \left\|\Pi_{\boldsymbol{G}}^\perp\right\|_2 \leq \frac{b_{11} + b_{22}}{2} \leq \sqrt{2},
$$

and

$$
\|\boldsymbol{A}\|_F \leq \sqrt{2d_h}\|\boldsymbol{A}\|_2 = \frac{\sqrt{2d_h}(b_{11} + b_{22})}{2} \leq 2\sqrt{d_h}.
$$

Thus by Lemma E.10 with $t = \log\frac{6}{\delta_2}$ we have w.p. $1 - \frac{\delta_2}{6}$,

$$
\frac{1}{d_h}\left|\boldsymbol{\xi}^\top \boldsymbol{A} \boldsymbol{\xi} - \mathbb{E}\left[\boldsymbol{\xi}^\top \boldsymbol{A} \boldsymbol{\xi}\right]\right| \leq \frac{1}{d_h}\left(2\|\boldsymbol{A}\|_F \sqrt{t} + 2\|\boldsymbol{A}\|_2 t\right) = 4\sqrt{\frac{\log\frac{6}{\delta_2}}{d_h}} + 2\sqrt{2}\frac{\log\frac{6}{\delta_2}}{d_h}..
$$

Thus we have

$$
\begin{aligned}
&\left|\frac{2}{d_h}\mathbf{b}^{(h+1)}(\boldsymbol{x}^{(1)})^\top \boldsymbol{W}^{(h+1)} \Pi_{\boldsymbol{G}}^\perp \boldsymbol{D} \Pi_{\boldsymbol{G}}^\perp \left(\boldsymbol{W}^{(h+1)}\right)^\top \mathbf{b}^{(h+1)}(\boldsymbol{x}^{(2)}) - \frac{2\mathrm{tr}\left(\boldsymbol{D}\right)}{d_h}\left\langle\mathbf{b}^{(h)}(\boldsymbol{x}^{(2)}), \mathbf{b}^{(h)}(\boldsymbol{x}^{(1)})\right\rangle\right| \\
&\leq \frac{2}{d_h}\left|\boldsymbol{\xi}^\top \boldsymbol{A} \boldsymbol{\xi} - \mathbb{E}\left[\boldsymbol{\xi}^\top \boldsymbol{A} \boldsymbol{\xi}\right]\right| + \left|2\mathbb{E}\left[\boldsymbol{\xi}^\top \boldsymbol{A} \boldsymbol{\xi}\right] - \frac{2\mathrm{tr}\left(\boldsymbol{D}\right)}{d_h}\left\langle\mathbf{b}^{(h)}(\boldsymbol{x}^{(2)}), \mathbf{b}^{(h)}(\boldsymbol{x}^{(1)})\right\rangle\right| \\
&\leq 8\sqrt{\frac{\log\frac{6}{\delta_2}}{d_h}} + 4\sqrt{2}\frac{\log\frac{6}{\delta_2}}{d_h} + \frac{4b_{12}}{d_h} \leq 14\sqrt{\frac{\log\frac{6}{\delta_2}}{d_h}} + \frac{4(1 + \epsilon_2)}{d_h} \quad (2\sqrt{2} \leq 3 \wedge \log\frac{6}{\delta_2} \leq d_h) \\
&\leq 16\sqrt{\frac{\log\frac{6}{\delta_2}}{d_h}} \quad (\epsilon_2 \leq 1 \wedge \sqrt{d_h \log 6} \geq 4).
\end{aligned}
$$

The main part of the claim is completed by taking union bound over $(\boldsymbol{x},\boldsymbol{x}),(\boldsymbol{x},\boldsymbol{x}'),(\boldsymbol{x}',\boldsymbol{x}')$. For the by-product, let $\boldsymbol{x}^{(2)}=\boldsymbol{x}^{(1)}$, and we have

$$\sqrt{\frac{2}{d_h}}\left\|\mathbf{b}^{(h+1)}(\boldsymbol{x}^{(1)})^\top\boldsymbol{W}^{(h+1)}\Pi_{\overline{G}}^\perp\boldsymbol{D}\right\|$$

$$\leq\sqrt{\left|\frac{2}{d_h}\mathbf{b}^{(h+1)}(\boldsymbol{x}^{(1)})^\top\boldsymbol{W}^{(h+1)}\Pi_{\overline{G}}^\perp\boldsymbol{D}\Pi_{\overline{G}}^\perp\left(\boldsymbol{W}^{(h+1)}\right)^\top\mathbf{b}^{(h+1)}(\boldsymbol{x}^{(2)})\right|}$$

$$\leq\sqrt{\frac{2\mathrm{tr}\left(\boldsymbol{D}\right)}{d_h}\left\langle\mathbf{b}^{(h)}(\boldsymbol{x}^{(2)}),\mathbf{b}^{(h)}(\boldsymbol{x}^{(1)})\right\rangle+\left(16\sqrt{\frac{\log\frac{6}{\delta_2}}{d_h}}\right)^2}$$

$$\leq\sqrt{4+\left(16\sqrt{\frac{\log\frac{6}{\delta_2}}{d_h}}\right)^2}$$

$$\leq 2+4\sqrt{\frac{\log\frac{6}{\delta_2}}{d_h}}\leq 6\qquad(\log\frac{6}{\delta_2}\leq d_h)$$

**Claim E.3.** *w.p.* $\geq 1-\frac{\delta_2}{2}$, *if* $\overline{\mathcal{A}}^L\left(\epsilon_1^2/2\right)\bigwedge\overline{\mathcal{B}}^{h+1}\left(\epsilon_2\right)\bigwedge\overline{\mathcal{C}}\left(\epsilon_3\right)\bigwedge\overline{\mathcal{D}}^h\left(\epsilon_4\right)$, *then*

$$\left\|\boldsymbol{\Pi_G}\left(\boldsymbol{W}^{(h+1)}\right)^\top\mathbf{b}^{(h+1)}(\boldsymbol{x})\right\|\leq 2\sqrt{\log\frac{8}{\delta_2}}+\sqrt{2}\epsilon_3,\left\|\boldsymbol{\Pi_G}\left(\boldsymbol{W}^{(h+1)}\right)^\top\mathbf{b}^{(h+1)}(\boldsymbol{x}')\right\|\leq 2\sqrt{\log\frac{8}{\delta_2}}+\sqrt{2}\epsilon_3.$$

*Proof.* It suffices to prove the claim for $\boldsymbol{x}$. We will denote $\boldsymbol{x}$ by $\boldsymbol{x}$, $\boldsymbol{g}^{(h)}(\boldsymbol{x})$ by $\boldsymbol{g}^{(h)}$ and $\mathbf{b}^{(h+1)}(\boldsymbol{x})$ by $\mathbf{b}^{(h+1)}$. We also define $\Pi_g$ as $\boldsymbol{g}\boldsymbol{g}^\top$, and $\Pi_{G/g}=\Pi_G-\Pi_g$. Clearly, $\Pi_{G/g}$ is still a projection matrix of rank 0 or 1.

Since $\left\|\boldsymbol{\Pi_G}\left(\boldsymbol{W}^{(h+1)}\right)^\top\mathbf{b}^{(h+1)}(\boldsymbol{x})\right\|\leq\left\|\boldsymbol{\Pi_g}\left(\boldsymbol{W}^{(h+1)}\right)^\top\mathbf{b}^{(h+1)}\right\|+\left\|\boldsymbol{\Pi}_{G/g}\left(\boldsymbol{W}^{(h+1)}\right)^\top\mathbf{b}^{(h+1)}\right\|$, it suffices to bound these two terms separately.

Recall $\mathbf{b}^{(h+1)}$ is defined as the gradient of $f(\boldsymbol{\theta},\boldsymbol{x})$ with respect to the pre-activation of layer $h+1$, $\boldsymbol{f}^{h+1}$, thus if we view $g$ as a function $\boldsymbol{g}^{(h)},\boldsymbol{W}^{(h+1)},\dots,\boldsymbol{W}^{(L+1)}$, by the rule of back propagation, we have

$$\frac{\partial g(\boldsymbol{g}^{(h)},\boldsymbol{W}^{(h+1)},\dots,\boldsymbol{W}^{(L+1)})}{\partial\boldsymbol{g}^{(h)}}=(\mathbf{b}^{(h+1)})^\top\boldsymbol{W}^{(h+1)}.$$

Note that relu is 1-homogeneous, namely $\forall\lambda\in\mathbb{R}^+,\sigma\left(\lambda z\right)=\lambda\sigma\left(z\right)$, the whole network is also 1-homogeneous in $\boldsymbol{g}^{(h)}$. In other words, we have

$$f(\boldsymbol{g}^{(h)},\boldsymbol{W}^{(h+1)},\dots,\boldsymbol{W}^{(L+1)})$$

$$=\left.\frac{\partial f(\lambda\boldsymbol{g}^{(h)},\boldsymbol{W}^{(h+1)},\dots,\boldsymbol{W}^{(L+1)})}{\partial\lambda}\right|_{\lambda=1}$$

$$=\left\langle\left.\frac{\partial f(\lambda\boldsymbol{g}^{(h)},\boldsymbol{W}^{(h+1)},\dots,\boldsymbol{W}^{(L+1)})}{\partial\lambda\boldsymbol{g}^{(h)}}\right|_{\lambda=1},\left.\frac{\partial\lambda\boldsymbol{g}^{(h)}}{\partial\lambda}\right|_{\lambda=1}\right\rangle$$

$$=\left\langle\frac{\partial g(\boldsymbol{g}^{(h)},\boldsymbol{W}^{(h+1)},\dots,\boldsymbol{W}^{(L+1)})}{\partial\boldsymbol{g}^{(h)}},\boldsymbol{g}^{(h)}\right\rangle$$

$$=(\boldsymbol{g}^{(h)})^\top\left(\boldsymbol{W}^{(h+1)}\right)^\top\mathbf{b}^{(h+1)}$$

By definition of $\Pi_g$, we have

$$\left\|\boldsymbol{\Pi_g}\left(\boldsymbol{W}^{(h+1)}\right)^\top\mathbf{b}\right\|=\left\|\frac{\boldsymbol{g}^{(h)}(\boldsymbol{g}^{(h)})^\top}{\left\|\boldsymbol{g}^{(h)}\right\|^2}\left(\boldsymbol{W}^{(h+1)}\right)^\top\mathbf{b}^{(h+1)})\right\|=\left|\frac{(\boldsymbol{g}^{(h)})^\top}{\left\|\boldsymbol{g}^{(h)}\right\|}\left(\boldsymbol{W}^{(h+1)}\right)^\top\mathbf{b}^{(h+1)})\right|=\frac{|f(\boldsymbol{\theta},\boldsymbol{x})|}{\left\|\boldsymbol{g}^{(h)}\right\|}.$$

Note that $g^{(h)}(x^{(0)})^\top g^{(h)}(x) \geq 1 - \epsilon_1^2/2 \geq \frac{1}{2}$, we have

$$\left\| \Pi_g \left( W^{(h+1)} \right)^\top \mathbf{b} \right\| = \frac{|f(\theta, x)|}{\|g^{(h)}\|} \leq \sqrt{2}\epsilon_3.$$

For the second term $\Pi_{G/g} \left( W^{(h+1)} \right)^\top b^{(h+1)}$, note that conditioned on $g^{(h)}, f^h = \frac{1}{\sqrt{d_{h+1}}} W^{(h+1)} g^{(h)}$ and all $\{W^{(h)}\}_{h'}^{L+1}$ (thus $\mathbf{b}^{(h+1)}$), by Lemma E.3, $\Pi_{G/g}(W^{(h+1)}) \overset{\mathrm{d}}{=\!=} \Pi_{G/g}\widetilde{W}$, where $\widetilde{W}$ is an iid copy of $W^{(h+1)}$. Thus if $\mathrm{rank}(\Pi_{G/g}) = 1$, suppose $\Pi_{G/g} = uu^\top$ for some unit vector $u$, we have

$$\left\| \Pi_{G/g} \left( W^{(h+1)} \right)^\top b^{(h+1)} \right\| = \left| u^\top \left( W^{(h+1)} \right)^\top b^{(h+1)} \right| \overset{\mathrm{d}}{=\!=} \left| u^\top \left( \widetilde{W} \right)^\top b^{(h+1)} \right| \overset{\mathrm{d}}{=\!=} |t|,$$

where $t \sim N(0, \|\mathbf{b}^{(h+1)}\|)$. Hence w.p. $\geq 1 - \delta_2/4$ over the randomness of $W^{(L)}$, $\left\| \Pi_{G/g} \left( W^{(h+1)} \right)^\top b^{(h+1)} \right\| \leq \sqrt{2 \log \frac{8}{\delta_2}} \|\mathbf{b}^{(h+1)}\| \leq \sqrt{2 \log \frac{8}{\delta_2}} \leq 2\sqrt{\log \frac{8}{\delta_2}}$ ($\epsilon_2 < 1$).

If $\mathrm{rank}(\Pi_{G/g}) = 0$, then $\left\| \Pi_{G/g} \left( W^{(h+1)} \right)^\top b^{(h+1)} \right\| = 0 < 2\sqrt{\log \frac{8}{\delta_2}}$. Thus w.p. $\geq 1 - \frac{\delta_2}{4}$,

$$\left\| \Pi_G \left( W^{(h+1)} \right)^\top \mathbf{b}^{(h+1)}(x) \right\| \leq \left\| \Pi_g \left( W^{(h+1)} \right)^\top \mathbf{b}^{(h+1)} \right\| + \left\| \Pi_{G/g} \left( W^{(h+1)} \right)^\top b^{(h+1)} \right\| \leq 2\sqrt{\log \frac{8}{\delta_2}} + \sqrt{2}\epsilon_3.$$

Thus by assumption $\log \frac{8}{\delta_2} \leq d_h$, we have $2\sqrt{\log \frac{8}{\delta_2}} + \sqrt{2}\epsilon_3 \leq 2\sqrt{d_h} + \sqrt{2} \leq 3\sqrt{2d_h}$. $\qquad\square$

Wrapping things up, by combining Claim E.2 and Claim E.3, we have w.p. $\geq 1 - \delta_2$, for any pair of $(x^{(1)}, x^{(2)}) \in \{(x, x), (x, x'), (x', x'))\}$,

$$
\begin{aligned}
&\left| \frac{2}{d_h} \mathbf{b}^{(h+1)}(x^{(1)})^\top \left( W^{(h+1)} \right) D^{(h)}(x^{(1)}) D^{(h)}(x^{(2)}) \left( W^{(h+1)} \right)^\top \mathbf{b}^{(h+1)}(x^{(2)}) - \right. \\
&\left. \frac{2}{d_h} \mathbf{b}^{(h+1)}(x^{(1)})^\top W^{(h+1)} \Pi_G^\perp D \Pi_G^\perp \left( W^{(h+1)} \right)^\top \mathbf{b}^{(h+1)}(x^{(2)}) \right| \\
&\leq \left\| \frac{2}{d_h} \mathbf{b}^{(h+1)}(x^{(1)})^\top W^{(h+1)} \Pi_G D \right\| \cdot \left\| D \Pi_G^\perp \left( W^{(h+1)} \right)^\top \mathbf{b}^{(h+1)}(x^{(2)}) \right\| \\
&+ \left\| \frac{2}{d_h} \mathbf{b}^{(h+1)}(x^{(1)})^\top W^{(h+1)} \Pi_G^\perp D \right\| \cdot \left\| D \Pi_G \left( W^{(h+1)} \right)^\top \mathbf{b}^{(h+1)}(x^{(2)}) \right\| \\
&+ \left\| \frac{2}{d_h} \mathbf{b}^{(h+1)}(x^{(1)})^\top W^{(h+1)} \Pi_G \right\| \cdot \left\| \Pi_G \left( W^{(h+1)} \right)^\top \mathbf{b}^{(h+1)}(x^{(2)}) \right\| \\
&\leq \left( 12\sqrt{2}\sqrt{\frac{\ln \frac{8}{\delta_2}}{d_h}} + 12\epsilon_3 \right) + \left( 12\sqrt{2}\sqrt{\frac{\ln \frac{8}{\delta_2}}{d_h}} + 12\epsilon_3 \right) + \left( 12\sqrt{2}\sqrt{\frac{\ln \frac{8}{\delta_2}}{d_h}} + 12\epsilon_3 \right) \\
&= 36\sqrt{\frac{2 \ln \frac{8}{\delta_2}}{d_h}} + 36\epsilon_3.
\end{aligned}
$$
(28)

Using Equation (28) together with Claim E.1 and Claim E.2, we've finished the proof for Lemma E.7.

$\qquad\square$

# F Proof of Theorem 3.2

In this section, we prove Theorem 3.2. At a high level, our proof first reduces the bounding the perturbation on the prediction to bounding perturbations on the kernel values between each pair of training points and between the testing point and each training point. We use the following notations. We let $X \in \mathbb{R}^{n \times d}$ be the training data. We define $\ker_t(x_{te}, X) \in \mathbb{R}^n$ as

$$[\ker_t(x_{te}, X)]_i = \left\langle \frac{\partial f(\theta(t), x_{te})}{\partial \theta}, \frac{\partial f(\theta(t), x_i)}{\partial \theta} \right\rangle$$

i.e., the kernel induced from the gradient of the prediction with respect to the parameters of the neural network at time $t$.

We also use the following notations for NTK. We let $\boldsymbol{H}^* \in \mathbb{R}^{n \times n}$ be the *fixed* kernel matrix defined in Equation (4). We let $\ker_{ntk}(\boldsymbol{x}_{te}, \boldsymbol{X}) \in \mathbb{R}^n$ be the kernel values between $\boldsymbol{x}_{te}$ and each training data. Note with this notation, we can write

$$f_{ntk}(\boldsymbol{x}_{te}) = \ker_{ntk}(\boldsymbol{x}_{te}, \boldsymbol{x})^\top (\boldsymbol{H}^*)^{-1} \boldsymbol{y}. \tag{29}$$

We prove a lemma to reduce the prediction perturbation bound to the kernel perturbation bound.

**Lemma F.1** (Kernel Value Perturbation $\Rightarrow$ Prediction Perturbation). *Fix $\epsilon_{\boldsymbol{H}} \leq \frac{1}{2}\lambda_0$. Suppose $|f_{nn}(\boldsymbol{\theta}(0), \boldsymbol{x}_i)| \leq \epsilon_{init}$ for $i = 1, \ldots, n$ and $|f_{nn}(\boldsymbol{\theta}(0), \boldsymbol{x}_{te})| \leq \epsilon_{init}$ and $\|\boldsymbol{u}_{nn}(0) - \boldsymbol{y}\|_2 = O(\sqrt{n})$. Furthermore, if for all $t \geq 0$ $\|\ker_{ntk}(\boldsymbol{x}_{te}, \boldsymbol{X}) - \ker_t(\boldsymbol{x}_{te}, \boldsymbol{X})\|_2 \leq \epsilon_{test}$ and $\|\boldsymbol{H}^* - \boldsymbol{H}(t)\|_2 \leq \epsilon_{\boldsymbol{H}}$, then we have*

$$|f_{ntk}(\boldsymbol{x}_{te}) - f_{nn}(\boldsymbol{x}_{te})| \leq O\left(\epsilon_{init} + \frac{\sqrt{n}}{\lambda_0}\epsilon_{test} + \frac{\sqrt{n}}{\lambda_0^2}\log\left(\frac{n}{\epsilon_{\boldsymbol{H}}\lambda_0\kappa}\right)\epsilon_{\boldsymbol{H}}\right).$$

*Proof.* Our proof relies a careful analysis on the trajectories induced by gradient flows for optimizing the neural network and the NTK predictor.

Note while Equation (29) is a closed-form formula, we can rewrite it in an integral form using the following observations. For any $\boldsymbol{x} \in \mathbb{R}^d$, we let $\phi(\boldsymbol{x})$ be the feature map induced by NTK. Note the expression in Equation (29) can be rewritten as $f_{ntk}(\boldsymbol{x}_{te}) = \kappa\phi(\boldsymbol{x}_{te})^\top \boldsymbol{\beta}_{ntk}$ where $\boldsymbol{\beta}_{ntk}$ satisfies

$$\min_{\boldsymbol{\beta}} \|\boldsymbol{\beta}\|_2$$

$$\text{such that } \kappa\phi(\boldsymbol{x}_i)^\top \boldsymbol{\beta} = y_i \text{ for } i = 1, \ldots, n.$$

The solution to this program can written as applying gradient flow on

$$\min_{\boldsymbol{\beta}} \sum_{i=1}^n \frac{1}{2n}\left(\kappa\phi(\boldsymbol{x}_i)^\top \boldsymbol{\beta} - y_i\right)^2$$

with initialization $\boldsymbol{\beta}(0) = \boldsymbol{0}$. We use $\boldsymbol{\beta}(t)$ to denote this parameter at time $t$ trained by gradient flow and $f_{ntk}(\boldsymbol{x}_{te}, \boldsymbol{\beta}(t))$ be the predictor for $\boldsymbol{x}_{te}$ at time $t$. With these notations, we rewrite

$$f_{ntk}(\boldsymbol{x}_{te}) = \int_{t=0}^\infty \frac{df_{ntk}(\boldsymbol{\beta}(t), \boldsymbol{x}_{te})}{dt} dt$$

where we have used the fact that the initial prediction is $0$. Now we take a closer look at the time derivative:

$$\begin{aligned}
\frac{df_{ntk}(\boldsymbol{\beta}(t), \boldsymbol{x}_{te})}{dt} &= \left\langle \frac{\partial f(\boldsymbol{\beta}(t), \boldsymbol{x}_{te})}{\partial \boldsymbol{\beta}(t)}, \frac{d\boldsymbol{\beta}(t)}{dt} \right\rangle \\
&= \left\langle \frac{\partial f(\boldsymbol{\beta}(t), \boldsymbol{x}_{te})}{\partial \boldsymbol{\beta}(t)}, -\frac{\partial L(\boldsymbol{\beta}(t), \{\boldsymbol{x}_i\}_{i=1}^n)}{\partial \boldsymbol{\beta}(t)} \right\rangle \\
&= -\frac{1}{n}\left\langle \frac{\partial f(\boldsymbol{\beta}(t), \boldsymbol{x}_{te})}{\partial \boldsymbol{\beta}(t)}, \sum_{i=1}^n (u_{ntk,i}(t) - y_i)\frac{\partial f(\boldsymbol{\beta}(t), \boldsymbol{x}_i)}{\boldsymbol{\beta}(t)} \right\rangle \\
&= -\frac{1}{n}\left\langle \kappa\phi(\boldsymbol{x}_{te}), \sum_{i=1}^n (u_{ntk,i}(t) - y_i)\kappa\phi(\boldsymbol{x}_i) \right\rangle \\
&= -\frac{\kappa^2}{n}\ker_{ntk}(\boldsymbol{x}_{te}, \boldsymbol{X})^\top (\boldsymbol{u}_{ntk}(t) - \boldsymbol{y})
\end{aligned}$$

where $u_{ntk,i}(t) = f_{ntk}(\boldsymbol{\beta}(t), \boldsymbol{x}_i)$ and $\boldsymbol{u}_{ntk}(t) \in \mathbb{R}^n$ with $[\boldsymbol{u}_{ntk}(t)]_i = u_{ntk,i}(t)$. Similarly, for the NN predictor, we can obtain a time derivative of the same form.

$$\frac{df_{nn}(\boldsymbol{\theta}(t), \boldsymbol{x}_{te})}{dt} = \kappa\left\langle \frac{\partial f(\boldsymbol{\theta}(t), \boldsymbol{x}_{te})}{\partial \boldsymbol{\theta}(t)}, \frac{d\boldsymbol{\theta}(t)}{dt} \right\rangle$$

$$= \kappa \left\langle \frac{\partial f(\boldsymbol{\theta}(t), \boldsymbol{x}_{te})}{\partial \boldsymbol{\theta}(t)}, -\frac{\partial L(\boldsymbol{\theta}(t), \{\boldsymbol{x}_i\}_{i=1}^n)}{\partial \boldsymbol{\theta}(t)} \right\rangle$$

$$= -\frac{\kappa^2}{n} \left\langle \frac{\partial f(\boldsymbol{\theta}(t), \boldsymbol{x}_{te})}{\partial \boldsymbol{\theta}(t)}, \sum_{i=1}^n (u_{nn,i}(t) - y_i) \frac{\partial f(\boldsymbol{\theta}(t), \boldsymbol{x}_i)}{\boldsymbol{\theta}(t)} \right\rangle$$

$$= -\frac{\kappa^2}{n} \ker_t(\boldsymbol{x}_{te}, \boldsymbol{X})^\top (\boldsymbol{u}_{nn}(t) - \boldsymbol{y})$$

We thus we analyze the difference between the NN predictor and NTK predictor via this integral form

$$|f_{nn}(\boldsymbol{x}_{te}) - f_{ntk}(\boldsymbol{x}_{te})|$$

$$= \left| f_{nn}(\boldsymbol{\theta}(0), \boldsymbol{x}_{te}) + \int_{t=0}^\infty \left( \frac{df_{nn}(\boldsymbol{\theta}(t), \boldsymbol{x}_{te})}{dt} - \frac{df_{ntk}(\boldsymbol{\beta}(t), \boldsymbol{x}_{te})}{dt} \right) dt \right|$$

$$= |f_{nn}(\boldsymbol{\theta}(0), \boldsymbol{x}_{te})| + \left| -\frac{\kappa^2}{n} \int_{t=0}^\infty \left( \ker_t(\boldsymbol{x}_{te}, \boldsymbol{X})^\top (\boldsymbol{u}_{nn}(t) - \boldsymbol{y}) - \ker_{ntk}(\boldsymbol{x}_{te}, \boldsymbol{X})^\top (\boldsymbol{u}_{ntk}(t) - \boldsymbol{y}) \right) dt \right|$$

$$\leq \epsilon_{init} + \frac{\kappa^2}{n} \left| \int_{t=0}^\infty (\ker_t(\boldsymbol{x}_{te}, \boldsymbol{X}) - \ker_{ntk}(\boldsymbol{x}_{te}, \boldsymbol{X}))^\top (\boldsymbol{u}_{nn}(t) - \boldsymbol{y}) dt \right|$$

$$+ \frac{\kappa^2}{n} \left| \int_{t=0}^\infty \ker_{ntk}(\boldsymbol{x}_t, \boldsymbol{X})^\top (\boldsymbol{u}_{nn}(t) - \boldsymbol{u}_{ntk}(t)) dt \right|$$

$$\leq \epsilon_{init} + \kappa^2 \max_{0 \leq t \leq \infty} \|\ker_t(\boldsymbol{x}_{te}, \boldsymbol{X}) - \ker_{ntk}(\boldsymbol{x}_{te}, \boldsymbol{X})\|_2 \int_{t=0}^\infty \|\boldsymbol{u}_{nn}(t) - \boldsymbol{y}\|_2 dt$$

$$+ \kappa^2 \max_{0 \leq t \leq \infty} \|\ker_{ntk}(\boldsymbol{x}_{te}, \boldsymbol{X})\|_2 \int_{t=0}^\infty \|\boldsymbol{u}_{nn}(t) - \boldsymbol{u}_{ntk}(t)\|_2 dt$$

$$\leq \epsilon_{init} + \kappa^2 \epsilon_{test} \int_{t=0}^\infty \|\boldsymbol{u}_{nn}(t) - \boldsymbol{y}\|_2 dt + \kappa^2 \max_{0 \leq t \leq \infty} \|\ker_{ntk}(\boldsymbol{x}_{te}, \boldsymbol{X})\|_2 \int_{t=0}^\infty \|\boldsymbol{u}_{nn}(t) - \boldsymbol{u}_{ntk}(t)\|_2 dt$$

For the second term, recall $\|\boldsymbol{H}^* - \boldsymbol{H}(t)\|_2 \leq \epsilon_{\boldsymbol{H}}$ by our assumption so $\lambda_{\min}(\boldsymbol{H}(t)) \geq \frac{1}{2}\lambda_0$. Using this fact we know $\|\boldsymbol{u}_{nn}(t) - \boldsymbol{y}\|_2 \leq \exp(-\frac{\kappa^2}{2}\lambda_0 t) \|\boldsymbol{u}_{nn}(0) - \boldsymbol{y}\|_2$. Therefore, we can bound

$$\int_0^\infty \|\boldsymbol{u}_{nn}(t) - \boldsymbol{y}\|_2 dt = \int_{t=0}^\infty \exp(-\frac{\kappa^2}{2}\lambda_0 t) \|\boldsymbol{u}_{nn}(0) - \boldsymbol{y}\|_2 dt = O\left( \frac{\sqrt{n}}{\kappa^2 \lambda_0} \right).$$

To bound $\int_{t=0}^\infty \|\boldsymbol{u}_{nn}(t) - \boldsymbol{u}_{ntk}(t)\|_2$, we observe that $\boldsymbol{u}_{nn}(t) \to \boldsymbol{y}$ and $\boldsymbol{u}_{ntk}(t) \to \boldsymbol{y}$ with linear convergence rate. Therefore, we can choose some $t_0 = \frac{C}{\lambda_0 \kappa^2} \log\left( \frac{n}{\epsilon_{\boldsymbol{H}} \lambda_0 \kappa} \right)$ so that

$$\int_{t_0}^\infty \|\boldsymbol{u}_{nn}(t) - \boldsymbol{u}_{ntk}(t)\|_2 dt$$

$$\leq \int_{t_0}^\infty \|\boldsymbol{u}_{nn}(t) - \boldsymbol{y}\|_2 dt + \int_{t_0}^\infty \|\boldsymbol{u}_{ntk}(t) - \boldsymbol{y}\|_2 dt$$

$$\leq O\left( \frac{1}{\lambda_0 \kappa^2} (\|\boldsymbol{u}_{nn}(t_0) - \boldsymbol{y}\|_2 + \|\boldsymbol{u}_{ntk}(t_0) - \boldsymbol{y}\|_2) \right)$$

$$\leq O\left( \frac{\sqrt{n}}{\lambda_0 \kappa} \exp\left(-\lambda_0 \kappa^2 t_0\right) \right)$$

$$\leq O(\epsilon_{\boldsymbol{H}}).$$

Thus it suffices to bound

$$\int_{t=0}^{t_0} \|\boldsymbol{u}_{nn}(t) - \boldsymbol{u}_{ntk}(t)\|_2 dt \leq t_0 \max_{0 \leq t \leq t_0} \|\boldsymbol{u}_{nn}(t) - \boldsymbol{u}_{ntk}(t)\|_2.$$

First observe that

$$\|\boldsymbol{u}_{nn}(t) - \boldsymbol{u}_{ntk}(t)\|_2$$

$$\leq \|\boldsymbol{u}_{nn}(0)\|_2 + \int_{\tau=0}^{t} \left\| \frac{d\left(\boldsymbol{u}_{nn}(\tau) - \boldsymbol{u}_{ntk}(\tau)\right)}{d\tau} \right\|_2 d\tau$$

$$\leq \epsilon_{init}\sqrt{n} + \int_{\tau=0}^{t} \left\| \frac{d\left(\boldsymbol{u}_{nn}(\tau) - \boldsymbol{u}_{ntk}(\tau)\right)}{d\tau} \right\|_2 d\tau.$$

Note

$$\frac{d\left(\boldsymbol{u}_{nn}(t) - \boldsymbol{u}_{ntk}(t)\right)}{dt}$$

$$= -\kappa^2 \boldsymbol{H}(t)\left(\boldsymbol{u}_{nn}(t) - \boldsymbol{y}\right) + \kappa^2 \boldsymbol{H}^*\left(\boldsymbol{u}_{ntk}(t) - \boldsymbol{y}\right)$$

$$= -\kappa^2 \boldsymbol{H}^*\left(\boldsymbol{u}_{nn}(t) - \boldsymbol{u}_{ntk}(t)\right) + \kappa^2\left(\boldsymbol{H}^* - \boldsymbol{H}(t)\right)\left(\boldsymbol{u}_{nn}(t) - \boldsymbol{y}\right)$$

Since $\boldsymbol{H}^*$ is positive semidefinite, $-\boldsymbol{H}^*\left(\tilde{\boldsymbol{u}}(t) - \boldsymbol{u}(t)\right)$ term only makes $\|\tilde{\boldsymbol{u}}(t) - \boldsymbol{u}(t)\|_2$ smaller. Therefore, we have

$$\|\boldsymbol{u}_{nn}(t) - \boldsymbol{u}_{ntk}(t)\|_2 \leq \kappa^2 \int_{\tau=0}^{t} \|\boldsymbol{u}_{nn}(\tau) - \boldsymbol{y}\|_2 \|\boldsymbol{H}(t) - \boldsymbol{H}^*\|_2$$

$$\leq t\kappa^2 \|\boldsymbol{u}_{nn}(0) - \boldsymbol{y}\|_2 \epsilon_{\boldsymbol{H}}$$

$$\leq O\left(t\kappa^2 \sqrt{n}\epsilon_{\boldsymbol{H}}\right).$$

Therefore, we have

$$\int_{t=0}^{t_0} \|\boldsymbol{u}_{nn}(t) - \boldsymbol{u}_{ntk}(t)\|_2 \, dt \leq O\left(t_0^2 \sqrt{n}\kappa^2 \epsilon_{\boldsymbol{H}}\right) = O\left(\frac{\sqrt{n}}{\lambda_0^2 \kappa^2} \log\left(\frac{n}{\epsilon_{\boldsymbol{H}}\lambda_0\kappa}\right) \epsilon_{\boldsymbol{H}}\right).$$

Lastly, we put things together and get

$$|f_{ntk}(\boldsymbol{x}_{te}) - f_{nn}(\boldsymbol{x}_{te})| \leq O\left(\epsilon_{init} + \epsilon_{test}\frac{\sqrt{n}}{\lambda_0} + \frac{\sqrt{n}}{\lambda_0^2}\log\left(\frac{n}{\epsilon_{\boldsymbol{H}}\lambda_0\kappa}\right)\epsilon_{\boldsymbol{H}}\right).$$

$\square$

*Proof of Theorem 3.2.* By Lemma F.1, the problem now reduces to (i) choose $\kappa$ small enough to make $\epsilon_{init} = O(\epsilon)$ and (ii) show when the width is large enough then $\epsilon_{\boldsymbol{H}}$ and $\epsilon_{test}$ are both $O(\epsilon)$. For (i), based on Theorem E.1 and the union bound, we can just choose $\kappa = O\left(\frac{\epsilon}{\log(n/\delta)}\right)$ to make $\epsilon_{init} = O(\epsilon)$ with probability $1 - \delta$. For (ii), we will use Theorem 3.1 and Lemma F.2 below, and then apply the union bound. $\square$

### F.1 Kernel Perturbation During Training

In this subsection we prove the following lemma.

**Lemma F.2** (Kernel Perturbation Bound During Training). *Fix $\omega \leq \mathrm{poly}(1/L, 1/n, 1/\log(1/\delta), \lambda_0)$. Suppose we set $m \geq \mathrm{poly}(1/\omega)$ and $\kappa \leq 1$. Then with probability at least $1 - \delta$ over random initialization, we have for all $t \geq 0$, for any $(\boldsymbol{x}, \boldsymbol{x}') \in \{\boldsymbol{x}_1, \ldots, \boldsymbol{x}_n, \boldsymbol{x}_{te}\} \times \{\boldsymbol{x}_1, \ldots, \boldsymbol{x}_n, \boldsymbol{x}_{te}\}$*

$$|\ker_t(\boldsymbol{x}, \boldsymbol{x}') - \ker_0(\boldsymbol{x}, \boldsymbol{x}')| \leq \omega$$

Recall for any fixed $\boldsymbol{x}$ and $\boldsymbol{x}'$, Theorem 3.1 shows $|\ker_0(\boldsymbol{x}, \boldsymbol{x}') - \ker_{ntk}(\boldsymbol{x}, \boldsymbol{x}')| \leq \epsilon$ if $m$ is large enough. The next lemma shows we can reduce the problem of bounding the perturbation on the kernel value to the perturbation on the gradient.

**Lemma F.3** (Gradient Perturbation $\Rightarrow$ Kernel Perturbation). *If $\left\| \frac{\partial f(\boldsymbol{\theta}(t), \boldsymbol{x})}{\partial \boldsymbol{\theta}} - \frac{\partial f(\boldsymbol{\theta}(0), \boldsymbol{x})}{\partial \boldsymbol{\theta}} \right\| \leq \epsilon$ and $\left\| \frac{\partial f(\boldsymbol{\theta}(t), \boldsymbol{x}')}{\partial \boldsymbol{\theta}} - \frac{\partial f(\boldsymbol{\theta}(0), \boldsymbol{x}')}{\partial \boldsymbol{\theta}} \right\| \leq \epsilon$, we have*

$$|\ker_t(\boldsymbol{x}, \boldsymbol{x}') - \ker_0(\boldsymbol{x}, \boldsymbol{x}')| \leq O(\epsilon)$$

*Proof.* By the proof of Theorem 3.1, we know $\left\| \frac{\partial f(\boldsymbol{\theta}(0), \boldsymbol{x})}{\partial \boldsymbol{\theta}} \right\|_2 = O(1)$. Then we can just use triangle inequality. $\square$

Now we proceed to analyze the perturbation on the gradient. Note we can focus on the perturbation on a single sample $\boldsymbol{x}$ because we can later take a union bound. Therefore, in the rest of this section, we drop the dependency on a specific sample. We use the following notations in this section. Recall $\boldsymbol{W}^{(1)}, \ldots, \boldsymbol{W}^{(L+1)} \sim \mathcal{N}(\boldsymbol{0}, \boldsymbol{I})$ and we denote $\triangle\boldsymbol{W}^{(1)}, \ldots, \triangle\boldsymbol{W}^{(L+1)}$ the perturbation matrices. We let $\widetilde{\boldsymbol{W}}^{(h)} = \boldsymbol{W}^{(h)} + \triangle\boldsymbol{W}^{(h)}$. We let $\tilde{\boldsymbol{g}}^{(0)} = \boldsymbol{g}^{(0)} = \boldsymbol{x}$ and for $h = 1, \ldots, L$ we define

$$\boldsymbol{z}^{(h)} = \sqrt{\frac{2}{m}}\boldsymbol{W}^{(h)}\boldsymbol{g}^{(h-1)}, \quad \boldsymbol{g}^{(h)} = \sigma\left(\boldsymbol{z}^{(h)}\right),$$

$$\tilde{\boldsymbol{z}}^{(h)} = \sqrt{\frac{2}{m}}\widetilde{\boldsymbol{W}}^{(h)}\tilde{\boldsymbol{g}}^{(h-1)}, \quad \tilde{\boldsymbol{g}}^{(h)} = \sigma\left(\tilde{\boldsymbol{z}}^{(h)}\right).$$

For $h = 1, \ldots, L$, $i = 1, \ldots, m$, we denote

$$[\boldsymbol{D}^{(h)}]_{ii} = \mathbf{1}\left\{\left[\boldsymbol{W}^{(h)}\right]_{i,:}\boldsymbol{g}^{(h-1)} \geq 0\right\}$$

$$[\widetilde{\boldsymbol{D}}^{(h)}]_{ii} = \mathbf{1}\left\{\left[\widetilde{\boldsymbol{W}}^{(h)}\right]_{i,:}\tilde{\boldsymbol{g}}^{(h-1)} \geq 0\right\}.$$

**Remark F.1.** *Note $\boldsymbol{z}^{(h)} = \sqrt{\frac{2}{m}}\boldsymbol{f}^{(h)}$. Here we use $\boldsymbol{z}^{(h)}$ instead of $\boldsymbol{f}^{(h)}$ for the ease of presentation.*

For convenience, we also define

$$\triangle\boldsymbol{D}^{(h)} = \widetilde{\boldsymbol{D}}^{(h)} - \boldsymbol{D}^{(h)}.$$

Recall the gradient to $\boldsymbol{W}^{(h)}$ is:

$$\frac{\partial f(\boldsymbol{\theta}, \boldsymbol{x})}{\partial \boldsymbol{W}^{(h)}} = \boldsymbol{b}^{(h)}\left(\boldsymbol{g}^{(h-1)}\right)^{\top}$$

Similarly, we have

$$\frac{\partial f(\boldsymbol{\theta}, \boldsymbol{x})}{\partial \widetilde{\boldsymbol{W}}^{(h)}} = \tilde{\boldsymbol{b}}^{(h)}\left(\tilde{\boldsymbol{g}}^{(h-1)}\right)^{\top}$$

where

$$\tilde{\boldsymbol{b}}^{(h)} = \begin{cases} 1 & \text{if } h = L+1 \\ \sqrt{\frac{2}{m}}\widetilde{\boldsymbol{D}}^{(h)}\left(\widetilde{\boldsymbol{W}}^{(h+1)}\right)^{\top}\tilde{\boldsymbol{b}}^{(h+1)} & \text{Otherwise} \end{cases}.$$

This gradient formula allows us to bound the perturbation on $\triangle\boldsymbol{g}^{(h)} \triangleq \tilde{\boldsymbol{g}}^{(h)} - \boldsymbol{g}^{(h)}$ and $\triangle\boldsymbol{b}^{(h)} \triangleq \tilde{\boldsymbol{b}}^{(h)} - \boldsymbol{b}^{(h)}$ separately. The following lemmas adapted from [Allen-Zhu et al., 2018b] show with high probability over the initialization, bounding the perturbation on $\triangle\boldsymbol{g}^{(h)}$ and $\triangle\boldsymbol{b}^{(h)}$ can be reduced to bounding the perturbation on weight matrices.

**Lemma F.4** (Adapted from Lemma 5.2 in [Allen-Zhu et al., 2018b]). *Suppose*

$$\omega \leq \text{poly}\left(1/n, \lambda_0, 1/L, 1/\log(m), \epsilon, 1/\log(1/\delta)\right).$$

*Then with probability at least $1 - \delta$ over random initialization, if $\left\|\triangle\boldsymbol{W}^{(h)}\right\|_2 \leq \sqrt{m}\omega$ for all $h = 1, \ldots, L$, we have $\left\|\triangle\boldsymbol{g}^{(h)}\right\|_2 = O(\omega L^{5/2}\sqrt{\log m})$ for all $h = 1, \ldots, L$.*

**Remark F.2.** *While Allen-Zhu et al. [2018b] did not consider the perturbation on $\boldsymbol{W}^{(1)}$, by scrutinizing their proof, it is easy to see that the perturbation bounds still hold even if there is a small perturbation on $\boldsymbol{W}^{(1)}$.*

The next lemma bounds the backward vector, adapted from

**Lemma F.5** (Adapted from Lemma 5.7 in [Allen-Zhu et al., 2018b]). *Suppose*

$$\omega \leq \text{poly}\left(1/n, \lambda_0, 1/L, 1/\log(m), \epsilon, 1/\log(1/\delta)\right).$$

*Then with probability at least $1 - \delta$ over random initialization, if $\left\|\triangle\boldsymbol{W}^{(h)}\right\|_2 \leq \sqrt{m}\omega$ for all $h = 1, \ldots, L+1$, we have for all $h = 1, \ldots, L+1$, $\left\|\tilde{\boldsymbol{b}}^{(h)} - \boldsymbol{b}^{(h)}\right\|_2 = O\left(\omega^{1/3}L^2\sqrt{\log m}\right).$*

**Remark F.3.** *While Allen-Zhu et al. [2018b] did not consider the perturbation on $\boldsymbol{W}^{(L+1)}$, by scrutinizing their proof, it is easy to see that the perturbation bounds still hold even if there is a small perturbation on $\boldsymbol{W}^{(L+1)}$.*

Combing these two lemmas and the result for the initialization (Theorem 3.1), we have the following "gradient-Lipschitz" lemma.

**Lemma F.6.** *Suppose $\omega \leq \text{poly}\left(1/n, \lambda_0, 1/L, 1/\log(m), \epsilon, 1/\log(1/\delta)\right)$. Then with probability at least $1 - \delta$ over random initialization, if $\left\|\triangle \boldsymbol{W}^{(h)}\right\|_2 \leq \sqrt{m}\omega$ for all $h = 1, \ldots, L+1$, we have for all $h = 1, \ldots, L+1$:*

$$\left\| \tilde{\boldsymbol{b}}^{(h)} \left(\tilde{\boldsymbol{g}}^{(h-1)}\right)^\top - \boldsymbol{b}^{(h)} \left(\boldsymbol{g}^{(h-1)}\right)^\top \right\|_F = O\left(\omega^{1/3} L^{5/2} \sqrt{\log m}\right)$$

*Proof.* We use the triangle inequality to bound the perturbation

$$\left\| \tilde{\boldsymbol{b}}^{(h)} \left(\tilde{\boldsymbol{g}}^{(h-1)}\right)^\top - \boldsymbol{b}^{(h)} \left(\boldsymbol{g}^{(h-1)}\right)^\top \right\|_F$$
$$\leq \left\| \tilde{\boldsymbol{b}}^{(h)} \left(\tilde{\boldsymbol{g}}^{(h-1)}\right)^\top - \boldsymbol{b}^{(h)} \left(\tilde{\boldsymbol{g}}^{(h-1)}\right)^\top \right\|_F + \left\| \boldsymbol{b}^{(h)} \left(\tilde{\boldsymbol{g}}^{(h-1)}\right)^\top - \boldsymbol{b}^{(h)} \left(\boldsymbol{g}^{(h-1)}\right)^\top \right\|_F$$
$$\leq \left\| \triangle\boldsymbol{b}^{(h)} \left(\boldsymbol{g}^{(h-1)} + \triangle\boldsymbol{g}^{(h-1)}\right)^\top \right\|_F + \left\| \boldsymbol{b}^{(h)} \left(\triangle\boldsymbol{g}^{(h-1)}\right)^\top \right\|_F$$
$$= O\left(\omega^{1/3} L^{5/2} \sqrt{\log m}\right).$$

$\square$

The following lemma shows for given weight matrix, if we have linear convergence and other weight matrices are only perturbed by a little, then the given matrix is only perturbed by a little as well.

**Lemma F.7.** *Fix $h \in [L+1]$ and a sufficiently small $\omega \leq \text{poly}\left(1/n, \lambda_0, 1/L, 1/\log(m), \epsilon, 1/\log(1/\delta), \kappa\right)$. Suppose for all $t \geq 0$, $\|\boldsymbol{u}_{nn}(t) - \boldsymbol{y}\|_2 \leq \exp\left(-\frac{1}{2}\kappa^2 \lambda_0 t\right)\|\boldsymbol{u}_{nn}(0) - \boldsymbol{y}\|_2$ and $\left\|\boldsymbol{W}^{(h')}(t) - \boldsymbol{W}^{(h')}(0)\right\|_F \leq \omega\sqrt{m}$ for $h' \neq h$. Then if $m \geq \text{poly}\left(1/\omega\right)$ we have with probability at least $1 - \delta$ over random initialization, for all $t \geq 0$*

$$\left\| \boldsymbol{W}^{(h)}(t) - \boldsymbol{W}^{(h)}(0) \right\|_F = O\left(\frac{\sqrt{n}}{\lambda_0}\right) \leq \omega\sqrt{m}.$$

*Proof.* We let $C, C_0, C_1, C_2, C_3 > 0$ be some absolute constants.

$$\left\| \boldsymbol{W}^{(h)}(t) - \boldsymbol{W}^{(h)}(0) \right\|_F$$
$$= \left\| \int_0^t \frac{d\boldsymbol{W}^{(h)}(\tau)}{d\tau} d\tau \right\|_F$$
$$= \left\| \int_0^t \frac{\partial L(\boldsymbol{\theta}(\tau))}{\partial \boldsymbol{W}^{(h)}(\tau)} d\tau \right\|_F$$
$$= \left\| \int_0^t \frac{1}{n} \sum_{i=1}^n (u_i(\tau) - y_i) \frac{\partial f_{nn}(\boldsymbol{\theta}(\tau), \boldsymbol{x}_i)}{\partial \boldsymbol{W}^{(h)}} d\tau \right\|_F$$
$$\leq \frac{1}{n} \max_{0 \leq \tau \leq t} \sum_{i=1}^n \left\| \frac{\partial f_{nn}(\boldsymbol{\theta}(\tau), \boldsymbol{x}_i)}{\partial \boldsymbol{W}^{(h)}} \right\|_F \int_0^t \|\boldsymbol{u}_{nn}(\tau) - \boldsymbol{y}\|_2 d\tau$$
$$\leq \frac{1}{n} \max_{0 \leq \tau \leq t} \sum_{i=1}^n \left\| \frac{\partial f_{nn}(\boldsymbol{\theta}(\tau), \boldsymbol{x}_i)}{\partial \boldsymbol{W}^{(h)}} \right\|_F \int_0^t \exp\left(-\kappa^2 \lambda_0 \tau\right) d\tau \|\boldsymbol{u}_{nn}(0) - \boldsymbol{y}\|_2$$
$$\leq \frac{C_0}{\sqrt{n}\lambda_0} \max_{0 \leq \tau \leq t} \sum_{i=1}^n \left\| \frac{\partial f_{nn}(\boldsymbol{\theta}(\tau), \boldsymbol{x}_i)}{\partial \boldsymbol{W}^{(h)}} \right\|_F$$

$$\leq \frac{C_0}{\sqrt{n}\kappa^2\lambda_0} \max_{0\leq\tau\leq t} \sum_{i=1}^{n} \left( \left\| \frac{\partial f_{nn}(\boldsymbol{\theta}(0), \boldsymbol{x}_i)}{\partial \boldsymbol{W}^{(h)}} \right\|_F + \left\| \frac{\partial f_{nn}(\boldsymbol{\theta}(\tau), \boldsymbol{x}_i)}{\partial \boldsymbol{W}^{(h)}} - \frac{\partial f_{nn}(\boldsymbol{\theta}(0), \boldsymbol{x}_i)}{\partial \boldsymbol{W}^{(h)}} \right\|_F \right)$$

$$\leq \frac{C_1}{\sqrt{n}\lambda_0} \max_{0\leq\tau\leq t} \sum_{i=1}^{n} \left( \left\| \frac{\partial f_{nn}(\boldsymbol{\theta}(0), \boldsymbol{x}_i)}{\partial \boldsymbol{W}^{(h)}} \right\|_F + \left\| \frac{\partial f_{nn}(\boldsymbol{\theta}(\tau), \boldsymbol{x}_i)}{\partial \boldsymbol{W}^{(h)}} - \frac{\partial f_{nn}(\boldsymbol{\theta}(0), \boldsymbol{x}_i)}{\partial \boldsymbol{W}^{(h)}} \right\|_F \right)$$

$$\leq \frac{C_2\sqrt{n}}{\lambda_0} + \frac{C_1\sqrt{n}}{\lambda_0} \max_{0\leq\tau\leq t} \left( \left\| \frac{\partial f_{nn}(\boldsymbol{\theta}(\tau), \boldsymbol{x}_i)}{\partial \boldsymbol{W}^{(h)}} - \frac{\partial f_{nn}(\boldsymbol{\theta}(0), \boldsymbol{x}_i)}{\partial \boldsymbol{W}^{(h)}} \right\|_F \right).$$

The last step we used $\left\| \frac{\partial f_{nn}(\boldsymbol{\theta}(0), \boldsymbol{x}_i)}{\partial \boldsymbol{W}^{(h)}} \right\|_F = O(1)$. Suppose there exists $t$ such that $\left\| \boldsymbol{W}^{(h)}(t) - \boldsymbol{W}^{(h)}(0) \right\|_F > \omega\sqrt{m}$. Denote

$$t_0 = \operatorname{argmin}_{t\geq 0} \left\{ \left\| \boldsymbol{W}^{(h)}(t) - \boldsymbol{W}^{(h)}(0) \right\|_F > \omega\sqrt{m}. \right\}.$$

For any $t < t_0$, we know for all $h' \in [L+1]$, $\left\| \boldsymbol{W}^{(h')}(t) - \boldsymbol{W}^{(h')}(0) \right\|_2 \leq \omega\sqrt{m}$. Therefore, by Lemma F.6, we know

$$\left\| \frac{\partial f_{nn}(\boldsymbol{\theta}(t), \boldsymbol{x}_i)}{\partial \boldsymbol{W}^{(h)}} - \frac{\partial f_{nn}(\boldsymbol{\theta}(0), \boldsymbol{x}_i)}{\partial \boldsymbol{W}^{(h)}} \right\|_F = C\omega^{1/3}L^{5/2}.$$

Therefore, using the fact that $\omega$ is sufficiently small we can bound

$$\left\| \boldsymbol{W}^{(h)}(t_0) - \boldsymbol{W}^{(h)}(0) \right\|_F \leq \frac{C_3\sqrt{n}}{\lambda_0}.$$

Since we also know $m$ is sufficiently large to make $\omega\sqrt{m} > \frac{C_3\sqrt{n}}{\lambda_0}$, we have a contradiction. $\qquad\square$

The next lemma shows if all weight matrices only have small perturbation, then we still have linear convergence.

**Lemma F.8.** *Suppose $\omega = \operatorname{poly}(1/n, \lambda_0, 1/L, 1/\log(m), \epsilon, 1/\log(1/\delta), \kappa)$. Suppose for all $t \geq 0$ $\left\| \boldsymbol{W}^{(h)}(t) - \boldsymbol{W}^{(h)}(0) \right\|_F \leq \omega\sqrt{m}$ for $h \in [L+1]$. Then if $m = \operatorname{poly}(1/\omega)$, we have with probability at least $1 - \delta$ over random initialization, for all $t \geq 0$*

$$\| \boldsymbol{u}_{nn}(t) - \boldsymbol{y} \|_2 \leq \exp\left( -\frac{1}{2}\kappa^2\lambda_0 t \right) \| \boldsymbol{u}_{nn}(0) - \boldsymbol{y} \|_2.$$

*Proof.* Under this assumption and the result of initialization, we know for all $t \geq 0$, $\lambda_{\min}(\boldsymbol{H}(t)) \geq \frac{1}{2}\lambda_0$. This in turn directly imply the linear convergence result we want. $\qquad\square$

Lastly, with these lemmas at hand, using an argument similar to [Du et al., 2019], we can show during training, weight matrices do not move by much.

**Lemma F.9.** *Let $\omega \leq \operatorname{poly}(\epsilon, L, \lambda_0, 1/\log(m), 1/\log(1/\delta), \kappa, 1/n)$. If $m \geq \operatorname{poly}(1/\omega)$, then with probability at least $1 - \delta$ over random initialization, we have for all $t \geq 0$, for all $h \in [L+1]$ we have*

$$\left\| \boldsymbol{W}^{(h)}(t) - \boldsymbol{W}^{(h)}(0) \right\|_F \leq \omega\sqrt{m}$$

*and*

$$\| \boldsymbol{u}_{nn}(t) - \boldsymbol{y} \|_2 \leq \exp\left( -\frac{1}{2}\kappa^2\lambda_0 t \right) \| \boldsymbol{u}_{nn}(0) - \boldsymbol{y} \|_2.$$

*Proof.* Let

$$t_0 = \operatorname{argmin}_t \left\{ \exists h \in [L+1], \left\| \boldsymbol{W}^{(h)}(t) - \boldsymbol{W}^{(h)}(0) \right\|_F > \omega\sqrt{m} \right.$$

$$\left. \text{or } \| \boldsymbol{u}_{nn}(t) - \boldsymbol{y} \|_2 > \exp\left( -\frac{1}{2}\kappa^2\lambda_0 t \right) \| \boldsymbol{u}_{nn}(0) - \boldsymbol{y} \|_2 \right\}.$$

We analyze case by case. Suppose at time $t_0$, $\left\|\boldsymbol{W}^{(h)}(t_0) - \boldsymbol{W}^{(h)}(0)\right\|_F > \omega\sqrt{m}$. By Lemma F.7, we know there exists some $0 \leq t_1 < t_0$ such that either there exists $h' \neq h$ such that

$$\left\|\boldsymbol{W}^{(h')}(t_1) - \boldsymbol{W}^{(h')}(0)\right\|_F > \omega\sqrt{m}$$

or

$$\|\boldsymbol{u}_{nn}(t_1) - \boldsymbol{y}\|_2 > \exp\left(-\frac{1}{2}\kappa^2\lambda_0 t_1\right)\|\boldsymbol{u}_{nn}(0) - \boldsymbol{y}\|_2.$$

However, this violates the minimality of $t_0$. For the other case, if

$$\|\boldsymbol{u}_{nn}(t_0) - \boldsymbol{y}\|_2 > \exp\left(-\frac{1}{2}\kappa^2\lambda_0 t_0\right)\|\boldsymbol{u}_{nn}(0) - \boldsymbol{y}\|_2,$$

By Lemma F.8, we know there exists $t_1 < t_0$ such that there exists $h \in [L+1]$,

$$\left\|\boldsymbol{W}^{(h)}(t_1) - \boldsymbol{W}^{(h)}(0)\right\|_F > \omega\sqrt{m}.$$

However, again this violates the minimality of $t_0$. $\qquad\square$

Now we can finish the proof of Lemma F.2.

*Proof of Lemma F.2.* By Lemma F.9, we know for $t \to \infty$, $\left\|\boldsymbol{W}^{(h)}(t) - \boldsymbol{W}^{(h)}(0)\right\|_F \leq O\left(\omega\sqrt{m}\right)$ for if $\omega$ is sufficiently. Applying Lemma F.6, we know we only have a small perturbation on the gradient. Applying Lemma F.3, we know we only have small perturbation on kernel values. $\qquad\square$

# G    CNTK Derivation

In this section we derive CNTK for vanilla CNN. Given $\boldsymbol{x} \in \mathbb{R}^{P \times Q}$ and $(i,j) \in [P] \times [Q]$, we define

$$\phi_{ij}(\boldsymbol{x}) = [\boldsymbol{x}]_{i-(q-1)/2:i+(q-1)/2, j-(q-1)/2:j+(q-1)/2}$$

i.e., this operator extracts the $(i,j)$-th patch. By this definition, we can rewrite the CNN definition:

- Let $\boldsymbol{x}^{(0)} = \boldsymbol{x} \in \mathbb{R}^{P \times Q \times C^{(0)}}$ be the input image where $C^{(0)}$ is the number of channels in the input image.
- For $h = 1, \ldots, H$, $\beta = 1, \ldots, C^{(h)}$, the intermediate outputs are defined as

$$\left[\tilde{\boldsymbol{x}}_{(\beta)}^{(h)}\right]_{ij} = \sum_{\alpha=1}^{C^{(h-1)}} \left\langle \boldsymbol{W}_{(\alpha),(\beta)}^{(h)}, \phi_{ij}\left(x_{(\alpha)}^{(h-1)}\right)\right\rangle, \quad \boldsymbol{x}_{(\beta)}^{(h)} = \sqrt{\frac{c_\sigma}{C^{(h)} \times q \times q}}\sigma\left(\tilde{\boldsymbol{x}}_{(\beta)}^{(h)}\right)$$

  where each $\boldsymbol{W}_{(\alpha),(\beta)}^{(h)} \in \mathbb{R}^{q \times q}$ is a filter with Gaussian initialization.
- The final output is defined as

$$f(\boldsymbol{\theta}, \boldsymbol{x}) = \sum_{\alpha=1}^{C^{(L)}} \left\langle \boldsymbol{W}_{(\alpha)}^{(L)}, \boldsymbol{x}_{(\alpha)}^{(L)}\right\rangle$$

  where $\boldsymbol{W}_{(\alpha)}^{(L)} \in \mathbb{R}^{P \times Q}$ is a weight matrix with Gaussian initialization.

## G.1    Expansion of CNTK

We expand $\Theta^{(L)}(\boldsymbol{x}, \boldsymbol{x}')$ to show we can write it as the sum of $(L+1)$ terms with each term representing the inner product between the gradients with respect to the weight matrix of one layer. We first define an linear operator

$$\mathcal{L} : \mathbb{R}^{P \times Q \times P \times Q} \to \mathbb{R}^{P \times Q \times P \times Q}$$

$$[\mathcal{L}(\boldsymbol{M})]_{k\ell, k'\ell'} = \frac{c_\sigma}{q^2}\mathrm{tr}\left([\boldsymbol{M}]_{\mathcal{D}_{k\ell, k'\ell'}}\right). \tag{30}$$

This linear operator is induced from convolutional operation. And here use zero padding, namely when the subscription exceeds the range of $[P] \times [Q] \times [P] \times [Q]$, the value of the element is zero.

We also define $\boldsymbol{I} \in \mathbb{R}^{P \times Q \times P \times Q}$ as the identity tensor, namely $\boldsymbol{I}_{i,j,i',j'} = \mathbf{1}\{i = i', j = j'\}$. And

$$\text{Sum}\,(\boldsymbol{M}) = \sum_{(i,j,i',j') \in [P] \times [Q] \times [P] \times [Q]} \boldsymbol{M}_{i,j,i',j'}.$$

The following property of $\mathcal{L}$ is immediate by definition: $\forall \boldsymbol{M}, \boldsymbol{N} \in \mathbb{R}^{P \times Q \times P \times Q}$, we have

$$\text{Sum}\,(\boldsymbol{M} \odot \mathcal{L}(\boldsymbol{N})) = \text{Sum}\,(\mathcal{L}(\boldsymbol{M}) \odot \boldsymbol{N}). \tag{31}$$

With this operator, we can expand CNTK as (for simplicity we drop on $\boldsymbol{x}$ and $\boldsymbol{x}'$)

$$\begin{aligned}
&\Theta^{(L)} \\
=&\text{tr}\left(\dot{\boldsymbol{K}}^{(L)} \odot \Theta^{(H-1)} + \boldsymbol{K}^{(L)}\right) \\
=&\text{tr}\left(\boldsymbol{K}^{(L)}\right) + \text{tr}\left(\dot{\boldsymbol{K}}^{(L)} \odot \mathcal{L}\left(\boldsymbol{K}^{(H-1)}\right)\right) + \text{tr}\left(\dot{\boldsymbol{K}}^{(L)} \odot \mathcal{L}\left(\dot{\boldsymbol{K}}^{(H-1)} \odot \Theta^{(H-2)}\right)\right) \\
=&\ldots \\
=&\sum_{h=0}^{L} \text{tr}\left(\dot{\boldsymbol{K}}^{(L)} \odot \mathcal{L}\left(\dot{\boldsymbol{K}}^{(H-1)} \mathcal{L}\left(\cdots \dot{\boldsymbol{K}}^{(h+1)} \mathcal{L}\left(\boldsymbol{K}^{(h)}\right) \cdots\right)\right).\right)
\end{aligned}$$

Here for $h = H$, the term is just $\text{tr}\left(\boldsymbol{K}^{(L)}\right)$.

In the following, we will show

$$\begin{aligned}
\left\langle \frac{\partial f(\boldsymbol{\theta}, \boldsymbol{x})}{\partial \boldsymbol{W}^{(h)}}, \frac{\partial f(\boldsymbol{\theta}, \boldsymbol{x}')}{\partial \boldsymbol{W}^{(h)}} \right\rangle \approx& \text{tr}\left(\dot{\boldsymbol{K}}^{(L)} \odot \mathcal{L}\left(\dot{\boldsymbol{K}}^{(H-1)} \odot \mathcal{L}\left(\cdots \dot{\boldsymbol{K}}^{(h)} \odot \mathcal{L}\left(\boldsymbol{K}^{(h-1)}\right) \cdots\right)\right)\right) \\
=& \text{Sum}\left(\boldsymbol{I} \odot \dot{\boldsymbol{K}}^{(L)} \odot \mathcal{L}\left(\dot{\boldsymbol{K}}^{(H-1)} \odot \mathcal{L}\left(\cdots \dot{\boldsymbol{K}}^{(h)} \odot \mathcal{L}\left(\boldsymbol{K}^{(h-1)}\right) \cdots\right)\right)\right).
\end{aligned}$$

which could be rewritten as the following by Property 31,

$$\left\langle \frac{\partial f(\boldsymbol{\theta}, \boldsymbol{x})}{\partial \boldsymbol{W}^{(h)}}, \frac{\partial f(\boldsymbol{\theta}, \boldsymbol{x}')}{\partial \boldsymbol{W}^{(h)}} \right\rangle \approx \text{Sum}\left(\mathcal{L}\left(\boldsymbol{K}^{(h-1)}\right) \odot \dot{\boldsymbol{K}}^{(h)} \odot \mathcal{L}\left(\dot{\boldsymbol{K}}^{(h+1)} \cdots \odot \mathcal{L}\left(\boldsymbol{I} \odot \dot{\boldsymbol{K}}^{(L)}\right) \cdots\right)\right).$$

## G.2 Derivation

We first compute the derivative of the prediction with respect to one single filter.

$$\begin{aligned}
\frac{\partial f(\boldsymbol{\theta}, \boldsymbol{x})}{\partial \boldsymbol{W}^{(h)}_{(\alpha),(\beta)}} =& \left\langle \frac{\partial f(\boldsymbol{\theta}, \boldsymbol{x})}{\partial \boldsymbol{x}_{(\beta)}}, \frac{\partial \boldsymbol{x}^{(h)}_{(\beta)}}{\partial \boldsymbol{W}^{(h)}_{(\alpha),(\beta)}} \right\rangle \\
=& \sum_{(i,j) \in [P] \times [Q]} \left\langle \frac{\partial f(\boldsymbol{\theta}, \boldsymbol{x})}{[\boldsymbol{x}^{(h)}_{(\beta)}]_{ij}}, \frac{\partial [\boldsymbol{x}^{(h)}_{(\beta)}]_{ij}}{\partial \boldsymbol{W}^{(h)}_{(\alpha),(\beta)}} \right\rangle \\
=& \sum_{(i,j) \in [P] \times [Q]} \frac{\partial f(\boldsymbol{\theta}, \boldsymbol{x})}{[\boldsymbol{x}^{(h)}_{(\beta)}]_{ij}} \sqrt{\frac{c_\sigma}{C^{(h)} q^2}} \sigma'\left(\left[\tilde{\boldsymbol{x}}^{(h)}_{(\beta)}\right]_{ij}\right) \phi_{ij}(\boldsymbol{x}^{(h-1)}_{(\alpha)}).
\end{aligned}$$

With this expression, we proceed to we compute the inner product between gradients with respect to the $h$-th layer matrix

$$\sum_{\alpha=1}^{C^{(h-1)}} \sum_{\beta=1}^{C^{(h)}} \left\langle \frac{\partial f(\boldsymbol{\theta}, \boldsymbol{x})}{\partial \boldsymbol{W}^{(h)}_{(\alpha),(\beta)}}, \frac{\partial f(\boldsymbol{\theta}, \boldsymbol{x}')}{\partial \boldsymbol{W}^{(h)}_{(\alpha),(\beta)}} \right\rangle \tag{32}$$

$$= \sum_{(i,j,i',j')\in[P]\times[Q]\times[P]\times[Q]} \frac{c_\sigma}{C^{(h)}q^2} \sum_{\beta=1}^{C^{(h)}} \left( \frac{\partial f(\boldsymbol{\theta},\boldsymbol{x})}{\partial[\boldsymbol{x}_{(\beta)}^{(h)}]_{ij}} \cdot \frac{\partial f(\boldsymbol{\theta},\boldsymbol{x}')}{\partial[\boldsymbol{x}_{(\beta)}'^{(h)}]_{i'j'}} \right) \left( \sigma'\left( \left[\tilde{\boldsymbol{x}}_{(\beta)}^{(h)}\right]_{ij} \right) \sigma'\left( \left[\tilde{\boldsymbol{x}}_{(\beta)}'^{(h)}\right]_{i'j'} \right) \right)$$

$$\cdot \left( \sum_{\alpha=1}^{C^{(h-1)}} \left\langle \phi_{ij}(\boldsymbol{x}_{(\alpha)}^{(h-1)}), \phi_{i'j'}(\boldsymbol{x}_{(\alpha)}'^{(h-1)}) \right\rangle \right).$$

Similar to our derivation to NTK, we can use the following approximation

$$\left( \sum_{\alpha=1}^{C^{(h-1)}} \left\langle \phi_{ij}(\boldsymbol{x}_{(\alpha)}^{(h-1)}), \phi_{i'j'}(\boldsymbol{x}_{(\alpha)}'^{(h-1)}) \right\rangle \right) \approx \text{tr}\left( \left[\boldsymbol{K}^{(h-1)}\right]_{\mathcal{D}_{ij,i'j'}} \right) = \mathcal{L}\left( \boldsymbol{K}^{(h-1)} \right).$$

Thus it remains to show that $\forall (i,j,i',j') \in [P] \times [Q] \times [P] \times [Q]$,

$$\sum_{\beta=1}^{C^{(h)}} \frac{c_\sigma}{C^{(h)}q^2} \left( \frac{\partial f(\boldsymbol{\theta},\boldsymbol{x})}{\partial[\boldsymbol{x}_{(\beta)}^{(h)}]_{ij}} \cdot \frac{\partial f(\boldsymbol{\theta},\boldsymbol{x}')}{\partial[\boldsymbol{x}_{(\beta)}'^{(h)}]_{i'j'}} \right) \left( \sigma'\left( \left[\tilde{\boldsymbol{x}}_{(\beta)}^{(h)}\right]_{ij} \right) \sigma'\left( \left[\tilde{\boldsymbol{x}}_{(\beta)}'^{(h)}\right]_{i'j'} \right) \right)$$

$$\approx \left[ \mathcal{L}\left( \dot{\boldsymbol{K}}^{(h+1)} \cdots \odot \mathcal{L}\left( \boldsymbol{I} \odot \dot{\boldsymbol{K}}^{(L)} \right) \cdots \right) \odot \dot{\boldsymbol{K}}^{(h)} \right]_{i,j,i',j'}$$

The key step of this derivation is the following approximation (Equation 33), which assumes for each $(i,j,i',j')$, $\frac{\partial f(\boldsymbol{\theta},\boldsymbol{x})}{\partial[\boldsymbol{x}_{(\beta)}^{(h)}]_{ij}} \cdot \frac{\partial f(\boldsymbol{\theta},\boldsymbol{x}')}{\partial[\boldsymbol{x}_{(\beta)}'^{(h)}]_{i'j'}}$ and $\sigma'\left( \left[\tilde{\boldsymbol{x}}_{(\beta)}^{(h)}\right]_{ij} \right) \sigma'\left( \left[\tilde{\boldsymbol{x}}_{(\beta)}'^{(h)}\right]_{i'j'} \right)$ are independent. This is used and made rigorous for ReLU activation and fully-connected networks in the proof of Theorem 3.1. Yang [2019] gave a rigorous statement of this approximation in an asymptotic way for CNNs.

$$\frac{1}{C^{(h)}} \sum_{\beta=1}^{C^{(h)}} \left( \frac{\partial f(\boldsymbol{\theta},\boldsymbol{x})}{\partial[\boldsymbol{x}_{(\beta)}^{(h)}]_{ij}} \cdot \frac{\partial f(\boldsymbol{\theta},\boldsymbol{x}')}{\partial[\boldsymbol{x}_{(\beta)}'^{(h)}]_{i'j'}} \right) \left( \sigma'\left( \left[\tilde{\boldsymbol{x}}_{(\beta)}^{(h)}\right]_{ij} \right) \sigma'\left( \left[\tilde{\boldsymbol{x}}_{(\beta)}'^{(h)}\right]_{i'j'} \right) \right)$$

$$\approx \left( \frac{1}{C^{(h)}} \sum_{\beta=1}^{C^{(h)}} \frac{\partial f(\boldsymbol{\theta},\boldsymbol{x})}{\partial[\boldsymbol{x}_{(\beta)}^{(h)}]_{ij}} \cdot \frac{\partial f(\boldsymbol{\theta},\boldsymbol{x}')}{\partial[\boldsymbol{x}_{(\beta)}'^{(h)}]_{i'j'}} \right) \left( \frac{1}{C^{(h)}} \sum_{\beta=1}^{C^{(h)}} \sigma'\left( \left[\tilde{\boldsymbol{x}}_{(\beta)}^{(h)}\right]_{ij} \right) \sigma'\left( \left[\tilde{\boldsymbol{x}}_{(\beta)}'^{(h)}\right]_{i'j'} \right) \right) \tag{33}$$

Note that

$$\frac{c_\sigma}{C^{(h)}q^2} \sum_{\beta=1}^{C^{(h)}} \sigma'\left( \left[\tilde{\boldsymbol{x}}_{(\beta)}^{(h)}\right]_{ij} \right) \sigma'\left( \left[\tilde{\boldsymbol{x}}_{(\beta)}'^{(h)}\right]_{i'j'} \right) \approx \left[ \dot{\boldsymbol{K}}^{(h)}\left(\boldsymbol{x},\boldsymbol{x}'\right) \right]_{ij,i'j'},$$

the derivation is complete once we show

$$\boldsymbol{G}^{(h)}(\boldsymbol{x},\boldsymbol{x}',\boldsymbol{\theta}) := \frac{1}{C^{(h)}} \sum_{\beta=1}^{C^{(h)}} \frac{\partial f(\boldsymbol{\theta},\boldsymbol{x})}{\partial \boldsymbol{x}_{(\beta)}^{(h)}} \otimes \frac{\partial f(\boldsymbol{\theta},\boldsymbol{x}')}{\partial \boldsymbol{x}_{(\beta)}'^{(h)}} \approx \mathcal{L}\left( \dot{\boldsymbol{K}}^{(h+1)} \cdots \odot \mathcal{L}\left( \boldsymbol{I} \odot \dot{\boldsymbol{K}}^{(L)} \right) \cdots \right).$$
$$\tag{34}$$

Now, we tackle the term $\left( \frac{\partial f(\boldsymbol{\theta},\boldsymbol{x})}{\partial[\boldsymbol{x}_{(\beta)}^{(h)}]_{ij}} \cdot \frac{\partial f(\boldsymbol{\theta},\boldsymbol{x}')}{\partial[\boldsymbol{x}_{(\beta)}'^{(h)}]_{i'j'}} \right)$. Notice that

$$\frac{\partial f(\boldsymbol{\theta},\boldsymbol{x})}{\partial \left[ \boldsymbol{x}_{(\beta)}^{(h)} \right]_{ij}} = \sum_{(k,\ell)\in[P]\times[Q]} \frac{\partial f(\boldsymbol{\theta},\boldsymbol{x})}{\partial \left[ \boldsymbol{x}_{(\gamma)}^{(h+1)} \right]_{k\ell}} \frac{\partial \left[ \boldsymbol{x}_{(\gamma)}^{(h+1)} \right]_{k\ell}}{\partial \left[ \boldsymbol{x}_{(\beta)}^{(h)} \right]_{ij}}.$$

and for $\gamma \in [C^{(h+1)}]$ and $(k,\ell) \in [P] \times [Q]$

$$\frac{\partial \left[ \boldsymbol{x}_{(\gamma)}^{(h+1)} \right]_{k\ell}}{\partial \left[ \boldsymbol{x}_{(\beta)}^{(h)} \right]_{ij}} = \begin{cases} \sqrt{\frac{c_\sigma}{C^{(h+1)}q^2}} \sigma'\left( \left[\tilde{\boldsymbol{x}}_{(\gamma)}^{(h+1)}\right]_{k\ell} \right) \left[ \boldsymbol{W}_{(\beta),(\gamma)}^{(h+1)} \right]_{i-k+q-1,j-\ell+q-1} & \text{if } (i,j) \in \mathcal{D}_{k\ell} \\ 0 & \text{otherwise} \end{cases}.$$

We then have

$$\left[ \boldsymbol{G}^{(h)}(\boldsymbol{x}, \boldsymbol{x}', \boldsymbol{\theta}) \right]_{ij, i'j'} = \frac{1}{C^{(h)}} \sum_{\beta=1}^{C^{(h)}} \frac{\partial f(\boldsymbol{\theta}, \boldsymbol{x})}{\partial \left[ \boldsymbol{x}_{(\beta)}^{(h)} \right]_{ij}} \frac{\partial f(\boldsymbol{\theta}, \boldsymbol{x}')}{\partial \left[ \boldsymbol{x}_{(\beta)}'^{(h)} \right]_{i'j'}}$$

$$= \sum_{k, \ell, k', \ell'} \frac{c_\sigma}{C^{(h+1)} q^2} \sum_{\gamma=1}^{C^{(h+1)}} \left( \frac{\partial f(\boldsymbol{\theta}, \boldsymbol{x})}{\partial \left[ \boldsymbol{x}_{(\gamma)}^{(h+1)} \right]_{k\ell}} \frac{\partial f(\boldsymbol{\theta}, \boldsymbol{x}')}{\partial \left[ \boldsymbol{x}_{(\gamma)}'^{(h+1)} \right]_{k'\ell'}} \right) \left( \sigma' \left( \left[ \tilde{\boldsymbol{x}}_{(\gamma)}^{(h+1)} \right]_{k\ell} \right) \sigma' \left( \left[ \tilde{\boldsymbol{x}}_{(\gamma)}'^{(h+1)} \right]_{k'\ell'} \right) \right)$$

$$\cdot \frac{1}{C^{(h)}} \sum_{\beta=1}^{C^{(h)}} \mathbf{1} \left\{ (i, j, i', j') \in \mathcal{D}_{k\ell, k'\ell'} \right\} \left[ \boldsymbol{W}_{(\beta), (\gamma)}^{(h+1)} \right]_{i-k+q-1, j-\ell+q-1} \left[ \boldsymbol{W}_{(\beta), (\gamma)}^{(h+1)} \right]_{i'-k'+q-1, j'-\ell'+q-1}$$

$$\approx \sum_{k, \ell, k', \ell'} \frac{c_\sigma}{C^{(h+1)} q^2} \sum_{\gamma=1}^{C^{(h+1)}} \left( \frac{\partial f(\boldsymbol{\theta}, \boldsymbol{x})}{\partial \left[ \boldsymbol{x}_{(\gamma)}^{(h+1)} \right]_{k\ell}} \frac{\partial f(\boldsymbol{\theta}, \boldsymbol{x}')}{\partial \left[ \boldsymbol{x}_{(\gamma)}'^{(h+1)} \right]_{k'\ell'}} \right) \left( \sigma' \left( \left[ \tilde{\boldsymbol{x}}_{(\gamma)}^{(h+1)} \right]_{k\ell} \right) \sigma' \left( \left[ \tilde{\boldsymbol{x}}_{(\gamma)}'^{(h+1)} \right]_{k'\ell'} \right) \right)$$

$$\cdot \mathbf{1} \left\{ (i, j, i', j') \in \mathcal{D}_{k\ell, k'\ell'}, i - k = i' - k', j - \ell = j' - \ell' \right\}$$

$$\approx \sum_{k, \ell, k', \ell'} \left( \frac{1}{C^{(h+1)}} \sum_{\gamma=1}^{C^{(h+1)}} \frac{\partial f(\boldsymbol{\theta}, \boldsymbol{x})}{\partial \left[ \boldsymbol{x}_{(\gamma)}^{(h+1)} \right]_{k\ell}} \frac{\partial f(\boldsymbol{\theta}, \boldsymbol{x}')}{\partial \left[ \boldsymbol{x}_{(\gamma)}'^{(h+1)} \right]_{k'\ell'}} \right) \left( \frac{c_\sigma}{q^2 C^{(h+1)}} \sum_{\gamma=1}^{C^{(h+1)}} \sigma' \left( \left[ \tilde{\boldsymbol{x}}_{(\gamma)}^{(h+1)} \right]_{k\ell} \right) \sigma' \left( \left[ \tilde{\boldsymbol{x}}_{(\gamma)}'^{(h+1)} \right]_{k'\ell'} \right) \right)$$

$$\cdot \mathbf{1} \left\{ (i, j, i', j') \in \mathcal{D}_{k\ell, k'\ell'}, i - k = i' - k', j - \ell = j' - \ell' \right\}$$

$$\approx \sum_{k, \ell, k', \ell'} \left( \frac{1}{C^{(h+1)}} \sum_{\gamma=1}^{C^{(h+1)}} \frac{\partial f(\boldsymbol{\theta}, \boldsymbol{x})}{\partial \left[ \boldsymbol{x}_{(\gamma)}^{(h+1)} \right]_{k\ell}} \frac{\partial f(\boldsymbol{\theta}, \boldsymbol{x}')}{\partial \left[ \boldsymbol{x}_{(\gamma)}'^{(h+1)} \right]_{k'\ell'}} \right) \left[ \dot{\boldsymbol{K}}^{(h+1)} (\boldsymbol{x}, \boldsymbol{x}') \right]_{\ell k, \ell' k'}$$

$$\cdot \mathbf{1} \left\{ (i, j, i', j') \in \mathcal{D}_{k\ell, k'\ell'}, i - k = i' - k', j - \ell = j' - \ell' \right\}$$

$$\approx \mathrm{tr} \left( \left[ \boldsymbol{G}^{(h+1)}(\boldsymbol{x}, \boldsymbol{x}', \boldsymbol{\theta}) \odot \dot{\boldsymbol{K}}^{(h+1)} (\boldsymbol{x}, \boldsymbol{x}') \right]_{\mathcal{D}_{ij, i'j'}} \right) \tag{35}$$

where the first approximation is due to our initialization of $\boldsymbol{W}^{(h+1)}$. In other words, we've shown

$$\boldsymbol{G}^{(h)}(\boldsymbol{x}, \boldsymbol{x}', \boldsymbol{\theta}) = \mathcal{L} \left( \boldsymbol{G}^{(h+1)}(\boldsymbol{x}, \boldsymbol{x}', \boldsymbol{\theta}) \odot \dot{\boldsymbol{K}}^{(h+1)} (\boldsymbol{x}, \boldsymbol{x}') \right). \tag{36}$$

Since we use a fully-connected weight matrix as the last layer, we have $\boldsymbol{G}^{(L)}(\boldsymbol{x}, \boldsymbol{x}', \boldsymbol{\theta}) \approx \boldsymbol{I}$.

Thus by induction with Equation 36, we have derived Equation 34, which completes the derivation of CNTK.

For the derivation of CNTK-GAP, the only difference is due to the global average pooling layer(GAP), $\boldsymbol{G}^{(L)}(\boldsymbol{x}, \boldsymbol{x}', \boldsymbol{\theta}) \approx \frac{1}{Q^2 P^2} \mathbf{1} \otimes \mathbf{1}$, where $\mathbf{1} \otimes \mathbf{1} \in \mathbb{R}^{P \times Q \times P \times Q}$ is the all one tensor.

# H    Formula of CNTK with Global Average Pooling

In this section we define CNN with global average pooling considered in this paper and its corresponding CNTK formula.

**CNN definition.**

- Let $\boldsymbol{x} = \boldsymbol{x}^{(0)} \in \mathbb{R}^{P \times Q \times C^{(0)}}$ be the input image and $C^{(0)}$ is the number of initial channels.
- For $h = 1, \ldots, L, \beta = 1, \ldots, C^{(h)}$, the intermediate outputs are defined as

$$\tilde{\boldsymbol{x}}_{(\beta)}^{(h)} = \sum_{\alpha=1}^{C^{(h-1)}} \boldsymbol{W}_{(\alpha), (\beta)}^{(h)} * \boldsymbol{x}_{(\alpha)}^{(h-1)}, \quad \boldsymbol{x}_{(\beta)}^{(h)} = \sqrt{\frac{c_\sigma}{C^{(h)} \times q^{(h)} \times q^{(h)}}} \sigma \left( \tilde{\boldsymbol{x}}_{(\beta)}^{(h)} \right).$$

- The final output is defined as

$$f(\boldsymbol{\theta}, \boldsymbol{x}) = \sum_{\alpha=1}^{C^{(L)}} W_{(\alpha)}^{(L+1)} \left( \frac{1}{PQ} \sum_{(i,j) \in [P] \times [Q]} \left[ \boldsymbol{x}_{(\alpha)}^{(L)} \right]_{ij} \right).$$

where $W^{(L+1)}_{(\alpha)} \in \mathbb{R}$ is a scalar with Gaussian initialization.

Besides using global average pooling, another modification is that we do not train the first and the layer. This is inspired by Du et al. [2018a] in which authors showed that if one applies gradient flow, then at any training time $t$, the difference between the squared Frobenius norm of the weight matrix at time $t$ and that at initialization is same for all layers. However, note that $\boldsymbol{W}^{(1)}$ and $\boldsymbol{W}^{(L+1)}$ are special because they are smaller matrices compared with other intermediate weight matrices, so *relatively*, these two weight matrices change more than the intermediate matrices during the training process, and this may dramatically change the kernel. Therefore, we choose to fix $\boldsymbol{W}^{(1)}$ and $\boldsymbol{W}^{(L+1)}$ to the make over-parameterization theory closer to practice.

**CNTK formula.** We let $\boldsymbol{x}, \boldsymbol{x}'$ be two input images. Note because CNN with global average pooling and vanilla CNN shares the same architecture except the last layer, $\boldsymbol{\Sigma}^{(h)}(\boldsymbol{x}, \boldsymbol{x}')$, $\dot{\boldsymbol{\Sigma}}^{(h)}(\boldsymbol{x}, \boldsymbol{x}')$ and $\boldsymbol{K}^{(h)}(\boldsymbol{x}, \boldsymbol{x}')$ are the same for these two architectures. the only difference is in calculating the final kernel value. To compute the final kernel value, we use the following procedure.

1. First, we define $\boldsymbol{\Theta}^{(0)}(\boldsymbol{x}, \boldsymbol{x}') = \boldsymbol{0}$. Note this is different from CNTK for vanilla CNN which uses $\boldsymbol{\Sigma}^{(0)}$ as the initial value because we do not train the first layer.
2. For $h = 1, \ldots, L-1$ and $(i, j, i', j') \in [P] \times [Q] \times [P] \times [Q]$, we define

$$
\left[ \boldsymbol{\Theta}^{(h)}(\boldsymbol{x}, \boldsymbol{x}') \right]_{ij,i'j'} = \operatorname{tr} \left( \left[ \dot{\boldsymbol{K}}^{(h)}(\boldsymbol{x}, \boldsymbol{x}') \odot \boldsymbol{\Theta}^{(h-1)}(\boldsymbol{x}, \boldsymbol{x}') + \boldsymbol{K}^{(h)}(\boldsymbol{x}, \boldsymbol{x}') \right]_{D_{ij,i'j'}} \right).
$$

3. For $h = L$, we define $\quad \boldsymbol{\Theta}^{(L)}(\boldsymbol{x}, \boldsymbol{x}') = \dot{\boldsymbol{K}}^{(L)}(\boldsymbol{x}, \boldsymbol{x}') \odot \boldsymbol{\Theta}^{(L-1)}(\boldsymbol{x}, \boldsymbol{x}')$.
4. Lastly, the final kernel value is defined as

$$
\frac{1}{P^2 Q^2} \sum_{(i,j,i',j') \in [P] \times [Q] \times [P] \times [Q]} \left[ \boldsymbol{\Theta}^{(L)}(\boldsymbol{x}, \boldsymbol{x}') \right]_{ij,i'j'}.
$$

Note that we ignore $\boldsymbol{K}^{(L)}$ comparing with the CNTK of CNN. This is because we do not train the last layer. The other difference is we calculate the mean over all entries, instead of calculating the summation over the diagonal ones. This is because we use global average pooling so the cross-variances between every two patches will contribute to the kernel.

# I  Fast Computation for ReLU-Activated CNTK

In this section we present our approach to compute CNTK *exactly*. Notably, most computation required by our new approach can be described as entry-wise operations over matrices and tensors, which allows efficient implementations on GPUs.

Following the formulas in Sections 4 and H, the trickiest part is computing the expectation of the post-activation output, i.e., Equations (11) and (12). These two expectations depend on (the same) $2 \times 2$ matrices $\left[ \boldsymbol{\Lambda}^{(h)}(\boldsymbol{x}, \boldsymbol{x}') \right]_{ij,i'j'}$. To obtain faster implementations, our key observation is that if the diagonal entries of $\left[ \boldsymbol{\Lambda}^{(h)}(\boldsymbol{x}, \boldsymbol{x}') \right]_{ij,i'j'}$ are all ones and the activation function is ReLU, there are closed-form formulas for the the corresponding expectations. To see this, let us suppose for now that $\boldsymbol{\Lambda} = \begin{pmatrix} 1 & \lambda \\ \lambda & 1 \end{pmatrix}$ for some $|\lambda| \leq 1$. When the activation function $\sigma(\cdot)$ is ReLU, one can show that

$$
\mathbb{E}_{(u,v) \sim \mathcal{N}(\boldsymbol{0}, \boldsymbol{\Lambda})} \left[ \sigma(u) \sigma(v) \right] = \frac{\lambda(\pi - \arccos(\lambda)) + \sqrt{1 - \lambda^2}}{2\pi} \tag{37}
$$

and

$$
\mathbb{E}_{(u,v) \sim \mathcal{N}(\boldsymbol{0}, \boldsymbol{\Lambda})} \left[ \dot{\sigma}(u) \dot{\sigma}(v) \right] = \frac{\pi - \arccos(\lambda)}{2\pi}. \tag{38}
$$

Now we let

$$
\boldsymbol{A}^{(h)} = \begin{pmatrix} \boldsymbol{\Sigma}^{(h-1)}(\boldsymbol{x}, \boldsymbol{x}) & \boldsymbol{\Sigma}^{(h-1)}(\boldsymbol{x}, \boldsymbol{x}') \\ \boldsymbol{\Sigma}^{(h-1)}(\boldsymbol{x}', \boldsymbol{x}) & \boldsymbol{\Sigma}^{(h-1)}(\boldsymbol{x}', \boldsymbol{x}') \end{pmatrix} \in \mathbb{R}^{2PQ \times 2PQ}.
$$

Here, we interpret $\boldsymbol{\Sigma}^{(h-1)}(\boldsymbol{x}, \boldsymbol{x})$, $\boldsymbol{\Sigma}^{(h-1)}(\boldsymbol{x}, \boldsymbol{x}')$, $\boldsymbol{\Sigma}^{(h-1)}(\boldsymbol{x}', \boldsymbol{x})$ and $\boldsymbol{\Sigma}^{(h-1)}(\boldsymbol{x}', \boldsymbol{x}')$ as matrices of size $PQ \times PQ$. If the diagonal entries of $A^{(h)}$ are all ones, then the diagonal entries of $\left[\boldsymbol{\Lambda}^{(h)}(\boldsymbol{x}, \boldsymbol{x}')\right]_{ij, i'j'}$ are all ones for all possible $(i, j, i', j') \in [P] \times [Q] \times [P] \times [Q]$, in which case we can calculate $\boldsymbol{K}^{(h)}(\boldsymbol{x}, \boldsymbol{x}')$ and $\dot{\boldsymbol{K}}^{(h)}(\boldsymbol{x}, \boldsymbol{x}')$ by simply applying the closed-form formulas described in (37) and (38) on $A^{(h)}$.

However, in general, the diagonal entries of $\boldsymbol{A}^{(h)}$ are not always all ones, in which case we resort to the homogeneity of the ReLU activation function. Suppose $\boldsymbol{\Lambda} = \begin{pmatrix} 1 & \lambda \\ \lambda & 1 \end{pmatrix}$ for some $|\lambda| \leq 1$, and $\boldsymbol{D} = \begin{pmatrix} c_1 & 0 \\ 0 & c_2 \end{pmatrix}$ for some $c_1, c_2 \geq 0$, then one can show that

$$\mathop{\mathbb{E}}_{(u,v) \sim \mathcal{N}(\boldsymbol{0}, \boldsymbol{D}\boldsymbol{\Lambda}\boldsymbol{D})} [\sigma(u)\sigma(v)] = \frac{\lambda(\pi - \arccos(\lambda)) + \sqrt{1 - \lambda^2}}{2\pi} \cdot c_1 c_2 \tag{39}$$

and

$$\mathop{\mathbb{E}}_{(u,v) \sim \mathcal{N}(\boldsymbol{0}, \boldsymbol{D}\boldsymbol{\Lambda}\boldsymbol{D})} [\dot{\sigma}(u)\dot{\sigma}(v)] = \frac{\pi - \arccos(\lambda)}{2\pi}. \tag{40}$$

Inspired by this, our final approach is described as follows.

1. Let $\boldsymbol{D} = \begin{pmatrix} \boldsymbol{D}_{\boldsymbol{x}} & \boldsymbol{0} \\ \boldsymbol{0} & \boldsymbol{D}_{\boldsymbol{x}'} \end{pmatrix}$, where $\boldsymbol{D}_{\boldsymbol{x}}$ and $\boldsymbol{D}_{\boldsymbol{x}'}$ are diagonal matrices whose diagonal entries are square roots of the diagonal entries of $\boldsymbol{\Sigma}^{(h-1)}(\boldsymbol{x}, \boldsymbol{x})$ and $\boldsymbol{\Sigma}^{(h-1)}(\boldsymbol{x}', \boldsymbol{x}')$, respectively.

2. Applying Equations (39) and (40) on $\boldsymbol{A}^{(h)} = \boldsymbol{D}\boldsymbol{\Lambda}^{(h)}\boldsymbol{D}$, where the diagonal entries of $\boldsymbol{\Lambda}^{(h)}$ are all ones.

Notice that the implementation above requires us to store the whole $\boldsymbol{A}^{(h)}$ matrix, which has size $2PQ \times 2PQ$. To further optimize the efficiency, we notice that to implement the approach described above, we only need to store the diagonal entries of $\boldsymbol{\Sigma}^{(h-1)}(\boldsymbol{x}, \boldsymbol{x})$ and $\boldsymbol{\Sigma}^{(h-1)}(\boldsymbol{x}', \boldsymbol{x}')$, together with the matrix $\boldsymbol{\Sigma}^{(h-1)}(\boldsymbol{x}, \boldsymbol{x}')$, which has size $PQ \times PQ$.