[Reviews · NeurIPS 2019]

Reviewer 1



The claim of this paper is clear and easy to read. Although the composition of CNTK is straightforward and its exact computation in ReLU networks seems to be a bit trivial, this is a natural direction to be explored and some researchers will be interested in experimental results reported in this paper. The non-asymptotic proof is also a novel contribution and will be helpful to further develop the theory of NTK. Basically, I believe that this paper is worth to be published. I think that clarifying the following points will further increase the significance of this paper. - In table 1, CNN-V (or CNN-GAP) trained by SGD achieved the highest performance compared to CNTKs. When the number of channels increase, does the performance of CNN-V (or CNN-GAP) converge to that of CNTKs? Because Authors argue the equivalence between the sufficiently wide net and kernel regression in the former part of the manuscript, some readers will be interested in experimental justification on such equivalence in CNNs. It will be also informative to empirically demonstrate in what finite number of channels the trained CNNs perform better than the kernel regression with the CNTKs. - In main theorems, the width m is evaluated by using the notation of poly(…), i.e., m >= poly(…). It seems to be almost enough to show that the lower bound is polynomial, it will be also helpful to show the specific rate of the polynomial order (I mean, when m>=poly(n), what is k satisfying m>=O(n^k)?). Such rates are explicitly shown in a part of lemmas and theorems (ex. Theorem E.1 and E.2), it is hard to follow them in the proofs of main theorems. It seems to be better to add a brief description of the rate of the polynomial order to Supplementary Material. - It may be better to remark on locally-connected network (LCN; CNN without weight sharing). [Novak et al 2019] claims that the kernel matrix (corresponding to weakly-trained nets) without global average pooling is equivalent between CNN and LCN. I am curious as to whether this equivalence holds in the case of NTKs. As shown in Table 1, the CNTK-GAP performed better than CNTK-V. This might be because the pooling makes correlations between different patches non-negligible and the advantage of weight sharing appear. - Lemma E.3 : I cannot fully understand the meaning of "random vector F=w^{\top}G". Does this mean that F is a fixed random vector independent of w and w is generated under the constraint of F=w^{\top}G after fixing G and F? It would be better to enrich description on its definitions. Typo: In Lemma F.9: m >= poly(\omega) -> m >= poly(1/\omega) --- After the rebuttal --- I have been almost satisfied with the authors' response and keep my score. I hope that Authors can improve the polynomial rate of the bounds in follow-up works.

Reviewer 2



POST-REBUTTAL UPDATE: I have read the rebuttal and other reviews; my score remains unchanged. ======================================================= Originality: This paper builds upon a line of works on the equivalence of DNNs and kernel methods and adds results for fully-trained CNNs to that list. The authors also prove a new convergence theorem for neural tangent kernels with a better convergence bound. Quality: The paper seems technically sound. However, I did not verify the provided proofs. Clarity: The paper is well-organized and clearly written. Significance: The provided experiments show that the proposed kernel outperforms existing kernels on CIFAR-10. It would be nice to see the performance on other datasets as well. It is not clear from the experiments how significant the results from the 21-layer kernel are. Does its performance degrade further with depth? Or is the difference between a 21-layer kernel and an 11-layer kernel not significant? Also, the idea of using less data for training to aid model selection is very widespread in the neural architecture search community. It is not clear why this could have been any different for CNTKs. I don't think these experiments are very relevant to the scope of the paper unless the authors show, for example, that CNTKs generalize better from less data or can be used for the design of finite CNNs. Finally,

Reviewer 3



The paper extends the previous fully-connected-layer based Neural Tangent Kernel (NTK) to Convolutional NTK (CNTK) and provides the first efficient exact computation algorithm. It results in a significant new benchmark for the performance of the pure kernel-based method on CIFAR-10. **Pros:** 1. The paper is clearly written and well-structured. 2. Theoretically, compared to previous results, it provides a non-asymptotic bound for NTK and looses the (sequential) limit of width requirements. It also shows the equivalence of a fully-trained sufficiently wide neural net and the deterministic kernel predictor using NTK. The theoretical analysis part is solid and important. 3. Technically, it extends NTK to CNTK (focusing on ReLU-based CNNs) for exact CNTK computation and provides some tricks (rely on the ReLU activation) to optimize the computation efficiency. It is important for both researchers practitioners, e.g., better understand the generalization performance of deep learning. **Cons:** 1. The section I in the appendix specifies the implementation tricks for GPUs, but the benefits (e.g., effectiveness, improved efficiency) of these tricks are unclear. A simple justification through some benchmarking experiments could be helpful. It would be great if the authors can also comment on the potential of giving the efficient exact algorithm to compute CNTK for other activation functions. 2. The lack of experimental details makes benchmarking results a bit less convincing. For example, what is the number of channels used for each layer, and what is its range? I think line 388 shows the number of channels is treated as a hyper-parameter and the best performance is reported among all hyper-parameters. Is it indicate that different depths might have different # of channels, or even different convolutional layers might have different # of channels? Is it possible to show a figure similar to Figure 6 in [Novak et al., 2019]? Otherwise, the comment (line 290) is less conclusive. Also, more justifications are required, e.g., why CNN-GAP suddenly has a quality loss for depth-11 but improves again for depth-21. BTW, how many repeated evaluations do Table 1 perform and what score is reported (average, or median as in Table 2)? **Minor comments:** 1. Line 126, ‘out’ should be ‘our’ 2. The proposed CNTK extension and its exact computation is a combination of the existing NTK with dynamic programming (specific to ReLU activation function). This part (Section 4) is a bit technical (with limited novelty) and independent to Section 3, while the title/introduction indeed is more focus on Section 4. Strengthening Section 4, e.g., including the residual layer, extending to more activation functions, or supporting softmax and cross-entropy, would make the whole story more solid. Open-sourcing an efficient NTK/CNTK computation framework could also connect these theoretical understanding to deep learning practitioners.

Reviewer 4



This paper provides two theoretical analyses of the neural tangent kernel (NTK); (i) one is the convergence guarantee of approximate NTK by finite-width DNNs to NTK and (ii) the other is a uniform bound on the gap between the limit point of gradient flow and a solution in RKHS associated with NTK. In addition, the paper introduces an efficient algorithm to compute a CNTK which is a CNN variant of NTK and experimentally demonstrate the superior performance of CNTK over the state-of-the-art kernel method and a random feature method of NTK. In short, I think this paper makes strong contributions to NeurIPS community as commented below. Clarity: The paper is well organized and easy to read. Quality: The work is of good quality and is technically sound. Two suggestions: - However, a major concern is the positivity of the gram-matrix $H^*$ of NTK. In the theoretical analysis, the positivity is assumed to define a solution in RKHS with NTK and to provide a convergence guarantee of the gradient flow. Therefore, I would like the author to discuss the validity of this assumption if possible. - It would be useful for the reader if the proof of the convergence of the linear dynamics (4) to a solution $f_{ntk}$ could be provided. Originality & Significance: In this paper, there are two interesting theoretical contributions. Theorem 3.1 refines the previous result obtained in the original NTK paper [Jacot2018+] via the non-asymptotic convergence analysis. In addition, Theorem 3.2 seems the first result that provides a uniform bound on the gap between the limit point of the gradient flow for DNN and a solution of NTK. In other words, these results connect DNNs with kernel methods and allow us to analyze the theoretical property of DNN trained by the gradient methods via a kernel perspective. Although there is still a non-negligible gap between the empirical performance of NTK and CNNs using several heuristics (which are turned off in the experiment) and there is also controversy over whether or not NTK is really useful in generalization analysis, this paper provides theoretical and empirical evidence of its usefulness to some extent. I believe that this paper will facilitate theoretical analysis of neural networks based on NTK. ----- I have read the authors' response and I will keep the score.

[Author Response · NeurIPS 2019]

We thank all the reviewers for their valuable feedback and appreciating our contributions. Please find our response to each individual reviewer below.

—— **To Reviewer #1** ——

**Empirically verify the equivalence between CNN and CNTK.** It would indeed be very interesting to empirically verify the equivalence. Note that the theoretical equivalence requires near-zero initialization, gradient flow (small learning rate), and a large number of channels. These require significant computation resource. We plan to conduct more thorough experiments along this line.

**Specific bound on $m$.** Our proof is based on the result in [Allen-zhu et al., 2018b], for which one needs a large polynomial like $m = \Omega(n^{24}L^{12})$. We believe such bound is not informative and can be significantly improved, so we did not specify the bound in the paper. Nevertheless we are happy to include it in the final version.

**LCN and CNN.** This is a good point. This equivalence still holds for CNTK-V but not for CNTK-GAP, because in CNTK-V we do not compute the cross-variance between patches at different positions but in CNTK-GAP we do (line 797). Thanks for the remark. We will add discussions in the final version.

**Notation in Lemma E.3.** Your explanation is correct. We will describe this more clearly. Thanks for pointing out!

—— **To Reviewer #5** ——

**Performance on other datasets.** Thanks for the suggestion. We will conduct experiments on other datasets, including CIFAR100 and SVHN, in the final version of the paper, and open-source our implementation.

**Significance of depth in the performance of kernels.** We have not gone beyond 21 layers due to computation constraints, but the difference between the 21-layer kernel and the 11-layer kernel doesn't seem significant. It would be a very interesting future direction to investigate the effect of depth on the performance.

**Using NTK for neural architecture search.** One of the advantages of kernel methods is that they require **little computation** on a small dataset, which is a very appealing feature for architecture search. So we believe using the relevance between CNTKs and CNNs, we can develop more effective CNNs based on the performance of CNTKs on small datasets. We will stress this point more clearly in the final version.

**Actual computation time for kernels.** For 21-layer CNTK, we used 1000 GPU hours.

**Generalization to other losses.** Once we have the kernel matrix, we can also minimize the cross-entropy loss on the prediction space. The current result for the equivalence between NN and NTK cannot generalize to other losses though.

**Why is global average pooling so crucial?** This is a very interesting question worth further investigation. A possible explanation is that global average pooling may impose a certain data augmentation effect.

—— **To Reviewer #7** ——

**Benefit of GPU implementation tricks & generalization to other activation.** The benefit is significant. We were able to compute the 21-layer CNTK on CIFAR-10 within 1000 GPU hours, while without these tricks this computation is simply infeasible (as mentioned in [Novak et al., 2019]). Note that another aspect that leads to the speed up is that we wrote native CUDA code. The efficient algorithm in Section I relies on homogeneity of the activation, so it can be generalized to Leaky ReLU. It is not clear how to deal with other activation functions.

**Experimental details.** We will describe the experimental details further in the full version. In particular: (1) # channels: The # channels is between 256 and 1024 in our experiments. All convolutional layers in a network have the same # channels. (2) Show a figure similar to Figure 6 in [Novak et al., 2019]? Thanks for the great suggestion. We will add this experiment in the final version of the paper.

**Strengthen Section 4.** The NTK formula for residual layers is not trivial to derive. It is possible to generalize to other loss functions (see response to Reviewer #5). We will open-source our implementation.

—— **To Reviewer #9** ——

**Positivity of $\boldsymbol{H}^*$.** The positivity of $\boldsymbol{H}^*$ is proved in [Du et al., 2019] for two-layer ReLU NN and in [Du et al., 2018b] for multi-layer NN with smooth activation. Since our setting is very similar to these two papers' settings, we believe $H^*$ is positive definite in general. Furthermore, experimentally we find that $H^*$ is always positive definite. We will add this discuss to the final version.

**Proof of convergence to $f_{ntk}$.** Thanks for the suggestion, and we will add a proof to the final version. It can be derived by looking at the dynamics of $f(\boldsymbol{\theta}(t), \boldsymbol{x})$ for a given test input $\boldsymbol{x}$.

[Meta-Review · NeurIPS 2019]

This paper has two main contributions. First, the convolutional extension of the neural tangent kernel (CNTK) was proposed and then an algorithm using CNTKs for "exact computation with an infinitely wide neural net" was designed. The algorithm allows squared-loss kernel regression with CNTKs corresponding to infinitely wide vanilla CNNs with ReLU activation as well as those also with global average pooling. Its time complexity is linear in the depth and quadratic in the amount of data and the height and width of the images. This time complexity, in previous papers, was believed to be impossible. Second, it has the first non-asymptotic proof of the equivalence between a fully-trained sufficiently wide neural net and the kernel regression predictor using NTK. The clarity, the novelty, and the significance are all above the corresponding thresholds of NeurIPS and thus it should clearly be accepted. I personally think this paper could even be a nice oral or spotlight presentation. The paper looks very theoretical at first glance, but in fact it is not---it is more than pure theory and is definitely of practical interests. The authors and the expert reviewers all ranked the first contribution higher; the experimental observations and thoughts on them in lines 273--293 should be interesting to the majority of machine learning researchers in the world. So at least to me, this paper is half theoretical and half empirical, and the second half is much more important. This paper (and the papers it relies on) has shown neural net is not the only thing that can go deep---kernel can go deep too as in Eq (9) and the equation below line 259 and also perform reasonably good as in Table 1. I do think understanding kernel methods better would help us understanding deep methods better. Therefore, I am recommending an oral or spotlight in order to draw the attention of DL practitioners to take a look at DL theory and consider what our next step should be (note that I am not from an area of any type of theory and I don't often stand on the theory side). In order to address the broader audience in the whole NeurIPS, I offer my quick thoughts on the paper (I didn't check the appendices due to limited time): A. The first impression (i.e., a theory paper) may stop many guys from going through your paper and reduce the impact. A possible reason may be that the paper has no figure and only one table. Since the current version uses itemize environments a lot, you may consider to make some "colorful" figures instead of nested lists. B. I feel that the messages in lines 273--293 are not fully conveyed in the introduction where only a subset of them is given. You may consider to separate the experimental observations from "our contributions", compress those observations and insert all of them after "our contributions". C. Up to now I am not sure who first proposed the concept of CNTKs due to some writing issues. It's a bit strange to me to say "one may also generalize the NTK to convolutional neural nets, and we call the corresponding kernel Convolutional Neural Tangent Kernel (CNTK)" because "one" suggests it's someone else. Moreover, you said "the random feature methods for approximating CNTK in earlier work" which again suggests CNTK exists already in earlier work. When I evaluated the novelty of the paper, I assumed this concept was proposed here, but this point should be clarified. D. Some claims are not supported by citations, for example in the introduction: It has long been known that weakly-trained convolutional nets have reasonable performance on MNIST and CIFAR-10. Weakly-trained nets that are fully-connected instead of convolutional, can also be thought of as “multi-layer random kitchen sinks,” which also have a long history. You are responsible to demonstrate "long been known" and "long history". The final version should be as friendly as possible to everybody attending the conference. Last but not least, there are still two concerns from the reviewers: one is about "specific bound on m" and one is about "positivity of H*". For the second concern, the reviewer thinks this theoretical guarantee should really be provided. Please address them in the final version.